The osteology and affinities of Eotyrannus lengi, a tyrannosauroid theropod from the Wealden Supergroup of southern England

Naish Darren 1 eotyrannus@gmail.com
Cau Andrea 2
1 School of Biological Sciences, Faculty of Environment and Life Sciences, University of Southampton , Southampton , UK
2 Independent , Parma , Italy
Farke Andrew
Electronic publication date: 2022 Jul 7
Publication date: 2022
Volume: 10
Electronic Location ID: e12727
Received 2017 Nov 27; Accepted 2021 Dec 10
Copyright: © 2022 Naish and Cau
Copyright year: 2022
Copyright holder: Naish and Cau
License: This is an open access article distributed under the terms of the Creative Commons Attribution License, which permits unrestricted use, distribution, reproduction and adaptation in any medium and for any purpose provided that it is properly attributed. For attribution, the original author(s), title, publication source (PeerJ) and either DOI or URL of the article must be cited.
License URL: https://creativecommons.org/licenses/by/4.0/

Keywords: Theropods, Tyrannosauroids, Dinosaurs, Cretaceous, Wealden, Coelurosaurs, Phylogeny

Funding: Gofundme campaign The publication costs of this manuscript were supported by a successful gofundme campaign of July 2018 (https://www.gofundme.com/the-eotyrannus-monograph) which raised the required amount in less than 24 h: (in alphabetical order) Devin Adler, Marko Bosscher, Neil Edmond, Dan Folkes, Everything Dinosaur, Timon Salar Gutleb, Lindsey Ireland, Ivan Kay, Daniel Laird, Alexander Lovegrove, Zach Miller, Dylan Thomas Murphree, Francesco Naldi, David Orr, Tom Parker, James Pascoe, Check Pell, Elizabeth Pendleton, Dick Poelen, Nathan Redland, Mark Scherz, Christina Shears-Ozeki, Rob Vickery, Mark Wildman, Grace Yon, Juan Yu See and five anonymous benefactors. The funders had no role in study design, data collection and analysis, decision to publish, or preparation of the manuscript.

==============================
Eotyrannus lengi Hutt et al., 2001 from the Lower Cretaceous Wessex Formation (part of the Wealden Supergroup) of the Isle of Wight, southern England, is described in detail, compared with other theropods, and evaluated in a new phylogenetic analysis. Eotyrannus is represented by a single individual that would have been c. 4.5 m long; it preserves the anterior part of the skull, a partial forelimb and pectoral girdle, various cervical, dorsal and caudal vertebrae, rib fragments, part of the ilium, and hindlimb elements excluding the femur. Lack of fusion with regard to both neurocentral and sacral sutures indicates subadult status. Eotyrannus possesses thickened, fused, pneumatic nasals with deep lateral recesses, elongate, tridactyl forelimbs and a tyrannosaurid-like scapulocoracoid. The short preantorbital ramus of the maxilla and nasals that are approximately seven times longer than they are wide show that Eotyrannus was not longirostrine. A posterodorsally inclined ridge on the ilium’s lateral surface fails to reach the dorsal margin: a configuration seen elsewhere in Juratyrant. Eotyrannus is not arctometatarsalian. Autapomorphies include the presence of curving furrows on the dentary, a block-like humeral entepicondyle, and a distoproximally aligned channel close to the distolateral border of the tibia. Within Tyrannosauroidea, E. lengi is phylogenetically intermediate between Proceratosauridae and Yutyrannus and the clade that includes Xiongguanlong, Megaraptora, Dryptosaurus and Tyrannosauridae. We do not find support for a close affinity between Eotyrannus and Juratyrant. Our analysis supports the inclusion of Megaraptora within Tyrannosauroidea and thus increases Cretaceous tyrannosauroid diversity and disparity. A proposal that Eotyrannus might belong within Megaraptora, however, is based on character states not present in the taxon. Several theropods from the Wessex Formation are based on material that overlaps with the E. lengi holotype but none can be shown to be synonymous with it.

Introduction

The remains of theropod dinosaurs have been known from the Wessex Formation of the Lower Cretaceous Wealden Group of the Isle of Wight, southern England, since the 1860s. Adequate, associated skeletons of Wessex Formation theropods were, however, unknown prior to 1978 when the holotype of the carcharodontosaurian allosauroid Neovenator salerii was discovered (Hutt, Martill & Barker, 1996). This taxon was monographed by Brusatte, Benson & Hutt (2008) and phylogenetic work indicates that it is part of a carcharodontosaurian clade that includes diverse allosauroid taxa (Benson, Carrano & Brusatte, 2010). Additional, fragmentary and isolated remains reveal the presence of non-coelurosaurian tetanurans, baryonychine spinosaurids, non-maniraptoran coelurosaurs and maniraptorans in the Wessex Formation (Naish, Hutt & Martill, 2001; Sweetman, 2004; Benson et al., 2009; Naish, 2011). Several names, including Calamospondylus oweni, Aristosuchus pusillus, Calamosaurus foxi and Thecocoelurus daviesi, are attached to certain of these specimens. A tendency to report and name newly discovered specimens, and to re-interpret them on a regular basis, has resulted in a complex taxonomy and a list of nomina dubia (Naish, Hutt & Martill, 2001; Naish, 2011).

A second associated Wessex Formation theropod was discovered on the Isle of Wight in 1997 and described in 2001. Given the large number of Wessex Formation theropod taxa named for fragmentary remains, it was initially assumed that the new specimen would prove referable to one of them. This proved not to be the case and the specimen was found to represent a new taxon, Eotyrannus lengi Hutt, Naish, Martill, Barker and Newbery, 2001. Hutt et al.’s (2001) primary contention was that E. lengi was a member of the tyrannosaur lineage, and specifically a non-tyrannosaurid tyrannosauroid. E. lengi has been discussed and partially illustrated in several publications since the appearance of that original paper (Holtz, 2004; Naish, Hutt & Martill, 2001; Naish & Martill, 2007; Naish, 2011) but a comprehensive description and analysis has been absent until now.

E. lengi is of substantial interest to those who specialise on the Lower Cretaceous theropods of the UK, those of the Wealden Supergroup, in particular. However, its global significance lies in the fact that it provides substantial new information on the early evolution of tyrannosauroids, and potentially on their ecology and interaction with other theropod and dinosaur lineages. Following recognition of the fact that the tyrannosaurids of the Late Cretaceous are not carnosaurs but coelurosaurs (Holtz, 1994), it became more likely that small “proto–tyrannosaurs” with elongate, tridactyl or tetradactyl forelimbs should await discovery in Jurassic or Lower Cretaceous strata. E. lengi validated this prediction, and recent finds show that it is only one of several non-tyrannosaurid members of the coelurosaurian clade Tyrannosauroidea, some of which are as old as Middle Jurassic. Since E. lengi was named in 2001, Aviatyrannis jurassica from the Kimmeridgian Alcobaça Formation of Portugal (Rauhut, 2003a), Dilong paradoxus from the Lower Cretaceous Yixian Formation of China (Xu et al., 2004), Guanlong wucaii from the Oxfordian Shishugou Formation of China (Xu et al., 2006), Sinotyrannus kazuoensis from the Albian Jiufotang Formation of China (Ji, Ji & Zhang, 2009), Kileskus aristocus from the Bathonian Itat Formation of western Siberia, Russia (Averianov, Krasnolutskii & Ivantsov, 2010), Timurlengia euotica from the Turonian Bissekty Formation of Uzbekistan (Brusatte et al., 2016), Moros intrepidus from the Cenomanian Cedar Mountain Formation of the USA (Zanno et al., 2019) and Suskityrannus hazelae from the Turonian Moreno Hill Formation of the USA (Nesbitt et al., 2019) have been published as additional non-tyrannosaurid tyrannosauroids. It has also become better established that the Late Jurassic Stokesosaurus, originally named for the Morrison Formation species S. clevelandi from the USA, originally suggested to be an early tyrannosaurid (Madsen, 1974), is also an early-diverging tyrannosauroid. The British tyrannosauroid Juratyrant langhami from the Tithonian Kimmeridge Clay Formation, first described as a new species of Stokesosaurus (Benson, 2008), is distinct from S. clevelandi in several respects, notably possessing a narrow, posterodorsally inclined ridge on the lateral surface of its ilium that stops short of the ilium’s dorsal magin. This configuration is present elsewhere (namely in Eotyrannus) but is not present in S. clevelandi or other tyrannosauroids (Brusatte & Benson, 2013). New analyses of Proceratosaurus bradleyi from the Bathonian Taynton Limestone Formation of the UK (Rauhut, Milner & Moore-Fay, 2010) and Dryptosaurus aquilunguis from the Maastrichtian New Egypt Formation of the USA (Brusatte, Benson & Norell, 2011) have established that these taxa are additional members of the tyrannosauroid radiation. Furthermore, both Xiongguanlong baimoensis from the Aptian-Albian Xinminpu Group of western China (Li et al., 2009) and Yutyrannus huali from the Lower Cretaceous Yixian Formation of China (Xu et al., 2012) have been recovered as outside the Dryptosaurus + Tyrannosauridae clade (Brusatte et al., 2010b, 2016; Zanno et al., 2019) while Appalachiosaurus montgomeriensis from the Demopolis Formation of the USA (Carr, Williamson & Schwimmer, 2005) and Bistahieversor sealeyi from the Campanian Kirtland Formation of the USA (Carr & Williamson, 2010) are larger–bodied taxa successively closer to Tyrannosauridae and more like tyrannosaurids in cranial and other characters. It has also been proposed that Bagaraatan ostromi from the Maastrichtian Nemegt Formation of Mongolia (Osmólska, 1996) and Santanaraptor placidus from the ?Albian Santana Formation of Brazil (Kellner, 1999) might be non-tyrannosaurid tyrannosauroids (Holtz, 2004; Choiniere et al., 2010). Mirisichia asymmetrica, also from the Santana Formation, has most often been interpreted as a compsognathid (Naish, Martill & Frey, 2004; Peyer, 2006; Rauhut, Milner & Moore-Fay, 2010) on the basis of its strong similarity with Compsognathus. However, the presence of a similarly proportioned pubis in Dilong (where the pubic foot is proportionally long and lacks an expansion anterior to the shaft; Xu et al., 2004), and the presence of what appears to be a dorsal concavity on the preacetabular process of the ilium (but see Brusatte et al. (2014) for a taphonomic interpretation of that feature) and a concave anterior margin to the pubic peduncle (characters typical of tyrannosauroids; Rauhut, 2003a, 2003b; Xu et al., 2004, 2006; Benson, 2008, Brusatte & Benson, 2013) render it possible that Mirisichia might also be a tyrannosauroid. A few other theropod taxa not typically considered part of Tyrannosauroidea have also been hypothesized to be additional members of the group, namely Tanycolagreus topwilsoni and Coelurus fragilis from the Morrison Formation: both were recovered as early-diverging tyrannosauroids by Senter (2007, 2010), Brusatte et al. (2016) and Zanno et al. (2019). Several additional studies have supported a tyrannosauroid placement of Tanycolagreus (Carr & Williamson, 2010; Brusatte et al., 2014; Choiniere et al., 2014).

A robust phylogenetic framework now exists for Tyrannosauroidea (Li et al., 2009; Loewen et al., 2013; Brusatte et al., 2010a, 2010b, 2016; Brusatte & Benson, 2013; Brusatte & Carr, 2016; Zanno et al., 2019). While conflicting results have led to uncertainty about the topology at the base of the clade, these differences are in part due to incomplete sampling. Holtz (2004) recovered a mostly pectinate arrangement for non-tyrannosaurid tyrannosauroids and found E. lengi to be closer to Tyrannosauridae than were Bagaraatan, Stokesosaurus and Dryptosaurus. Senter (2007, 2010) found E. lengi to be closer to Tyrannosauridae than were Guanlong and Dilong. Li et al. (2009) found E. lengi and Dilong to form a polytomy with a Xiongguanlong + Tyrannosauridae clade. Brusatte et al. (2010b), Brusatte & Carr (2016), Zanno et al. (2019) and Nesbitt et al. (2019) recovered E. lengi as belonging to a clade that also included Stokesosaurus and Juratyrant and was closer to Tyrannosauridae than to Dilong and Proceratosauridae, Rauhut, Milner & Moore-Fay (2010, fig. 24) depicted E. lengi as part of an unresolved polytomy alongside Proceratosauridae, Aviatyrannis, Stokesosaurus and a Dilong + Tyrannosauridae clade, and both Loewen et al. (2013) and Brusatte et al. (2016) found E. lengi closer to a Dryptosaurus + Tyrannosauridae clade than were Proceratosauridae and Dilong (and not in a clade with Stokesosaurus and Juratyrant). Finally, the enigmatic tetanuran clade Megaraptora has recently been placed among non-tyrannosaurid tyrannosauroids by Novas et al. (2013). This controversial hypothesis was further corroborated by the discovery of several tyrannosauroid-like features in a new specimen of Megaraptor (Porfiri et al., 2014); these authors also found Eotyrannus among megaraptorans, though they did note that corroboration was required. The substantial new character information described in the present study allows us to better establish the phylogeny of Tyrannosauroidea.

Context and history of discovery

The E. lengi holotype was discovered in September 1997 by amateur collector Gavin Leng approximately 12 m above beach level near Grange Chine on the south-west coast of the Isle of Wight (Fig. 1). As is the case for most Isle of Wight dinosaur specimens, it was preserved in a plant debris bed of the Barremian (Allen & Wimbledon, 1991) Wessex Formation. The Wessex Formation is a red-bed sequence that comprises varicolored mudstones interbedded with sandstones and subordinate intraformational conglomerates, crevasse splay deposits and plant debris beds (Stewart, 1978, 1981; Insole & Hutt, 1994). It was deposited on a near-shore floodplain crossed by a large west-to-east meandering river (Radley, 1994; Wright et al., 1998). Plant debris beds (sensu Oldham, 1976) represent fusain-rich units formed of siltstone and mudstone; they are mostly less than 1 mm thick so the thickness of the bed that yielded E. lengi may indicate that it was deposited following an especially large or severe flood event. Stewart (1978, 1981) regarded plant debris beds as representing extrabasinal flood events that carried debris onto the Wessex Formation alluvial plain, but Insole & Hutt (1994) argued that they were the result of local storm events and hence that any incorporated material was of local origin. The rarity of E. lengi has led to the speculation that it was not an inhabitant of the floodplain or its immediate surrounds (Naish, Hutt & Martill, 2001). Stewart (1978) assigned bed numbers to each of the plant debris beds within the Wessex Formation and E. lengi was recovered from L11, the plant debris bed above the Grange Chine Sandstone (Fig. 2).

Figure 1 Map of the Isle of Wight to show the geographical and geological context of Eotyrannus lengi.

(A) Geological map of the Isle of Wight, the Wealden Supergroup being most prominent in the island’s south-west but also present in the east. (B) Enlarged area showing key dinosaur-bearing sites on south-west coast. The Eotyrannus lengi holotype was discovered at Grange Chine.

Figure 2 Stratigraphic position of the bed that yielded Eotyrannus lengi.

(A) Schematic relationship of the Wessex Formation to other Wealden Supergroup strata; (B) column showing Wessex Formation exposure between Sudmoor Point and Cowleaze Chine, depicting beds from which E. lengi and some other Wessex Formation dinosaurs were recovered. Modified from Sweetman (2004). (B) Produced with kind cooperation of Chris Barker.

Leng initially recovered only a manual ungual from the site; he took this to S. Hutt (then curator at the Museum of Isle of Wight Geology, Sandown). Hutt realised its significance and (with P. Newbery) visited the site and removed the rest of the skeleton from the outcrop (Hutt et al., 2001; Naish, Hutt & Martill, 2001; Hutt, 2002). The nature of the matrix in which the specimen was preserved made both initial recovery, and preparation in the laboratory, slow and difficult.

Systematic Palaeontology

Theropoda Marsh, 1881

Tetanurae Gauthier, 1986

Coelurosauria Huene, 1914

Tyrannosauroidea Osborn, 1905

Eotyrannus lengi Hutt et al., 2001

Holotype. A partial, disarticulated skeleton (IWCMS: 1997.550) consisting of the anterior portion of the skull, a partial forelimb and pectoral girdle, several cervical, dorsal and caudal vertebrae, rib fragments, part of the ilium, and elements of both hindlimbs. The taxon is known from the holotype alone.

Locality and horizon. The holotype was recovered from Grange Chine on the south-west coast of the Isle of Wight, from the L11 plant debris bed above the Grange Chine Sandstone of the Wessex Formation of the Wealden Supergroup. It dates to the Barremian (see Martill & Naish, 2001).

The osteology of Eotyrannus lengi: general comments

The holotype of Eotyrannus lengi is–after the holotype of Neovenator salerii Hutt et al., 1996 (NHMUK R10001/MIWG 6348) (Hutt, Martill & Barker, 1996, 2001; Naish, Hutt & Martill, 2001; Brusatte, Benson & Hutt, 2008)–the most complete theropod yet reported from the Wessex Formation. However, while the E. lengi holotype includes a substantial number of bones, many of them are broken or even fragmentary. The specimen is embedded within particularly hard sideritic mudstone. Consequently, matrix remains adhered to some of the elements and it should be emphasised that, where matrix obscures part of a given element, the matrix cannot be removed without risk of substantial damage.

The taphonomy of the E. lengi holotype was discussed by Hutt et al. (2001, p. 240) and Martill (2001). Several images exist of the specimen prior to its preparation and provide data on the original orientation and disposition of its various bones. Evidently, the skeleton was substantially disarticulated prior to fossilisation, with elements scattered throughout the area in which it was preserved. None of the vertebrae, for example, are preserved in articulation. Those that are preserved consist of separated neural arches and centra, indicating that the holotype was skeletally immature (Brochu, 1996); it is inferred to represent a subadult pending histological analysis. It is therefore possible that skeletally mature individuals were somewhat larger than the c. 4.5 m we estimate for the holotype (see discussion below). We suggest on the basis of the subadult condition of the holotype, however, that more mature individuals were not much larger.

The only elements that retain close natural association are the left scapula and coracoid and the left tibia, fibula and metatarsal IV. Much of the skull is preserved, though the bones are mostly disarticulated, broken and/or distorted during diagenesis. Some relatively delicate fragments, including a partial surangular and the palatines, are nevertheless well preserved. Hutt et al. (2001, p. 240) suggested that fractured ends present on some of the bones are indicative of pre-burial trampling. However, there are no clear indications of trampling, such as splintered bone or spiral fractures (Hill, 1980; Bilbey, 1999). Bones and teeth from a dryosaurid (assumed to be Valdosaurus sp. and accessioned as IWCMS: 1997.885) are jumbled among the remains of E. lengi. These remains were discussed by Barrett et al. (2011).

Methods

Our description of E. lengi is based on direct examination of the holotype specimen IWCMS: 1997.550, visited on several occasions over the course of this study. Comparisons with other taxa were made via direct examination where possible or through examination of the published literature. Measurements were taken with a variety of rulers and tape measures. The phylogenetic analyses were performed in TNT vers. 1.5 (Goloboff, Farris & Nixon, 2008). Analysis protocol consisted of a first round of 100 heuristic search replications using the following ‘New Technology’ settings: driven search, using sectorial searches and tree fusing. Maxtree was set at 99,999 (maximum storage in TNT). The most parsimonious trees (MPTs) found during the first search round were then submitted to an additional round of tree bisection and reconnection (TBR) branch swapping to more exhaustively explore the recovered tree islands. Nodal Support (Decay Index) for nodes was calculated by saving 50,000 suboptimal topologies up to 10 steps longer than the MPTs in TNT.

The cranial skeleton

The E. lengi holotype preserves more cranial material than any other Wessex Formation theropod, including the holotype of Neovenator salerii. Most of the unambiguously identified cranial bones of E. lengi belong to the part of the skull anterior to the orbit. However, the right surangular and right quadrate are preserved as well. Some of the description provided here necessarily repeats information previously included within Hutt et al. (2001). For measurements of cranial elements, see Table 1.

Table 1 Measurements (in millimetres) of the cranial elements of E.lengi.

Premaxilla	
preserved height	44	
height, body ventral to naris	30	
preserved length, body	36	
mediolateral thickness, posterior end of body	10	
height, most anterior interdental plate	2.5	
length, most anterior interdental plate	2.5	
Maxilla	
preserved length	95	
preserved height	72	
mediolateral thickness	15	
height, 4th interdental plate	24	
length, 4th interdental plate	18	
height, 5th interdental plate	24	
length, 5th interdental plate	20	
length, third alveolus	23	
width, 3rd alveolus	11	
length, 4th alveolus	22	
width, 4th alveolus	13	
Fused nasals	
length	220	
width, mid-length	33	
maximum width	57	
width, posterior end	43	
maximum thickness	20	
preserved length, dorsal border of right naris	15	
depth of right naris at posterior end	15	
Lacrimal	
preserved height	95	
preserved length, dorsal end	47	
preserved length, ventral end	30	
length, mid-shaft	15	
Palatine	
preserved maximum length	88	
maximum width, body	24	
Quadrate	
maximum preserved height	82	
width, across ventral condyles	40	
Left dentary	
preserved length	147	
height	40	
length, 2nd interdental plate	11	
height, 2nd interdental plate	12	
length, 3rd interdental plate	12	
height, 3rd interdental plate	15	
Right dentary	
preserved length	130	
height	46	
length, 1st interdental plate	12	
length, 2nd interdental plate	18	
height, 2nd interdental plate	13	
length, 3rd interdental plate	12	
height, 3rd interdental plate	17	
length, 4th interdental plate	20	
height, 4th interdental plate	17	
height, 5th interdental plate	16	
Surangular	
preserved length anterior half	121	
width, anterior half	1	
preserved length posterior half	115	
Note:

Some measurements are approximate.

Premaxilla

The right premaxilla of E. lengi consists of an almost complete premaxillary body and the base of the nasal process (Fig. 3). The premaxillary ventral margin is mostly complete but its posterior and posterodorsal margins are damaged. Dorsally, the ventral edge of the naris is preserved adjacent to the base of the nasal process.

Figure 3 Incomplete right premaxilla of Eotyrannus lengi IWCMS: 1997.550.

(A) Lateral view; (B) medial view; (C) ventral view; (D) dorsal view; (E) anterior view; alv alveoli, dp dorsal process, intpl interdental plates, mn margin of external naris.

The premaxillary body is longer than it is tall. It is 36 mm long vs 30 mm deep subnarially, resulting in a length/height ratio of 1.2. It appears to be proportionally small relative to the maxilla. As described by Hutt et al. (2001), the premaxillary body is typical for tyrannosauroids in having a high premaxillary angle (the angle between the alveolar margin and anterior border) of 90°. This recalls the condition present in Guanlong (Xu et al., 2006), Proceratosaurus (Rauhut, Milner & Moore-Fay, 2010), tyrannosaurids (Brochu, 2003; Currie, 2003; Hurum & Sabath, 2003) and the premaxilla referred to Stokesosaurus by Madsen (1974) (but see Benson (2008)). Reconstructions that show the premaxilla of E. lengi as having a sloping anterior border (Naish, Hutt & Martill, 2001, text-fig. 9.31; Holtz, 2004, fig. 5.25) are inaccurate since the morphology of the bone shows that its anterior border was perpendicular to the alveolar margin. As noted by Hutt et al. (2001), the premaxillary body expands mediolaterally as it extends ventrally, thus giving it a triangular cross-section. The posterior part of the body adjacent to the maxillary contact is eroded and the maxillary process is absent.

The lateral surface is partly obscured by adhering matrix that covers the region medial to the anteroventral border of the narial fossa. Numerous small foramina are present across the lateral surface of the premaxillary body, the largest of which are situated near the bone’s anterior border. Some of the foramina are located within short, shallow canals that are mostly oriented posteroventrally (Hutt et al., 2001). A shallow, indistinct groove housing numerous foramina extends ventrally from the anteroventral corner of the external naris. This structure is likely homologous with similar indistinct grooves present in Guanlong (Xu et al., 2006) and Proceratosaurus (Rauhut, Milner & Moore-Fay, 2010).

The dorsal process is incomplete, consisting only of its base, and is subtriangular in cross-section. It extends vertically from the premaxillary body and then curves slightly laterally (Figs. 3D, 3E). This might be due to distortion as there are several cracks at its base. In medial view, the dorsal process has a relatively long anteroposterior exposure. In lateral view, the anteroposterior exposure is short because the posterior edge of the process is emarginated by a weakly developed narial fossa, part of which is infilled by matrix.

Four subcircular alveoli are present (Fig. 3C). These are smaller than those on the dentary and maxilla. With both premaxillae imagined in articulation, the premaxillary arcade is broad and U-shaped (Hutt et al., 2001) and the second tooth would have been located almost as far anteriorly as the first. The third tooth would have been located as far anteriorly as the posterior margin of the second tooth, and the fourth tooth would have been located as far anteriorly as the posterior margin of the third tooth. Distinct interdental plates are not present on the medial surface of the premaxilla. They may have been absent but a sheet of bone that extends as far ventrally as the ventral edge of the lateral surface appears to be formed of fused interdental plates (Fig. 3B). Presence and fusion of the plates appears more likely than absence in view of the fact that plates are present in the maxilla and dentaries. However, a poorly developed vertical groove does appear to represent the junction between the second and third plates. Regardless, the medial surface of the premaxillary body dorsal to the fused interdental plates is perforated by several foramina, the anterior-most of which is posterodorsal to the first alveolus and close to or at the junction between what appears to be the first plate and the rest of the medial surface. This is also the largest foramen on the medial surface: it is at the anterior end of a line of perhaps four foramina, the most posterior of which is present close to the posterior border of the premaxilla and dorsal to the fourth alveolus. All of these foramina are in a position equivalent to the junction between the fused interdental plates and the rest of the medial surface. The medial surfaces of the plates are covered with far smaller foramina connected by tiny canals.

Maxilla

Only the preantorbital ramus of the left maxilla is preserved (Fig. 4), although a poorly preserved, fragmentary element tentatively identified as a partial right maxilla is preserved within a block where it is held together by matrix. The fragment of left maxilla preserves intact anterior, anterodorsal and ventral margins but is broken posterior to the anteriormost rim of the antorbital fossa. Only the base of the nasal ramus is preserved, projecting posterodorsally at approximately 45°. The preserved portion is 95 mm long and has a maximum height of 72 mm. Posteriorly, the edge of the nasal ramus is continuous with the anterior rim of the bony margin of the antorbital fossa (Fig. 4E). Medial to the rim is a dorsally convex (and intact) section of maxilla that would have formed part of the wall of the antorbital fenestra ventral to the maxillary foramen.

Figure 4 Incomplete preantorbital ramus of left maxilla of Eotyrannus lengi IWCMS: 1997.550.

(A) Lateral view; (B) medial view; (C) ventral view showing alveoli and ventral surface of maxillary shelf; (D) oblique dorsomedial view to show the five (presumably pneumatic) crater-like concavities; (E) oblique posterolateral view to show anatomy of antorbital fossa margin; (F) medial view to emphasise form of the only distinct interdental plates; (G) anterior view; (H) detail of medial surface to show maxillary shelf; (I) lateral surface with majority of neurovascular foramina and their associated furrows emphasised. alv alveoli, bnr base of nasal ramus, intpl interdental plates, maf margin of antorbital fossa, ms maxillary shelf, nefo neurovascular foramina. Images (C) and (G) were kindly provided by Roger Benson.

The anteroventral rim of the antorbital fossa is sharply delineated and comparable to that of several other coelurosaurs, including Proceratosaurus, Scipionyx and members of Compsognathidae and Tyrannosauridae (Currie & Dong, 2001; Hwang et al., 2004; Xu et al., 2004; Rauhut, Milner & Moore-Fay, 2010; Dal Sasso & Maganuco, 2011). The prominence of this rim varies with ontogeny in tyrannosaurids: Carr & Williamson (2004, p. 517) noted its prominence in juveniles but obliteration during adulthood as the maxilla becomes thicker. Its sharp delineation in E. lengi may therefore be an ontogenetic feature.

The body of the maxilla is mediolaterally thick (Fig. 4G). An anterior ramus, like that present in Guanlong (Xu et al., 2006), Proceratosaurus (Rauhut, Milner & Moore-Fay, 2010) and Sinotyrannus (Ji, Ji & Zhang, 2009), is absent but a prominent change in the angle of the anterior margin is obvious: the anteriormost margin is inclined at an angle of c 70° relative to the alveolar margin while the anterodorsal section of the margin is inclined at a shallower angle of c 30° relative to the alveolar margin. The overall impression is of a short, truncated preantorbital ramus. A furrow on the anteromedial part of the maxilla probably received the maxillary process of the premaxilla like that present in Kileskus, Guanlong and Proceratosaurus (Xu et al., 2006; Averianov, Krasnolutskii & Ivantsov, 2010; Rauhut, Milner & Moore-Fay, 2010) while a slot dorsal to this furrow may have received the premaxillary process of the nasal. The part of the maxilla between these facets is dorsally convex and does not appear to have been overlapped by any bony process. Accordingly, this part of the maxilla probably contributed to the ventral part of the external naris. This contrasts with the more typical tyrannosauroid condition (present even in those with an enlarged external naris) where a long, slender maxillary process on the premaxilla contacts the premaxillary process of the nasal (Brochu, 2003; Xu et al., 2006; Ji, Ji & Zhang, 2009; Rauhut, Milner & Moore-Fay, 2010; Averianov, Krasnolutskii & Ivantsov, 2010). A small notch 26 mm dorsal to the lateral alveolar margin marks the position of the subnarial foramen.

The maxilla’s lateral surface is flat. Foramina of diverse sizes are scattered across this surface: a row of tiny foramina are aligned along the ventral margin, adjacent to the alveolar margin, while larger foramina, some of which are at the dorsal ends of short channels (Hutt et al., 2001), are present across the more dorsal part of the surface (Fig. 4I). A series of deep depressions are arranged in an approximate line dorsal to the alveolar margin. Several small foramina are present within these depressions. This line of structures might be homologous with the alveolar row of foramina present in Guanlong, Proceratosaurus and tyrannosaurids (Currie, 2003; Xu et al., 2006; Rauhut, Milner & Moore-Fay, 2010) and we suggest that future phylogenetic work on tyrannosauroids incorporate this feature as a potential character state description. Several poorly differentiated depressions are present in the anteroventral region of the maxilla, one of which is deeper than the others. This is suggestive of the novel maxillary opening present in Guanlong (Xu et al., 2006) but is less close to the premaxillary contact. This density of apparently pneumatic structures implies that E. lengi’s maxilla was highly pneumatised, at least in its ventral third or so. The alveolar margin of the bone is straight in lateral view.

On the medial surface, the maxillary shelf is dorsal to the alveolar margin (Figs. 4B, 4F–4H). The shelf has a subtle posterodorsal inclination and is only as long as the base of the nasal ramus; its posterior end terminates with an irregular break and its full extent is unknown. The shelf’s anterior part is smooth medially and forms what appears to be a concave facet for articulation with an adjacent element (Fig. 4H), presumably the vomer. A similar facet was illustrated for Tarbosaurus bataar (Hurum & Sabath, 2003). The anterior end of the shelf bears a horizontal groove for articulation with the premaxillary palatal process. The limited medial prominence of the maxillary shelf shows that maxillary contribution to the palate was modest. Dorsomedial to the shelf, five crater-like concavities are present, the largest (c. 20 mm long) and most posterior of which probably represents part of a promaxillary recess (Fig. 4D). It is assumed that these concavities are pneumatic, in which case the dorsomedial part of the preantorbital ramus at least was extensively pneumatised.

Immediately ventral to the palatal shelf, a damaged strip of maxillary wall is marked with a series of poorly defined concavities, at least two of which appear to have a one-to-one correspondence with the more ventrally positioned interdental plates. The homology of these concavities is uncertain but it is possible that they were formed during life by the tips of the dentary teeth: in tyrannosaurid specimens preserved with closed jaws, the dentary teeth are found resting in similar concavities (Currie, 2003). Concavities of this sort are known for tyrannosaurids of all main lineages (Currie, 2003; Brusatte, Carr & Norell, 2012). Five interdental plates are present, though the anterior three are poorly differentiated from the rest of the maxilla and from one another. The two posterior plates are deep relative to the height of the maxilla (Fig. 4F). They are deeper than they are long and subrectangular, though with ventral edges that taper to a point. Interdental plates of this form are typical for tyrannosauroids (Currie, 2003; Brusatte, Benson & Norell, 2011). They are separated by a vertical gap confluent at its dorsal end with a subhorizontal fissure–the groove for the dental lamina–that separates the interdental plates from the rest of the maxilla. Fine, irregularly oriented, anastomosing grooves and small foramina cover their medial surfaces, forming a texture different from the rest of the maxilla. A covering of tiny pits is typical for tyrannosauroids (Currie, 2003; Rauhut, Milner & Moore-Fay, 2010; Brusatte, Carr & Norell, 2012); anastomosing grooves like those present in E. lengi do not seem to be a typical tyrannosauroid feature. Interdental plates are typically not fused in tyrannosauroids (e.g., Currie, 2003; Hurum & Sabath, 2003; Averianov, Krasnolutskii & Ivantsov, 2010; Rauhut, Milner & Moore-Fay, 2010; Brusatte et al., 2010b; Brusatte, Carr & Norell, 2012), though Tanycolagreus appears to be an exception (Carpenter, Miles & Cloward, 2005).

If the large opening present anterolaterally on the maxilla (but posterodorsally on the preserved fragment) is the maxillary fenestra, then E. lengi lacked a promaxillary fenestra. Though primitively present in Theropoda, this structure was lost several times (Rauhut, 2003b). However, it is also possible that the preserved opening is the promaxillary fenestra, and that the maxillary fenestra was located posterodorsal to it and hence not preserved. This latter alternative would imply that the promaxillary fenestra of E. lengi must have been proportionally large compared to that of Guanlong, Dilong, Proceratosaurus, Bistahieversor and tyrannosaurids (Xu et al., 2004, 2006; Carr, Williamson & Schwimmer, 2005; Carr & Williamson, 2010; Rauhut, Milner & Moore-Fay, 2010; Brusatte, Carr & Norell, 2012). The promaxillary fenestra is both comparatively large, and visible in lateral view, in some maniraptorans (Currie & Varricchio, 2004). However, the typical condition for tyrannosauroids is that the promaxillary fenestra is smaller than the maxillary fenestra and tucked up against the rim of the antorbital fossa such that it is partly concealed from lateral view (Xu et al., 2004, 2006; Carr, Williamson & Schwimmer, 2005; Carr & Williamson, 2010; Rauhut, Milner & Moore-Fay, 2010; Brusatte, Carr & Norell, 2012). This strengthens the view that the opening preserved in E. lengi is the maxillary fenestra, and that the promaxillary fenestra was absent. It is also possible that the preserved opening is a combined promaxillary-maxillary fenestra. Monolophosaurus exhibits only a single opening in the anteroventral part of its antorbital fossa (Zhao & Currie, 1994), and while it is in the right place to be a promaxillary fenestra, it appears too large for this, leading Witmer (1997, p. 44) to propose that the two fenestrae had been united by the loss of the promaxillary strut. The presence of this large anterior opening, overlapped ventrolaterally by the prominent rim of the antorbital fossa, is tentatively interpreted as a possible autapomorphy of E. lengi: ultimately, poor preservation limits our ability to be confident about the anatomy of this region.

The maxillary alveoli are subrectangular, longer than wide, and with thin bony walls separating the alveoli. Five alveoli are present, though the fifth is represented only by its anterior-most 5 mm and only the third and fourth can be measured accurately (Fig. 4C).

Nasals

Both conjoined nasals are known for E. lengi (Fig. 5). They are thick and dorsally convex in their anterior two-thirds, the two meeting at their suture at a low angle to create a vaulted anatomy. Posteriorly, they are flattened and with raised posterolateral crests. Both nasals are marked on their dorsal surfaces with large foramina. Both are fused into a single unit with an obliterated suture, although this fusion is incomplete posteriorly: here, the two nasals are distinct and separated by a suture on the dorsal side. A keel representing the suture between the two nasals is visible on the ventral surface (Fig. 5C).

Figure 5 Fused nasals of Eotyrannus lengi IWCMS: 1997.550.

(A) Dorsal view, anterior to left; (B) right lateral view; (C) ventral view, anterior to left. extn external nostril, lacproc lacrimal process; latrec lateral recess; mlri midline ridge, mpp medial premaxillary process, ndpp notch for dorsal process of premaxilla, nefo neurovascular foramina, vpp ventral premaxillary process. Images kindly provided by Alex Peaker.

The left nasal is damaged anteriorly and the narial border is absent, only part of the medial premaxillary process being preserved (Fig. 5A). The right nasal is more complete, preserving part of the border to the external nasal though the anterior tips of both its premaxillary process and ventral premaxillary process are missing (Fig. 5A). This damage to the anterior parts of both nasals mean that it cannot be determined whether nasal fusion had occurred here. Nevertheless, the preserved anterior regions are fully fused. In overall form, the fused nasals are highly similar to those of tyrannosaurids (Hutt et al., 2001; Currie, 2003; Currie, Hurum & Sabath, 2003; Hurum & Sabath, 2003; Holtz, 2004; Snively, Henderson & Phillips, 2006; Brusatte, Carr & Norell, 2012) and, to a lesser degree, those of Guanlong and Dilong (Xu et al., 2004, 2006). The fused nasals of E. lengi are longer, proportionally, than those of Guanlong or Dilong: in these taxa, the fused nasals are approximately four times longer than they are wide at mid-length (Xu et al., 2004, 2006) whereas the fused nasals of Eotyrannus have a far more ‘stretched’ middle section, meaning that they are approximately seven times as long as they are wide at mid-length. The latter condition is much like that of tyrannosaurids (Currie, 2003; Hurum & Sabath, 2003; Brusatte, Carr & Norell, 2012). The fact that the fused nasals are not especially slender relative to those of longirostrine tyrannosauroids like Alioramus (Brusatte, Carr & Norell, 2012)–combined with the shape of the preantorbital ramus of the maxilla–again indicates that E. lengi was not longirostrine.

On the right side, the border of the external naris is well preserved and the right ventral premaxillary process is present (though broken), while on the left both structures are absent (Figs. 5A, 6A). At mid-length the nasals have a maximum width of 33 mm, and are widest 15 mm anterior to the posterior end. As noted above, the fused nasals are dorsally convex for most of their length, but the posterior 60 mm form a flattened region bounded laterally by low ridges. The nasals are similar in width for the anterior two-thirds of their length but widen gradually posteriorly, becoming dorsoventrally flattened as they do so. Five large, asymmetrically arranged dorsal and dorsolateral foramina are present across the middle of the nasals; the three largest and most prominent are on the right nasal where two are close to the midline and one is closer to the lateral edge (Fig. 7). These foramina are deep and subcircular or oval: they have measurements of 6 × 4 mm, 6 × 7 mm, 9 × 4 mm, 8 × 4 mm, and 7 × 5 mm, respectively. A sixth, posteriorly located concavity, positioned on the left nasal and close to the midline, is more elongate anteroposteriorly than these foramina (19 × 3 mm) and may be the result of fusion between two foramina. Some ambiguously shaped concavities cannot be identified as foramina with certainty but probably represent additional examples. Small, widely scattered foramina are common on the nasals of tyrannosauroids (Currie, 2003; Hurum & Sabath, 2003; Xu et al., 2004; Snively, Henderson & Phillips, 2006; Brusatte, Carr & Norell, 2012) but no taxon described thus far has foramina that are as proportionally large as those of E. lengi. Some Tyrannosaurus rex specimens come closest (Snively, Henderson & Phillips, 2006).

Figure 6 Middle section of fused nasals of Eotyrannus lengi IWCMS: 1997.550.

(A) and (B) lateral recess and adjacent area on lateral surface of left nasal; (C) and (D) lateral recess and adjacent area on lateral surface of right nasal; (E) and (F) dorsal view of middle section of fused nasals, anterior to left. bofr bone fragment, intla internal lamina, latrec lateral recess, nefo neurovascular foramina.

Figure 7 Detailed views of anterior and posterior sections of the fused nasals of Eotyrannus lengi IWCMS: 1997.550.

(A) Preserved anterior part of fused nasals in dorsal view, anterior to left; (B) posterior part of fused nasals in dorsal view, anterior to left; (C) posterior part of fused nasals in oblique dorsolateral view, anterior to left; (D) interpretative diagram of same. dolari dorsolateral ridge, extn external nostril, lacproc lacrimal process, mpp medial premaxillary process, ndpp notch for dorsal process of premaxilla, pomepr posteromedial process, vpp ventral premaxillary process.

The medial premaxillary process of the right nasal diverges laterally as it extends anteriorly (Fig. 6A). This indicates that the medial premaxillary processes were spread apart to form a V-shaped notch for reception of the dorsal processes of the premaxillae, as is typical for tyrannosauroids. The preserved anterior border of the right nasal forms the edge of the posterior part of a subovoid naris. However, the ventral premaxillary process may be slightly displaced dorsomedially, meaning that the naris may originally have been deeper. The latter process extends 10 mm anteroventral to the main body of the right nasal and is square in cross-section. The lateral surface of the ventral premaxillary process bears a flat facet for reception of the nasal ramus of the maxilla, 6 mm tall dorsoventrally, that continues posteriorly and extends along the lateral surface of the right nasal body for c. 60 mm (Fig. 5B). The margin of the nasal bearing this facet is missing from the left side.

Posterior to this facet, the lateral surface of the right nasal possesses a deep subtriangular embayment 53 mm long (Figs. 5B, 7C, 7D), here termed the ‘lateral recess’. It does not resemble the concave lateral structures seen on the nasals of allosauroids since those are clearly confluent with the antorbital fossa and are not separated from it by a prominent rim (Rauhut, 2003b), nor can it be for reception of the lacrimal, as suggested by Hutt et al. (2001, p. 230), since it is positioned too far anteriorly. It appears that the recess in E. lengi is dorsal to the antorbital fossa and was not continuous with it. The recess is deepest posteriorly, increasing in height from 3 mm anteriorly to 9 mm posteriorly. Its ventral floor is flat and smooth; the ventral side of the recess, however, is defined by a sharp, low, lateral ridge that extends the full length of the recess and meets the dorsal margin at an acute angle. The dorsal margin of the recess has a convex lateral edge that is continuous with the dorsal surface of the nasal and marks the junction between the lateral and dorsal surfaces of the nasal. Internal vertical bony struts indicate some form of partitioning of this recess, although damage and matrix infill preclude a full investigation of their morphology. A vertical lamina extends from the floor to the roof of the recess c. 30 mm from the recess’s anterior end; what appears to be another lamina is located closer to the posterior end. The lateral recess on the left side is similar but with less well-preserved margins and extends further posteriorly than the recess on the right side, being 70 mm long. At least one subvertical, although posterodorsally inclined, lamina is present 36 mm posterior to the recess’s anterior end (Figs. 7A, 7B).

Pneumatic recesses of various kinds have been reported in other theropods. The abelisaurid Majungasaurus atopus possesses a subcircular recess, continuous with internal hollows, half-way along each nasal (Sampson et al., 1998; Tykoski & Rowe, 2004). Nasal recesses are also present in Monolophosaurus and members of Allosauroidea (Madsen, 1976; Zhao & Currie, 1994) where they occur within the antorbital fossa (Currie & Zhao, 1994, fig. 1). These structures are different in shape to the recesses of E. lengi and (combined with the disparate phylogenetic positions of these taxa) are assumed to be non-homologous. Within Tyrannosauroidea, Guanlong and Dilong both possess nasal recesses. Xu et al. (2004, fig. 1A-B) figured two elongate recesses in Dilong located dorsal to the anterior half of the antorbital fenestra. They interpreted these as belonging to the laterodorsal part of the maxilla but they more likely belong to the nasals as they do in E. lengi. In Dilong, the recess is very similar to that of E. lengi: it is subtriangular, being deepest posteriorly; a prominent lateral ridge forms its floor and separates it from the antorbital fossa; and a lamina divides it at mid-length into anterior and posterior portions (Xu et al., 2004). Guanlong also possesses elongate openings on the lateral surfaces of its nasals (Xu et al., 2006), dorsal to the anterior part of the antorbital fenestra. However, these are located on the sides of the large nasal crest of this taxon, and–if assumed to be homologous to the recesses of other tyrannosauroids–evidently migrated dorsally as the nasals themselves evolved into a tall, laterally compressed crest. Pneumatisation of the nasals is also known for Proceratosaurus (Rauhut, Milner & Moore-Fay, 2010), although it is unknown whether this taxon possessed lateral recesses. It may therefore be that pneumatic nasals are ubiquitous among early tyrannosauroids but were lost in the Xiongguanlong + Tyrannosauridae clade (Li et al., 2009).

Posterior to the lateral recess, the lateral edge of each nasal is convex and smooth (Fig. 5B). This contrasts with the tyrannosaurid condition where transverse ridges and grooves are present (Hurum & Sabath, 2003, p. 169). There are no distinct lateral facets for reception of the dorsal end of the lacrimal or the prefrontal. Dorsolaterally, the edges of both nasals form low, blunt ridges that (as measured on the more complete left side) are 60 mm long. In dorsal view, the ridges diverge posterolaterally away from the skull’s midline (Figs. 6B–6D). The ridges do not describe perfectly straight lines, but are slightly curved, being convex laterally. At their anterior ends, both ridges grade into the convex dorsal surfaces of the more anterior parts of the nasals, but for most of their length they are taller than the adjacent flattened medial portions of the nasals. The result is a Y-shaped arrangement of raised surfaces on the fused nasals. The same configuration is present in Dilong (Xu et al., 2004), the primary difference being that Dilong’s nasals are much shorter. Posterolaterally, the ridges of E. lengi extend posteriorly as prong-like structures separate from the rest of the nasals (Figs. 5A, 5C, 6B–6D), though this is only preserved on the left side. These structures are superficially similar to the lacrimal processes identified in some tyrannosaurids (Hurum & Sabath, 2003) as well as in Carnotaurus (Bonaparte, Novas & Coria, 1990, fig. 2), Ceratosaurus (Madsen & Welles, 2000, plate 3) and some allosauroids (Currie & Zhao, 1994, fig. 5) where they articulate with the dorsal process of the lacrimal. However, because the prong-like structures in E. lengi are continuous with the posterolaterally located nasal ridges and located far posteriorly on the nasals, they are likely not homologous with the lacrimal processes discussed by Hurum & Sabath (2003). In fact, based on comparison with Dilong (Xu et al., 2004), the structures in E. lengi must have been located posterior to the descending ramus of the lacrimal. It remains unknown whether these prong-like structures had any direct relationship with the lacrimals and must instead have articulated with the frontals.

Posteriorly, and between the nasal ridges, a concave area is continuous with paired, posteromedial processes that would have met the frontals (Figs. 5A, 6B–6D). Together, these give the posterior end of the fused nasals a breadth of 43 mm. The open suture separating the posterior ends of the nasals extends anteriorly for 40 mm, or half the length of the nasal ridges. The paired posteromedial processes are large: they have subparallel medial and lateral margins but rounded (albeit incompletely preserved) posterior edges. In life, both would have overlapped the frontals. The amount of overlap appears to have been extensive, the nasals forming an ‘m’-shaped region dorsal to the anterior edges of the frontals. This amount of overlap is confirmed by the scarified ventral surfaces of the posteromedial processes. A mid-line lappet of bone emerging from the nasals–as is seen in some tyrannosaurids (Currie, 2003)–is not present. The ventral surface of the fused nasals reveals little detail. It is flat, the internasal suture forming a low keel that extends for most of the nasals’ length (Fig. 5C). Foramina occur irregularly along this surface. This contrasts with the condition reported for tyrannosaurids (Hurum & Sabath, 2003, p. 169) where the ventral surface is smooth and transversely concave.

The nasals have been CT-scanned and a separate study discussing their internal morphology is in preparation.

Lacrimal

The right lacrimal of E. lengi consists of a descending ramus and an incomplete anterior process (the lateral surface of which is mostly obscured by irremoveable matrix) that would have been parallel to the side of the nasal (Fig. 8). As preserved, the bone has a height of 95 mm. In dorsal view the lacrimal is subrectangular and flat (Fig. 8A), with no trace of a dorsally inflated region, ridge or cornual process like those present in Appalachiosaurus and tyrannosaurids (Carr, Williamson & Schwimmer, 2005). Guanlong and Dilong also possess the same type of lacrimal as E. lengi (Xu et al., 2004, 2006).

Figure 8 Right lacrimal (and possible prefrontal) and partial jugal of Eotyrannus lengi IWCMS: 1997.550.

(A) Dorsal view; (B) lacrimal and partial jugal in lateral view; (C) lacrimal in medial view; (D) oblique posteromedial view; (E) lacrimal shaft in anterior view. df dorsoventral furrow, latf lateral flange, medr medial ridges, pospr possible prefrontal, vgm ventral groove for posteroventral part of maxilla.

The anterior and descending rami of E. lengi meet at an angle of c. 90°, giving the lacrimal the form of an inverted ‘L’: this more closely recalls the condition present in Guanlong, Dilong and the majority of tyrannosauroids and theropods (Xu et al., 2004, 2006) than the ‘7-shaped’ lacrimal present in several tyrannosaurids (Brusatte, Carr & Norell, 2012). However, the ventral edge of the preserved fragment of jugal (which articulates tightly with the ventral end of the lacrimal’s descending ramus) indicates that the descending ramus of the lacrimal was somewhat posterodorsally inclined in life. This matter is discussed further below.

The dorsolateral part of the lacrimal is obscured by matrix: it is assumed that a pneumatic foramen was present here since this is the plesiomorphic state for Tetanurae (Sereno et al., 1994, 1996; Witmer, 1997; Rauhut, 2003b), being absent only in ornithomimosaurs and most maniraptorans. The medial surface of the dorsal end is slightly concave but it is not possible to articulate the lacrimal with the lateral surface of the right nasal.

The incomplete anterior process of the lacrimal is mediolaterally narrow, being 6 mm wide at most. What appears to be a concave furrow at its anterodorsal extremity may have received an articular process from the nasal. The ventral edge of the anterior process joins the anterior edge of the descending ramus via a continuous curved border, this defining the posterodorsal edge of the antorbital fossa.

The descending ramus is straight (Figs. 8B–8E) as it is in Guanlong and Dilong (Xu et al., 2004, 2006), not bowed anteriorly as it is in Appalachiosaurus and tyrannosaurids (Russell, 1970; Carr, 1999; Brochu, 2003; Currie, 2003; Hurum & Sabath, 2003; Carr, Williamson & Schwimmer, 2005). The descending ramus is formed of distinct lateral and medial laminae. In lateral view, the lateral lamina obscures the medial lamina except ventrally, close to the bone’s contact with the jugal. Here, the medial lamina is exposed and the anterior edge of the lateral lamina is directed posteroventrally. In this respect, the lacrimal of E. lengi is like that of Dilong, Proceratosaurus and tyrannosaurids (Hurum & Sabath, 2003; Xu et al., 2004; Rauhut, Milner & Moore-Fay, 2010; Brusatte, Carr & Norell, 2012) more than that of Guanlong where the medial lamina is more extensively exposed laterally (Xu et al., 2006). The descending ramus of Guanlong also appears more robust than it is in other tyrannosauroids (Xu et al., 2006). The anterior face of the descending ramus of E. lengi is deeply concave, with a dorsoventral furrow extending along its length (Fig. 8E), the lateral and medial boundaries of which are formed from the anterior edges of the lateral and medial laminae. Several foramina and recesses are located within this furrow. An especially large, ovoid concavity, the edges of which are obscured by irremoveable matrix and broken bone, is present at the dorsal end of the furrow. We were not able to determine whether it is a blind recess or penetrates deeply into the bone but it appears homologous with the pneumatic foramen, presumably associated with the lacrimal canal, present in the same position in tyrannosaurids (Currie, 2003; Brusatte, Carr & Norell, 2012). Ventral to this large opening, a series of smaller foramina are present, at least two of the more ventrally positioned of which are associated with dorsoventrally aligned grooves. These structures indicate that the descending ramus was extensively pneumatised: foramina positioned within this groove have been described in tyrannosaurids (Currie, 2003; Brusatte, Carr & Norell, 2012) but they do not extend as far ventrally as they do in E. lengi. Currie (2003, fig. 19) referred to these foramina as lacrimal ducts but this may be incorrect given that they appear to be pneumatic.

The medial surface of the descending ramus bears two ridges that extend from the posterodorsal region of the ramus to its anteroventral third. They may be the anterior and posterior margins of a single elongate facet that extends for much of the length of the descending ramus. Ridges on the medial surface of the descending process of the lacrimal are a typical feature of tyrannosauroids and have been reported in Appalachiosaurus (Carr, Williamson & Schwimmer, 2005) and several tyrannosaurids (Currie, 2003; Brusatte, Carr & Norell, 2012). Carr, Williamson & Schwimmer (2005) termed the medial ridge in Appalachiosaurus the orbitonasal ridge and noted that it functioned in separating the “orbit and paranasal cavity” (p. 124). An alternative and complementary possibility is that it provided mechanical strength (Currie, 2003, p. 201). These ridges differ in position and form among taxa. In E. lengi, the ridges are closer to the posterior edge of the ramus than the anterior one. In Appalachiosaurus and Alioramus, the ridge is close to the anterior edge of the ramus (Carr, Williamson & Schwimmer, 2005; Brusatte, Carr & Norell, 2012) while in Albertosaurus it is close to the posterior edge. The thickness of the ridge is known to be variable with ontogeny (Brusatte, Carr & Norell, 2012), so it is conceivable that its orientation and position may have varied as the animal matured. In E. lengi there are at least two foramina on the medial surface of the descending ramus, posterolateral to these ridges.

Ventrally, the descending ramus flares anteroposteriorly so that the ventralmost part would have been c. 30 mm long, and thus wider than the shaft is at mid-height. The ventralmost end curves medially. The ventral termination of the bone is damaged; however, some of the bone shards are preserved adhering to the dorsal edge of the partial jugal, meaning that both can be articulated with a good degree of fit.

Possible prefrontal

What might be a damaged prefrontal is preserved in association with the dorsomedial part of the lacrimal, immediately dorsal to the ascending ramus, though it is difficult to determine if cracking of the periosteum simply creates the impression of a separate ossification (Fig. 8A). It appears to be a block-shaped bone, separated from the lacrimal by a curving, dorsally convex line that could represent a suture. In tyrannosaurids, the prefrontal is a crescentic element that separates the lacrimal from the posterolateral part of the nasal and anterolateral part of the frontal, distinct prefrontal facets on the dorsomedial lacrimal being anterior to a contact zone with the frontal (Currie, 2003; Brusatte, Carr & Norell, 2012). The fragmented structure present in E. lengi is in the right position to represent the prefrontal; furthermore, the presence of an articulated prefrontal is consistent with the fact that a prefrontal facet is not visible on the lacrimal.

Jugal

Two incomplete sections of the body of the right jugal (66 mm long) are preserved as lateromedially flattened plates with slightly concave lateral surfaces. The larger fragment is 66 mm long and 36 mm tall while the smaller one is 36 mm long and 29 mm tall. The fragments do not articulate well and additional portions of the bone are clearly missing. They provide little information but the ventral edge of the larger fragment bears a 23 mm long facet, shaped like an inverted ‘V’ and separated from the lateral surface by a convex longitudinal ridge. A similar ridge is present on the lateral surface of the smaller fragment which also possesses part of a V-shaped facet along its ventral border. It is assumed that both of these facets were originally continuous, and presumably for articulation with the maxilla. It is also assumed on the basis of comparison with articulated tyrannosauroid skulls that this facet was aligned subparallel to the skull’s long axis. The larger section fits well against the broken ventral end of the lacrimal (Fig. 8B). Accordingly, the articulated jugal and lacrimal must originally have been oriented such that the descending ramus of the lacrimal was posterodorsally inclined relative to the alveolar margin. A cross-sectional view of the smaller fragment reveals that its medial and lateral walls form the sides of a 6 mm wide internal cavity.

Palatine and possible vomers

An incomplete left palatine, 88 mm long, is preserved on a block of matrix (Fig. 9). A similar but less complete element represents the posterior part of the same bone from the right side. The more complete palatine consists of a flattened, subrectangular body 19–24 mm wide, the anterior end of which supports two short processes (the vomeropterygoid and maxillary processes) while the posterior end gives rise to a large, posterodorsally projecting structure that is incomplete and damaged (the pterygoid process). E. lengi’s palatine is elongate and shallow relative to the palatines of Appalachiosaurus and tyrannosaurids (e.g., Carr, Williamson & Schwimmer, 2005, fig. 11; Brusatte, Carr & Norell, 2012, fig. 25). Few data on non-tyrannosaurid tyrannosauroid palatines are available but the palatine of E. lengi is similar to that of Guanlong (Xu et al., 2006), albeit longer and with a longer, straighter dorsal margin.

Figure 9 Palatines and possible vomers of Eotyrannus lengi IWCMS: 1997.550.

(A) Left palatine as preserved on block of matrix in lateral view, anterior to the left. (B) Left palatine digitally removed from matrix. (C) Incomplete fragment of anterior end of right palatine and possible posterior ends of vomers, as preserved on block of matrix in lateral view. bofr bone fragment, cho border of choana, juar jugal articulation, juar jugal articulation, maxa maxillary articulation, ppr palatine pneumatic recess, ptpr pterygoid process, sr sinuous ridge, vo vomer, vproc vomeropterygoid process.

The middle of the palatine body is damaged but the remnants of several openings are present. These are presumably palatine recesses homologous to those present in tyrannosaurids, allosauroids and other tetanurans (Witmer, 1997). They are consistent with the exposed surface being the lateral one. The anterior end of the palatine is concave: this concave edge (representing the posterior border of the choana) is ventrally continuous with the maxillary process, the anterior part of which forms a pointed projection. This projection comes to a natural termination that does not extend anteriorly any further than the base of the vomeropterygoid process. This condition contrasts with that in Appalachiosaurus (Carr, Williamson & Schwimmer, 2005) and tyrannosaurids where the maxillary process extends anterior to the vomeropterygoid process (Currie, 2003; Hurum & Sabath, 2003; Brusatte, Carr & Norell, 2012). The condition in non-tyrannosaurid tyrannosauroids is not clear due to poor preservation and a lack of good disarticulated cranial material (Xu et al., 2004; Li et al., 2009; Rauhut, Milner & Moore-Fay, 2010) but the palatine of Guanlong appears similarly proportioned to that of E. lengi (Xu et al., 2006). It is typical in theropods for the vomeropterygoid process of the palatine to be longer than the maxillary process (Dal Sasso & Maganuco, 2011).

Much of the ventral edge of the maxillary process is broken but it appears to be continuous with the ventral edge of the palatine body, as is typical of theropods. The broad base of the vomeropterygoid process projects anterodorsally: the process projects at an angle of c. 40° relative to the palatine’s long axis and is incomplete, terminating with a jagged break. The vomeropterygoid process of E. lengi is unusual in that a sinuous ridge, approximately perpendicular to the skull’s long axis, extends across the base. The part of the process dorsal to this ridge is inset or embayed relative to the ventral part: the latter part is continous with the palatine body. A ridge of this sort has not been described in any other tyrannosauroid, to our knowledge, and it may be an autapomorphy.

The dorsal margin of the palatine is subparallel to the ventral margin and forms a relatively long, dorsally concave edge between the vomeropterygoid process and posterodorsally projecting pterygoid process. The length of this edge is unusual relative to other tyrannosauroids, all of which possess a shorter edge in the same region (Currie, 2003; Carr, Williamson & Schwimmer, 2005; Xu et al., 2006; Brusatte, Carr & Norell, 2012). At its base, the pterygoid process is 11 mm wide but it expands to 27 mm posteriorly. It lies in the same plane as the palatine body; it cannot be determined if this is natural or the result of compaction. The ventral margin of this process describes a wide, shallow arc. The broken surface of the fragile pterygoid process reveals little anatomical detail and its dorsal end is damaged and incomplete. A small, triangular, posteroventral prominence presumably represents the area of articulation with the jugal.

The incomplete fragment of right palatine is here interpreted as the anterior part of the bone preserved in lateral view (Fig. 9C). An anterodorsally projecting bar represents the incomplete anterior section of the base of the vomeropterygoid process, the anterior margin of which is continuous with the concave edge that would have formed the posterior border of the choana. At the ventral end of this concavity, an anteriorly projecting, triangular prominence represents the maxillary process: it is better preserved and more complete than the one preserved on the left palatine and confirms that the process in E. lengi is far shorter anteroposteriorly than the vomeropterygoid process. The ventral edge of the process is continuous with the ventral edge of the palatine body, as is also the case on the left palatine. A few slender, horizontally aligned bone fragments, marked with longitudinal striations, are preserved adjacent to the anterodorsal end of the vomeropterygoid process. They are perhaps fragments of the vomers and superficially resemble those described for other tetanurans (e.g., Madsen, 1976; Molnar, 1991); if so, they demonstrate the presence of paired, parallel, slender, rod-like components of these elements.

Quadrate

The single preserved quadrate of E. lengi was briefly described by Hutt et al. (2001, pp. 231–232) where it was identified as a left quadrate; it is reidentified here as a right quadrate. It is mostly complete although the head and adjacent part of the shaft are missing (Fig. 10). The gracile shaft has subparallel medial and lateral margins in posterior view. The lateral side of the shaft immediately dorsal to the lateral condyle is expanded mediolaterally forming a prominent lateral flange that articulated with the quadratojugal (which is not preserved). The dorsal margin of this flange forms a shoulder where it abruptly grades into the dorsal half of the lateral margin of the quadrate shaft. A shallow, dorsoventrally elongate posterior fossa (Hendrickx, Araújo & Mateus, 2015) is present near the middle of the shaft’s posterior surface. The quadratojugal contact area is limited to the ventral part of the quadrate: the quadrate foramen is positioned in between the quadrate shaft and the (unknown) quadratojugal, as is the case in other tyrannosauroids (Carr, 1999; Carpenter, Miles & Cloward, 2005; Li et al., 2009; Rauhut, Milner & Moore-Fay, 2010; Brusatte, Carr & Norell, 2012). The medial embayment of the quadrate’s shaft dorsal to the quadratojugal contact area further shows that the quadrate foramen was large and dorsoventrally elongate and thus similar to the quadrate foramen of Xiongguanlong (Li et al., 2009) and tyrannosaurids (Carr, 1999; Brusatte, Carr & Norell, 2012). The foramen of Proceratosaurus (Rauhut, Milner & Moore-Fay, 2010) is much smaller.

Figure 10 Incomplete right quadrate of Eotyrannus lengi IWCMS: 1997.550.

(A) Posterior view; (B) anterior view; (C) medial view; (D) ventral view. latc lateral condyle, medc medial condyle, mfos medial fossa, ptpr pterygoid process, qfor quadrate foramen, qfos quadrate fossa, qjfl quadratojugal flange, qr quadrate ridge.

A flattened, laterally directed area on the lateral side of the quadratojugal flange, measuring c. 18 mm deep dorsoventrally and 9 mm anteroposteriorly, represents the facet for the quadratojugal.

The medial edge of the posterior surface of the shaft possesses a pillar-like dorsoventrally aligned quadrate ridge that, at the mid-height of the shaft, forms the medial border to a concave region on the shaft’s posterior surface (Fig. 10C). The quadrate ridge is also obvious as a pillar-like thickening when the quadrate is viewed medially: in this view it forms the posterior border to the prominent medial fossa (Hendrickx, Araújo & Mateus, 2015). Quadrate ridges are present in theropods of many lineages (Hendrickx, Araújo & Mateus, 2015): within Tyrannosauroidea they are present in both Proceratosaurus (Rauhut, Milner & Moore-Fay, 2010) and Tyrannosauridae (Brusatte, Carr & Norell, 2012).

The anterodorsally projecting pterygoid process has its ventral margin well dorsal to the condyles (Fig. 10C). This is also the case in some allosauroids (Madsen, 1976), Zuolong (Choiniere et al., 2010), Tanycolagreus (Carpenter, Miles & Cloward, 2005), Proceratosaurus (Rauhut, Milner & Moore-Fay, 2010) and tyrannosaurids (Molnar, 1991; Currie, 2003; Brusatte, Carr & Norell, 2012). A large, deep opening–the medial fossa (Hendrickx, Araújo & Mateus, 2015)–is present on the medial surface of the process, close to its junction with the medial edge of the shaft (Fig. 10C).

The ventral condyles are bulbous and similar in size; they are short anteroposteriorly. The long axis of the medial condyle is near-perpendicular to the mediolateral axis of the quadrate’s shaft whereas the long axis of the lateral condyle is oriented at about 45° relative to the mediolateral axis of the quadrate’s shaft (Fig. 10D). The medial condyle is bulbous and convex ventrally such that it extends further ventrally than the lateral condyle; a similar degree of ventral convexity to the medial condyle is seen in some allosauroids (Madsen, 1976), Tanycolagreus (Carpenter, Miles & Cloward, 2005) and some tyrannosaurids (Brusatte, Carr & Norell, 2012). A proportionally wide channel–similar in width to the medial condyle at 4 mm–separates the condyles. Some tyrannosauroids (Dilong and Tyrannosauridae) possess a pneumatic foramen or recess dorsal to the condyles on the anterior surface of the quadrate shaft (Brusatte, Carr & Norell, 2012; Hendrickx, Araújo & Mateus, 2015). No such structure is present in E. lengi.

The quadrate morphology of E. lengi is typical for a tyrannosauroid and similar to that of Tanycolagreus and tyrannosaurids. The enlarged quadrate fenestra indicates that E. lengi is closer to tyrannosaurids than Proceratosaurus and similar taxa. If the depression on the medial surface of the pterygoid process is indicative of quadratic pneumaticity, E. lengi is more like tyrannosaurids than like Tanycolagreus, Guanlong or Proceratosaurus, since quadrate pneumaticity is absent in those taxa (Carpenter, Miles & Cloward, 2005; Rauhut, Milner & Moore-Fay, 2010; Brusatte, Carr & Norell, 2012).

Dentary

Both dentaries are known for E. lengi. The left dentary is incomplete (Hutt et al., 2001, fig. 3D), terminating posterior to the 9th alveolus with a jagged break (Figs. 11A–11F). The right dentary is less well preserved and is distorted, being strongly bent anterolaterally (Figs. 11G–11L). It is preserved in two pieces, with the 37 mm long anterodorsal tip being separate from the rest of the bone. This tip is duller in colour than the rest of the bone and presumably experienced weathering prior to collection. Its dorsoventral height is only measureable at its anterior end where it is 46 mm.

Figure 11 Incomplete left and right dentaries of Eotyrannus lengi IWCMS: 1997.550.

(A) Left dentary in lateral view; (B) left dentary in lateral view with lateral furrows emphasised; (C) left dentary in oblique dorsomedial view; (D) left dentary in medial view; (E) left dentary in dorsal view; (F) oblique dorsolateral view of anterior end of left dentary; (G) anterior end of right dentary in lateral view; (H) anterior end of right dentary in medial view, rotated such that laterally deflected tip is better visible in medial view; (I) right dentary in medial view (laterally deflected tip thus directed away from viewer and partly obscured); (J) right dentary in medial view with most obvious interdental plates emphasised; (K) right dentary in dorsal view; (L) right dentary in ventral view. intpl interdental plates, latf lateral furrows, mg Meckelian groove, nefo neurovascular foramina, rosno rostral notch, sg secondary groove.

The broken posterior ends of both dentaries reveal at least two internal cavities, both taller than wide. The ventral cavity is smaller (9 × 6 mm) than the dorsal one. It is not possible to determine how far dorsally the more dorsal cavity extends. The bone wall forming the dentary’s ventral margin is thicker (5 mm) than the medial and lateral walls (both c. 3 mm). At its posterior end, the dentary is 14 mm wide.

Seven alveoli are preserved on the left dentary, the three anterior-most alveoli appearing sub-circular in outline while the more posterior ones are sub-rectangular. There is space at the posterior end for an eighth and possibly a ninth, but their margins are obscured. As discussed below, interdental plates are present in E. lengi (contra Hutt et al., 2001) and are inset relative to the rest of the medial surface (Fig. 11D). A narrow shelf c. 1 mm wide, located 26–30 mm dorsal to the ventral edge of the dentary, demarcates the flat medial surface from the interdental plates.

An unusual notch is present on the lateral and medial sides of the first alveolus on the dorsal surface of the dentary (Figs. 11A, 11B, 11F, 11G). This was not described by Hutt et al. (2001) but a dotted line in fig. 3D indicates that the notch was regarded as a result of damage to the dentary’s tip. However, though some of the ‘notched’ bone surrounding the first alveolus is obscured or damaged, some of it is complete, well preserved and intact, and an identical notch is present on the right dentary. The notch thus appears to have been a natural feature. The junction between the anterior margin and ventral edge of the dentary forms a smooth convex arc and differs from the condition in tyrannosaurids where a distinct angle is present between the anterior and ventral surfaces (Currie, 2003; Holtz, 2004). A distinct angle is also present in Bagaraatan (Osmólska, 1996).

The left dentary’s lateral side is marked by several large foramina (Fig. 11A). The largest foramen (c. 7 × 3 mm) is anteriorly located, and just posteroventral to the notched edge of the first alveolus. Two smaller foramina (each c. 1 × 1 mm) are located approximately ventral to this large one and a line of at least six are spaced along the dentary posterodorsal to the largest one. These latter foramina are shallower than the large foramen and arranged in a line that extends subparallel to the dentary’s lateral margin. All are c. 8 mm ventral to the lateral alveolar margin and appear to represent the more dorsally located section of the alveolar row of dentary foramina: in tyrannosauroids generally, the more posterior foramina are located farther ventrally on the dentary’s surface (Brochu, 2003; Currie, 2003; Xu et al., 2004, 2006; Brusatte et al., 2010a, 2010b; Rauhut, Milner & Moore-Fay, 2010). On the right dentary, a row of foramina subparallel to the alveolar margin also appears to be present, though only two of the foramina are clearly preserved. In Guanlong, Proceratosaurus and Sinotyrannus, some of the dentary foramina are located within a groove that parallels the dentary’s dorsal margin (Xu et al., 2006; Ji, Ji & Zhang, 2009; Rauhut, Milner & Moore-Fay, 2010), but no such structure is present in E. lengi. The pattern of foramina at the anterior tip of the right dentary is similar and to that on the left dentary, but better preserved, with two large foramina (5 × 3 mm and 3 × 2 mm respectively) present posterior to the largest one (6 × 4 mm). These additional foramina are only preserved as ambiguous concavities on the left dentary. The right dentary also preserves a prominent anteroventral foramen (4 × 2 mm) that is preserved in the same position as that occupied by a pair of foramina on the left dentary.

On the lateral side of the left dentary, extending across the surface ventral to alveoli 2–5, are five anterodorsally curving, shallow furrows (Fig. 11B) that terminate posteriorly at a single small concavity (c. 8 × 4 mm), located ventral to the junction between alveoli 5 and 6. This concavity may house a foramen. The furrows consist of a ventral horizontal portion and a raised, anterodorsally curving portion. The raised portion is inclined at a shallower angle (of c. 30°) in the most anterior furrow relative to the higher angle (of c. 70°) of the most posterior one. The furrows positioned between these two are inclined at intermediate angles. The furrows are far less obvious on the right side, though fracturing of the bone’s surface and strong bending to the right have obscured its original detailed structure. Curved furrows of this sort have not been reported in any other theropod to our knowledge and they are hence regarded as an autapomorphy of E. lengi.

The dentary’s medial surface is largely flat, though slightly convex in its ventral third. Anteriorly it lacks a distinct symphyseal area and there is no suggestion of a medial inflection. A low ridge and parallel, shallow groove are present on the anteromedial edge of the dentary. Both are presumably symphyseal features for articulation with the opposite dentary. The ridge continues dorsally to form a bony projection anteromedial to the first alveolus. The Meckelian groove is straight and shallow, merges smoothly into the medial surface of the bone, and is located some distance dorsal to the dentary’s ventral edge, lying about half-way up the medial surface (Fig. 11I). It does not extend to the dentary’s anterior end. A very similar condition is present in Dryptosaurus, and indeed a ‘centred’ position of the Meckelian groove on the medial surface of the dentary appears to be typical for tyrannosauroids (Brusatte, Benson & Norell, 2011). A shallow medial groove on the ventral 26–30 mm of the dentary–arbitrarily labelled ‘secondary groove’ in Fig. 11–is deepest (c. 7 mm) at its posterior-most end and becomes shallow anteriorly, eventually merging imperceptibly with the rest of the dentary’s medial surface (Fig. 11D). It is more obvious on the left dentary than the right due to the cracked and distorted nature of the latter. It is possible that this is an additional autapomorphy of E. lengi.

Hutt et al. (2001, p. 232) were unsure as to the presence of interdental plates in E. lengi but several of the statements made about interdental plate morphology are incorrect. Hutt et al. (2001) wrote that “the interdentary [sic] plates … cannot be reliably distinguished from the bone on the dentary’s labial [sic] surface” (p. 232). In the latter statement, the word ‘labial’ should read ‘lingual’. It was further stated “In Eotyrannus the plates may, therefore, be fully fused or, as is the case with Deinonychus, reference to these structures as interdental plates may be a question of semantics” (p. 232). Interdental plates can, in fact, be distinguished from the rest of the medial surface, and the interdental plates themselves are not fused at all. They appear similar in form and proportions to those of allosauroids, tyrannosaurids and other groups (e.g., Madsen, 1976; Currie, 2003). Four interdental plates–the most anterior ones–can be distinguished on the left dentary (Fig. 11D). Another four are probably present but cannot be identified unambiguously. The most anterior interdental plate is incomplete, with only 6 mm of its length being visible. It is not adjacent to the first alveolus but rather to the anterior half of the second. Whether an interdental plate was associated with the first alveolus is unknown. Neither dentary preserves evidence of a plate in this location but this may be due to loss or damage.

Five interdental plates are visible on the right dentary (Figs. 11I, 11J). As on the left dentary the first plate is smallest in terms of both length and height (breakage creates the impression that two interdental plates are present here). The more posterior interdental plates on both dentaries are all similar in morphology, consisting of an approximately square-shaped body capped by a triangular apex. The tip of the triangle forms the dorsal projection of the alveolar septum. The medial surfaces of the plates have a distinctive wrinkled surface texture distinct from that of the rest of the dentary and similar to the texture present on the maxillary interdental plates.

Surangular

The incomplete right surangular, 121 mm long, went unmentioned by Hutt et al. (2001). It consists of a shallow, subrectangular, laterally compressed body that, along its dorsal edge, has overhanging shelves on both its medial and lateral sides. The cotylar region and retroarticular process are intact (Figs. 12A–12C). In overall form it is similar to the surangulars of Guanlong, Dilong, Proceratosaurus and Alioramus (Xu et al., 2006; Rauhut, Milner & Moore-Fay, 2010; Brusatte, Carr & Norell, 2012) and less deep than the surangulars of Bistahieversor and non-alioramine tyrannosaurids (Molnar, 1991; Currie, 2003; Carr & Williamson, 2010).

Figure 12 Anterior and posterior sections of the surangular of Eotyrannus lengi IWCMS: 1997.550.

(A) Posterior section of right surangular in medial view; (B) lateral view; (C) dorsal view; (D) anterior section of left surangular in dorsal view (anterior to right); (E) lateral view; (F) medial view. acp anterior cotylar prominence, adch anterodorsal channel, doch dorsal channel, maco mandibular cotyle, pcp posterior cotylar prominence, retpr retroarticular process, smp subtriangular medial process, sush surangular shelf, trr transverse ridge.

The cotyle appears deep and U-shaped in lateral view but, viewed dorsally, it is broad and shallow. A subtriangular eminence forms its posterior border. A shallow, anteroventrally inclined fossa is present on the lateral surface of this posterior eminence, but there is no obvious lateral concavity continuous with the cotyle as there is in tyrannosaurids (Carr, 1999; Currie, 2003). The process anterior to the cotyle is continuous anteriorly with a prominent dorsolateral ridge–sometimes termed the surangular shelf–that projects laterally from the bone’s surface, subparallel to the bone’s long axis. This ridge is similar to the one present in Dryptosaurus and tyrannosaurids but its lateral edge does not curve ventrally as is the case in Daspletosaurus, Tarbosaurus and Tyrannosaurus (Brochu, 2003; Currie, 2003; Hurum & Sabath, 2003). A far less prominent ridge is present in Guanlong, Dilong and Proceratosaurus (Xu et al., 2004, 2006; Rauhut, Milner & Moore-Fay, 2010); that of Proceratosaurus is positioned close to the dorsal edge of the bone and is more dorsally positioned than typical for tyrannosauroids. In Dryptosaurus and tyrannosaurids the extreme posterior end of the ridge overhangs an enlarged posterior surangular foramen (Currie, 2003; Brochu, 2003; Holtz, 2004; Carr, Williamson & Schwimmer, 2005), and no such structure is present in E. lengi. Guanlong, Dilong and Proceratosaurus also lack posterior surangular foramina (Xu et al., 2004, 2006; Rauhut, Milner & Moore-Fay, 2010).

Medial to the ridge, the dorsal surface of the surangular forms the posterior part of the adductor muscle channel (Currie, 2003) which extends to the preserved anterior margin. The part of the bone ventral to the surangular shelf forms a mediolaterally compressed, blade-like region, the ventral edge of which is rounded. The anterior and anteroventral parts of the bone are absent.

A large, dorsally projecting process forms the anterior border of the mandibular cotyle, but the shape of the process cannot be determined because of damage at the apex. This process is continuous with a low transverse ridge that extends across the dorsal surface of a subtriangular medial process. The latter is continuous posteriorly with the retroarticular process and anteriorly with the posteromedial part of the surangular shelf. An extremely similar morphology is present in tyrannosaurids (Lambe, 1917). A short retroarticular process is present posterior to the cotylar region. In contrast to the tyrannosaurid condition, the dorsal surface of the retroarticular process is not separated from the posterior cotylar prominence by a concave area.

What appears to be the anterior end of a left surangular is well preserved (though only as fragments that had to be glued together), despite its delicate form: it is c. 1 mm thick except along its dorsal margin where a dorsomedial groove and accompanying medial shelf increase the thickness to c. 6–8 mm. The bulk of this bone fragment is composed of a thin, vertical lamina. Dorsally, this meets the subhorizontal dorsal margin (Figs. 12D–12F). The presumed anterior tip is missing, as is some of the ventral margin. Dorsomedially, a longitudinal shelf overhangs the medial surface and forms the medial border of a shallow gutter that extends to the presumed anterior tip.

What appears to be the lateral surface is convex and is deepest at a point just posterior to the termination of the shallow anterior gutter. A low dorsal peak is present here and is flush with the lateral surface. Immediately posterior to this convexity, a laterally directed concave area is present: it is bordered anteriorly and anteroventrally by a low rim. A subhorizontally oriented, anterodorsally located channel extends approximately in parallel with the bone’s dorsal margin (Fig. 12E). At the posterior end of this channel, an oval foramen perforates the bone: a delicate lamina extends dorsoventrally across part of this foramen. While the channel is inset medially into the bone, the lamina is continuous with the bone’s lateral surface.

Dentition

Approximately 17 teeth are known for E. lengi, some of which are preserved within their premaxillary, maxillary or dentary alveoli. The premaxillary teeth are typical for a tyrannosauroid while the maxillary and dentary teeth are of typical theropod morphology. The TCH (total crown height) of each tooth was measured and recorded if it was possible to distinguish the crown from the root. Given the ambiguous nature of the crown-root junction, the latter measurements are often approximate. Where possible the FABL (fore-aft basal length) was also measured and the denticle size difference index (DSDI) was calculated following Rauhut & Werner (1995). All tooth measurements are given in Table 2.

Table 2 Tooth measurements and denticle counts of selected teeth.

	Total preserved length	TCH	FABL	Serrations per 5 mm, mesial carina	Serrations per 5 mm, distal carina	DSDI	
pmx tooth 1	27	14	c. 5	–	–	–	
pmx tooth 2	27	18	7	15	14	1.071	
pmx tooth 3	51	c. 17	8	14	14	1.0	
l dentary tooth	11	11	c. 8	17	14	1.214	
lat tooth 1	59	c. 24	c. 12.5	–	16	–	
lat tooth 2	50	c. 26	c. 14	–	–	–	
lat tooth 3	23	23	c. 13	–	16	–	
lat tooth 4	13	–	–	19	14	1.357	
lat tooth 5	19	–	c. 12	–	14	–	
lat tooth 6	36	–	15	–	22	–	
lat tooth 7	>19	c. 19	–	c. 16	15	c. 1.067	
lat tooth 8	26	26	c. 13	–	–	–	
lat tooth 9	19	19	11	–	–	–	
Notes:

TCH, tooth crown height; FABL, fore-aft (mesial-distal) basal length, DSDI, denticle size difference index.

All measurements (excepting DSDI) in millimetres.

Premaxillary teeth

The premaxillary teeth of E. lengi are U-shaped in cross-section, as is typical for tyrannosauroids (Holtz, 2004; Xu et al., 2004; Hendrickx, Mateus & Araújo, 2015). At least three isolated E. lengi premaxillary teeth occur in the assemblage: they are easy to identify because of their cross-sectional shape and because their serrated carinae are restricted to the flat lingual surface (Figs. 13A–13C). A tooth that seems to be from the left premaxilla was figured in oblique lingual view by Hutt et al. (2001, fig. 8). A change in the colour and texture of the tooth indicates the position of the crown-root junction and suggests that c. 18 mm of the tooth was exposed as the crown. The crown is strongly convex labially while the lingual side is flat and bears an apicobasally elongate depression 4 mm long near the apex (Fig. 11C). This is presumably a wear facet. The preserved part of the root possesses a roughened external texture that appears to have resulted from bioerosion of some kind. The second premaxillary tooth is near-complete, attached to the right premaxilla and only exposed in lingual view: this reveals an oval depression 1.5 mm long near the tip of the lingual surface that resembles its counterpart on the first premaxillary tooth. The third specimen lacks any such lingual depression or facet.

Figure 13 Premaxillary and lateral teeth of Eotyrannus lengi IWCMS: 1997.550.

(A) Premaxillary tooth in lingual view; (B) oblique lingual view; (C) tip of premaxillary tooth in lingual view; (D) unidentified lateral tooth in lingual or labial view; (E) distal carina of lateral tooth; (F) tip of in-situ maxillary tooth in lingual view (the same tooth is visible in place on the maxilla in Fig. 4); (G) tip of in-situ dentary teeth. Images (A)–(C) and (F)–(G) were kindly provided by Christophe Hendrickx.

Dentary teeth

Four emergent tooth crowns are preserved within the left dentary, but only one of them (the 7th) protrudes dorsal to the alveolar margins (Fig. 13G). The crown tips preserved in the 1st, 3rd and 5th alveoli must have only recently emerged and the remains of a crown preserved anterodorsally to the 5th crown tip indicate that the newly emergent crown was in the process of displacing an older tooth. The location of the remnant of the older tooth relative to the emergent crown tip implies that replacement teeth emerged from behind their predecessors. The carinae of all the dentary teeth (with one exception) face mesially and distally. However, the tiny replacement tooth in the first alveolus, though broken and incomplete (consisting only of c. 2 mm of the base of the crown), is preserved with its longest axis directed labiolingually (Fig. 11F). The tooth is lenticular in cross-section and what appear to be unserrated carinae are preserved both lingually and labially. This suggests that the first dentary tooth differed strongly in shape from the other dentary teeth, and that this might be linked to the unusual morphology of the first alveolus. Unfortunately, it is possible that the tooth is not in situ, given its broken condition and non-central placement within the alveolus. Furthermore, no tooth is preserved in the first alveolus of the right dentary, making it impossible to confirm that the morphology of the first left dentary tooth is typical.

Remaining lateral teeth

At least 14 lateral teeth (sensu Hendrickx, Mateus & Araújo, 2015) are known for E. lengi, including isolated crowns, isolated partial roots, isolated crowns with roots, and teeth still embedded in the left maxilla (Fig. 13F) and dentary (Fig. 13G). An intact tooth crown representing an emergent tooth that has not fully descended is present in the first alveolus of the maxilla and a broken tooth crown is present in the third alveolus, the latter being 9 mm long mesiodistally and 5 mm wide labiolingually. The crowns have a lenticular cross-section and are of the form typical for ziphodont theropods; their roots are mediolaterally compressed and subrectangular in cross-section. All denticles terminate in squared-off ends, do not exhibit apical hooking, are slightly inflated apico-basally and slightly waisted, and are continuous across the crown apex (Figs. 11D–11G). The interdenticle pits are U-shaped. The distal denticles of E. lengi are notably taller (in terms of their height perpendicular to the tooth’s long axis) than the mesial denticles, with a height to basal width ratio of >1.5 for unworn distal denticles (Sweetman, 2004). At its base, the mesial carina appears prominent relative to the crown’s mesial margin.

Hutt et al. (2001, p. 230) reported a DSDI of 1.5 for E. lengi and noted that this was high compared to tyrannosaurids. Sweetman (2004) noted that this value should be considered unreliable as it was based on a partially erupted maxillary tooth in which denticle density could only be measured at the tooth tip. Remeasurement provided DSDIs of 1.03, 1.06, 1.25 and 1.31, with a mean of 1.16 (Sweetman, 2004). Similar DSDIs (1.21, 1.36 and 1.06, mean = 1.21) were calculated in the present study.

The axial skeleton of E. lengi

The vertebral formula of E. lengi is unknown but the number of vertebrae present in each segment of the vertebral column can be estimated based on the condition in other coelurosaurs. Tyrannosaurids and other typical non-avialan tetanurans possess 10 cervical, 13 dorsal, five sacral, and more than 35 caudal vertebrae (Makovicky, 1995; Holtz, 2004; Holtz, Molnar & Currie, 2004). These numbers are assumed for E. lengi. Hutt et al. (2001, p. 232) assumed that E. lengi possessed 14 dorsal vertebrae because of an adherence to the convention used by Madsen (1976) for Allosaurus. Madsen (1976), in turn, followed Osborn (1906, 1917) whose identification of nine cervical vertebrae for Tyrannosaurus rex was in error: though, to be fair, he noted how difficult it was to distinguish the last cervical from the first dorsal (Osborn, 1917, p. 765). It was subsequently argued by Makovicky (1995) that the eleventh presacral should be identified as the first dorsal since this is the first presacral to possess a hypapophysis. Brochu (2003) argued that any distinction made between cervical and dorsal vertebrae in tyrannosaurids is arbitrary, and subsequently referred to both simply as presacrals. It is of course unknown whether it would be possible to identify the cervical-dorsal junction in E. lengi. In order to facilitate description, the traditional distinction between these segments of the column is maintained here.

Cervical vertebrae: neural arches

No complete cervical vertebrae are preserved for E. lengi but two near-complete, isolated neural arches and two isolated centra are present (for measurements, see Tables 3, 4). The axial neural arch is embedded within a block that also includes a cervical centrum and the proximal ends of some probable metacarpals. A second neural arch is preserved on the same block as another cervical centrum and several probable cervical rib shaft fragments.

Table 3 Measurements of preserved cervical neural arches of E. lengi.

	Axial n. a.	2nd cervical n. a	
Neural arch length	50	72	
Width, across prezygs	63	–	
Width, space between prezygs	18	–	
Height, neural spine	10	–	
Length, neural spine	36	35	
Notes:

All measurements in millimetres.

Prezygs, prezygapophyses; n. a., neural arch.

Table 4 Measurements of preserved cervical vertebrae of E. lengi.

	Axial centrum	2nd cervical centrum	
Centrum length	40	37	
Width of anterior articular surface	38	35*	
Height of anterior articular surface	26	25*	
Mid width of centrum		–	
Width of posterior articular surface	20	–	
Height of posterior articular surface	–	–	
Notes:

Centrum length measured along ventral mid-line.

All measurements in millimetres.

* = estimated.

The axial neural arch (Figs. 14A–14D) is identified as such because of its flaring postzygapophyses, anteroposteriorly long neural spine and strong similarity to the axial centrum of Deinonychus (Ostrom, 1969, fig. 28D) and Xiongguanlong (Li et al., 2009, fig. 2c). The prezygapophyses are short, subtriangular prongs that extend 10 mm anterior to the neural spine. Neither preserves a complete articular facet. A broad, subtriangular space separates the prezygapophyses in dorsal view (Fig. 14C). The neural spine is low, subrectangular in lateral view, and extends along the entire length of the neural arch (Fig. 14B). It is somewhat distorted and appears to be missing its apex along its length, its posterodorsal portion being especially incomplete. There is no indication of the spine table present in other tyrannosauroids and coelurosaurs (Makovicky, 1995; Brochu, 2003; Li et al., 2009), presumably because of this damage. A small concavity (c. 1 mm in dorsoventral height) on the anterior face of the neural spine might be a ligament fossa. The prezygapophyses and postzygapophyses are at about the same horizontal level and are connected by a horizontal shelf that projects 11 (left side) and 13 (right side) mm lateral to the neural spine. None of the structures ventral to this shelf are preserved. The postzygapophyses flare posterolaterally and a low, mound-like, partly eroded epipophysis is present on the right side (Fig. 14D); the left epipophysis is missing entirely due to erosion. The epipophysis terminates at the posterior edge of the postzygapophyseal facet but may originally have been more extensive. Distinct postzygapophyseal facets are not preserved but appear to have been located in the typical position. There is no indication of a preserved axial intercentrum.

Figure 14 Cervical vertebrae of Eotyrannus lengi IWCMS: 1997.550.

(A) Axial neural arch in anterior view; (B) left lateral view; (C) dorsal view; (D) posterior view; (E) isolated cervical centrum in anterior view; (F) left lateral view; (G) left lateral view; (H) ventral view; (I) post-axial cervical neural arch in dorsal view; (J) oblique dorsolateral view, anterior to right; (K) right lateral view; (L) in posterior view; (M) in right lateral view. aas anterior articular surface, cenpozyglam centropostzygapophyseal lamina; cri cervical rib shaft, epi epipophysis, ligfo ligament fossa, ncs neurocentral suture, ns neural spine, par parapophysis, pnfo pneumatic foramen, poz postzygapophysis, prez prezygapophysis, prezf prezygapophyseal facet.

The second neural arch preserves all of its processes, though all are incomplete and many areas are damaged or obscured by irremoveable matrix (Figs. 14I–14M). In dorsal view the zygapophyses diverge laterally from the mid-line, creating an X-like shape (Figs. 14I–14K). The right prezygapophysis preserves a flat, dorsomedially directed articular facet. It is not possible to examine the space between the prezygapophyses, and the existence of interspinous ligament fossae remains uncertain. A displaced rod-like bone, possibly a cervical rib shaft, is preserved between the prezygapophyses and is described below. The broken neural spine is restricted to the posterior half of the neural arch. It extends posteriorly as far as the preserved posterior-most tips of the postzygapophyses (Figs. 14I, 14K, 14M). The latter are incomplete and there is no clear indication of epipophyses. On the left side the prezygapophysis is connected to the postzygapophysis by a near-horizontal shelf of bone, and the dorsal-most points of both the prezygapophysis and postzygapophysis are approximately at the same horizontal level (Fig. 14M). The postzygapophyses have their long axes directed posterodorsally; the precise orientation of their facets cannot be determined but they were evidently directed ventrally. On the left side, a partially visible centropostzygapophyseal lamina joins the underside of the postzygapophysis to the posterolateral part of the centrum (Fig. 14M).

Cervical centra and cervical ribs

A cervical centrum is preserved on the same block as the axial neural arch and may represent the axial centrum (Figs. 14E–14H). Its anterior articular surface is flat, and broader than it is deep. Both parapophyses are preserved; the right parapophysis bears a concave articular surface, though it is not possible to determine whether this is natural or the result of damage. Posterodorsal to the right parapophysis is a deep oval pneumatic foramen. If this centrum is indeed the axial one, then E. lengi shares with Dilong (Xu et al., 2004) and Xiongguanlong (Li et al., 2009) the primitive condition of possessing a single foramen on each side of the axis, rather than the two foramina per side present in tyrannosaurids (Makovicky, 1995; Brochu, 2003). What might be the serrated neurocentral suture is visible on the right side. The posterior articular surface is damaged but the curved form of the posterolateral rim of the centrum suggests that the articular surface was concave. The ventral surface of the centrum is flat but the junctions between the ventral and lateral surfaces are smoothly convex. There is no ventral keel or concavity. The posterior part of the centrum is narrower than the anterior articular surface.

A second, less well-preserved cervical centrum is preserved adjacent to the second neural arch. Most of its surfaces are damaged, but we describe here the better preserved left side. It is deeper and shorter than the other cervical centrum, and the parapophysis is in an anteroventral position with a deep oval pneumatic foramen located posterodorsal to it. The bone around the edges of the foramen slopes into this depression. As in the other cervical centrum, the junction between the lateral and ventral surfaces is smoothly convex. What is preserved of the articular surfaces indicates amphicoely or weak opisthocoely: the former condition is typical for tyrannosauroids (Li et al., 2009) while the latter is known in Juratyrant (Benson, 2008) and some tyrannosaurids (Holtz, 2004).

A rod-like bone fragment 55 mm long, preserved on the side of the block opposite the second cervical centrum, may be a cervical rib shaft. It cannot be determined which end is the proximal one, but a broken end reveals a subtriangular cross-section. A similar rod-like bone fragment c. 40 mm long is preserved in association with the neural arch, lying diagonally between the prezygapophyses (Figs. 14I–14K). This element appears circular in cross-section. The fact that two such rod-like elements are both located adjacent to cervical elements provides circumstantial support for their identification as cervical rib shafts.

Dorsal vertebrae

The presence of dorsal vertebrae in the holotype of E. lengi was mentioned by Hutt et al. (2001, p. 228) and several dorsal centra were alluded to in the description (p. 232). Five of these centra seem to be E. lengi dorsals but some additional centra are more problematic: two may not be dorsal vertebrae. All specimens have separated from their neural arches at the neurocentral sutures, but an isolated fragment of neural arch is also preserved. For measurements see Table 5. Pneumatic foramina are absent on all preserved dorsals.

Table 5 Measurements of preserved dorsal vertebrae of E. lengi.

	DV1	DV2	DV3	DV4	DV5	DV6	DV7	
Centrum length	69	65	60	50	c. 50	–	–	
Width, anterior articular surface	43	c. 45*	42	36†	c. 35†	39	54†	
Height, anterior articular surface	37	–	38	42†	40†	c. 32	–	
Mid width, centrum	22	30	25		19	–	–	
Width, posterior articular surface	51	52	55	37†	c. 35†	–	50†	
Height, posterior articular surface	c. 52	c. 50*	47	42†	–	–	–	
Notes:

Centrum length measured along ventral mid-line.

All measurements in millimetres. DV, dorsal vertebra.

* = estimated, as centrum incomplete.

† = identification of this articular surface as anterior or posterior was arbitrary, and the identification was made for ease of tabulation.

A large centrum that possibly represents the 3rd, 4th or 5th dorsal (on account of its proportional similarity to these vertebrae in other coelurosaurs: Makovicky, 1995; Brochu, 2003) is hourglass-shaped in dorsal view and ventrally concave in lateral view, with the deepest part of the concavity being 10 mm dorsal to the rims of the articular ends (Figs. 15A–15F). A faintly developed ventral keel is present, and the lateral side of the dorsal surface of the centrum (lateral to the neural canal) is concave. A neurovascular foramen c. 1.5 mm long is present on the left side. One articular surface, presumably the posterior one, is smaller than the other. The presumed anterior articular surface is flat, though near the left dorsolateral margin it forms an anterolaterally sloping surface which is surrounded posteriorly and laterally by a raised bony rim. The left parapophysis was presumably located here. The posterior articular face is concave, and the ventral, lateral and dorsal portions of the bony rim that surrounds it are more prominent than those surrounding the anterior articular face. The neural canal (c. 7 mm wide) is shallow for most of its length but becomes deep anteriorly. On the left side, the dorsal part of the centrum that borders the neural canal preserves a flat area for articulation with the neural arch.

Figure 15 Dorsal and sacral vertebrae of Eotyrannus lengi IWCMS: 1997.550.

(A) Centrum possibly representing 3rd, 4th or 5th dorsal vertebra in anterior view; (B) left lateral view; (C) ventral view; (D) posterior view; (E) right lateral view; (F) dorsal view; (G) second, less complete dorsal centrum in possible anterior view; (H) possible left lateral view; (I) ventral view; (J) possible posterior view; (K) possible right lateral view; (L) dorsal view; (M) middle dorsal centrum in presumed anterior view; (N) presumed left lateral view; (O) ventral view; (P) presumed posterior view; (Q) presumed right lateral view; (R) dorsal view; (S) probable posterior dorsal centrum in possible anterior view; (T) possible left lateral view; (U) ventral view; (V) possible posterior view; (W) possible right lateral view; (X) dorsal view; (Y) fragmentary unidentified dorsal centrum in ventral view; (Z) possible anterior view; (A′) possible posterior view; (B′) fragmentary unidentified dorsal centrum in ventral view; (C′) oblique probable posterolateral view; (D′) probable posterior view; (E′) sacral vertebra in anterior view; (F′) left lateral view; (G′) ventral view; (H′) posterior view; (I′) right lateral view; (J′) dorsal view; (K′) incomplete neural arch in anterior view; (L′) lateral view; (M′) probable fragment of transverse process, as visible in cross-section at break. latfo lateral fossa, lsp lateral subcircular pit, nc neural canal, rag radiating groove, rar radiating ridge, vk ventral keel.

The ventral half of a second, less complete dorsal centrum is medially constricted at mid-length when viewed dorsally or ventrally (Figs. 15G–15L). There is no trace of a ventral keel, unlike in tyrannosaurids and most other tetanurans (Rauhut, 2003b). One articular surface extends further ventrally than the other: in the dorsal vertebrae of tyrannosauroids and other tetanurans, it may be either the anterior or the posterior articular surface that extends furthest ventrally (Harris, 1998; Brochu, 2003; Brusatte, Carr & Norell, 2012), rendering it impossible to decide with certainty which end the ventrally descending one represents. The preserved height of the more extensive articular surface is 32 mm at most, but when complete it was probably c. 50 mm tall. The opposite articular face was probably c. 45 mm wide when complete. The incompleteness of the articular faces makes it difficult to determine their original shape but they seem to have been flat.

A third, robust centrum possesses concave lateral surfaces and a flatter ventral surface than the two preceding elements (Figs. 15M–15R). Again, it cannot be determined with confidence which end is which but the end where the articular surface descends further ventally is assumed to be the posterior one. The neural canal and adjacent structures are not preserved. Bony rims surround both articular surfaces. The presumed anterior articular surface is flat while the presumed posterior one is slightly concave and slopes anterodorsally. On the basis of how it compares with the other dorsal vertebrae known for E. lengi, this vertebra was presumably posterior to the 4th or 5th position and may belong to the middle part of the dorsal series.

A somewhat distorted fourth dorsal centrum is 52 mm long on one side and 58 mm long on the other (Figs. 15S–15X). Again, it is not possible to determine which end is which: one articular surface is flat and the other is concave; the latter is arbitrarily identified as ‘anterior’. This centrum is probably the one described by Hutt et al. (2001, p. 232) as representing dorsal 14 and being 52 mm long. The centrum is elongate and deeply concave ventrally. Both sides exhibit oval concavities that are located slightly closer to the neurocentral sutures than to the ventral surface and occupy much of the length of the centrum between the edges of the articular faces (Figs. 15T, 15W). The concavities are asymmetrical, partly due to distortion of the centrum: one is 34 × 15 mm and the other 27 × 17 mm. There are no indications of pneumaticity within the concavities. Lateral concavities of this sort, albeit not as well defined, have been illustrated for some allosauroids (Madsen, 1976; Currie & Zhao, 1994). The bony rims around the articular faces flare laterally and ventrally. The ventral surface of the centrum is convex with no midline keel (Fig. 15U).

A poorly preserved, highly pyritised core of what appears to be the centrum of a fifth, camellate dorsal vertebra reveals little detail but does display an hourglass-like shape in ventral or dorsal view. The orientation of the element cannot be determined (it is missing all external bone texture and is embedded in matrix on most sides), but one articular end measures 40 mm dorsoventrally and c. 35 mm mediolaterally. The opposite articular end is also c. 35 mm wide but its depth cannot be measured. The preserved part of the vertebra represents either the dorsal or the ventral part of the centrum (rather than its middle) and consequently the centrum may have been even narrower closer to its middle. The ‘waisted’ proportions of this centrum appear typical for a tetanuran. There is no way of determining its position within the vertebral sequence. A second probable centrum ‘core’ is enclosed in matrix. Its approximate dimensions are 50 × 50 mm, but it cannot be determined whether it belongs to E. lengi or the associated dryosaurid.

Several additional dorsal vertebrae are preserved in the assemblage, mostly represented by fragmentary and distorted sections of centra (Figs. 15Y–15D′). It is not possible to determine their positions within the sequence, or even be confident that they belong to E. lengi. A probable fragment of a dorsal neural arch (Figs. 15K′–15L′) is also present. It is robust with a maximum length of 57 mm and a maximum breadth of 47 mm. Its incompleteness makes interpretation difficult and no part of the element is bilaterally symmetrical. It is most likely from the right posterolateral part of the neural arch, in which case the process projecting from it represents an incomplete transverse process. A low ridge with a length of 37 mm extends along what may be the ventrolateral part of the specimen. The large size of this fragment indicates that it belonged to a dorsal vertebra but it is not possible to be more specific.

The relatively long dorsal centra of E. lengi are unlike those of tyrannosaurids. Measurements of the Daspletosaurus torosus holotype NMC 8506 given by Russell (1970) show this specimen to have dorsal centrum length: height ratios ranging from 0.62 to 0.83, with a mean (n = 7) of 0.70. E. lengi has much higher ratios ranging from 1.19 to 1.86, with a mean (n = 5) of 1.44. The dorsal vertebrae of Dilong were described as “relatively long” (Xu et al., 2004, p. 681) and this also appears to have been the case for Guanlong (Xu et al., 2006, fig. 1). In Juratyrant, length: height ratios range from 1.03 to 1.38, with a mean (n = 3) of 1.16 (Benson, 2008). We included posteriormost dorsal centra length relative to height in our analysis (char. 232) and found relatively elongate dorsal centra (longer than or subequal to anterior centrum height) to be a typical and plesiomorphic feature of non-tyrannosaurid tyrannosauroids.

Sacral vertebra

A single sacral centrum is known for E. lengi (Figs. 15E′–15J′) and was described by Hutt et al. (2001, p. 232), who suggested that it was the last sacral. It is here regarded as the first due to reinterpretation of the heavily scarred articular face of the centrum as the posterior one (for measurements, see Table 6), a decision guided by other aspects of anatomy (like the position of the paired oval foramina within the floor of the neural canal, discussed below). As with the better preserved of the dorsal vertebrae, it is mostly complete ventral to the neurocentral suture. The neural arch is absent. It is shallow and broad compared to the dorsal centra and has a wide, deep neural canal.

Table 6 Measurements of preserved sacral and caudal vertebrae of E. lengi.

	CC	sacral	CV1	CV2	CV3	CV5	CV6	
Preserved length**	36	68	25	40	25	32	24	
Width, anterior articular surface	32†	47	40†	37†	20†	–	–	
Height, anterior articular surface	28*†	26	–	–	29†	–	–	
Mid width, centrum	–	–	32	–	–	–	–	
Width, anterior articular surface	–	53*	–	–	–	25*	22	
Height, posterior articular surface	–	30*	–	–	–	22	–	
Height, neural canal	–	–	–	–	–	6	6	
Width, neural canal	–	30	–	–	–	5	5	
Notes:

Length measured along ventral mid-line.

All measurements in millimetres.

CC, “cervical/caudal vertebra”: a specimen whose position within the vertebral column could not be determined with certainty (see text), CV, caudal vertebra, Two asterisks (**) equals measurements are preserved lengths because none of these vertebrae (except the sacral vertebra) are complete, One asterisk (*) equals estimated, as centrum incomplete, † = identification of this articular surface as proximal or distal was arbitrary, and the identification was made for ease of tabulation.

The anterior articular face is broad, slightly concave and shallow dorsoventrally. Large, rugose, concave facets for reception of the sacral ribs and transverse processes are present dorsolateral to the articular face. That on the left is more complete and has a width of 28 mm and a length of 25 mm. An oblique groove divides the left facet into two halves, possibly demarcating the attachment area for the sacral rib from that of the transverse process.

Approximately halfway along the length of the centrum, the ventral halves of large, dorsally positioned oval foramina are present (Figs. 15F′, 15I′). That on the left side is better preserved and has a complete ventral bony rim. It is 11 mm long and 5 mm tall as preserved. The right foramen is less well preserved and has incomplete margins but appears to have had similar dimensions. Both foramina are set within larger lateral concavities and communicate medially with the neural canal. These foramina are in the same position as the large sacral nerve foramina illustrated by Welles (1984) for Dilophosaurus and similar foramina are present at mid-centrum, at the neurocentral suture, in Juratyrant (Benson, 2008). However, openings interpreted as pneumatic foramina occur in precisely the same location in tyrannosaurids (Brochu, 2003, p. 89). In E. lengi, the connection of the foramina to the neural canal indicates that they are spinal nerve foramina. The neural canal is shallowest anteriorly, deepening posteriorly and also widening to 30 mm (Fig. 15J′). Posteriorly, two lateral subcircular pits are present on its floor, each c. 10 × 10 mm (Fig. 15J′). These structures are rarely described or illustrated but the pits seen in E. lengi appear typical for theropods and perhaps for saurischians as a whole. Osmólska, Roniewicz & Barsbold (1972) described (but did not illustrate) “a pair of large and deep pits for the spinal ganglions in the anterior portion of each vertebra” in the 2nd and 3rd sacral vertebrae of Gallimimus bullatus (p. 122) and apparently homologous structures were figured in a titanosauriform sauropod by Carpenter & Tidwell (2005, fig. 3.7G).

The posterior articular face of the centrum is broader and deeper than the anterior surface. The posterior face is flat but bears a series of radiating grooves and ridges that create a star-burst pattern (Fig. 15H′) that is typical of unfused sacral centra (e.g., Madsen, 1976, plates 25–27). The posterior articular surface is deeper than the anterior one, the posterior end descending further ventrally. The ventral surface is concave in lateral view. A low ventral keel is present along the midline and is most pronounced over the anteriormost 30 mm of the centrum (Fig. 15G′).

Caudal vertebrae

Five poorly preserved, incomplete caudal vertebrae are tentatively identified as belonging to the distal part of E. lengi’s caudal skeleton: this cannot be confirmed, however, since they provide little information and do not possess any features that are typical for coelurosaurs (such as long prezygapophyses that overlap the centrum of the adjacent vertebra) (for measurements, see Table 6). Three of these vertebrae are represented only by partial centra preserving parts of their articular surfaces. Where known, the ventral surfaces of the centra are convex and lack midline keels or other structures. The additional two distal caudals–both of which preserve a partial neural arch–are incomplete proximally. One has a proximal centrum width of 15 mm but flares outwards distally such that the concave distal articular face would have been at least 25 mm wide at mid-height. Camellate bone texture is visible on the floor of the neural canal. A neural spine was probably absent, suggesting that this vertebra was distal to c30. The incomplete right postzygapophysis extends 3 mm distal to the articular face, is positioned close to the midline, and curves distomedially, indicating that the prezygapophyses in this region of the tail must have been very close together. At most, the postzygapophysis is 12 mm dorsal to the centrum, whereas the neural arch at the proximal end of the centrum is only c. 5 mm dorsal to the centrum. A shallow, laterally facing concavity separates the base of the postzygapophysis from the centrum. The lateral surfaces of the centrum are convex but bear low proximodistally oriented ridges. The ventral surface is flattened and chevron facets are absent.

A second, smaller distal caudal vertebra is even simpler in structure. The postzygapophyses were clearly short and close to the midline and a neural spine seems to have been absent. Again, poorly developed ridges are present on the laterodorsal region of the centrum and are parallel to the centrum’s long axis. The centrum flares laterally at its distal end. The maximum width of the centrum at its broken proximal end is 16 mm. The postzygapophyses of both of these vertebrae are unusually short when compared with those of the distal caudal vertebrae of other tyrannosauroids (Brochu, 2003; Carr, Williamson & Schwimmer, 2005) but are comparable in approximate proportions to those present in some other coelurosaurs, including compsognathids and Bagaraatan (Ostrom, 1978; Osmólska, 1996; Currie & Chen, 2001).

Dorsal ribs

Several rib fragments are associated with the E. lengi holotype (Fig. 16), though this was not mentioned by Hutt et al. (2001). None are preserved in articulation or association with vertebrae, so the precise positions of these fragments cannot be determined. Given the presence of a dryosaurid in the assemblage, it is possible that some of these bones do not belong to E. lengi, and only the more informative specimens are described here. Ribs are described as imagined in articulation, with directional terms corresponding to those that would apply to a complete ribcage.

Figure 16 The more informative of the several rib fragments known for Eotyrannus lengi IWCMS: 1997.550.

(A) Dorsal rib segment in oblique view to show cross-sectional shape of capitulum; (B) same dorsal rib segment but in presumed anterior view; (C) dorsal rib segment in presumed anterior view, showing pneumatic recess; (C) and (D) partial rib shaft with flange-like lateral extension, shown in anterior and posterior views. cap capitulum, cog costal groove, intr intercostal ridge, pnre pneumatic recess, tub tuberculum.

A rib fragment 80 mm long, preserving the bases of both the capitulum and tuberculum, adheres to a block and is exposed in anterior view (Figs. 16A, 16B). The capitulum is compressed, being 16 mm deep but just 6 mm wide. The tuberculum is also compressed: in cross-section it is 3 mm wide at its medial end but c. 12 mm wide laterally. Its lateral edge is continuous with the intercostal ridge. The anteromedial edge of the rib shaft is thickened and anteriorly convex but the shaft becomes concave toward its lateral margin. The anterior surface of the shaft, when complete, was concave for all or most of its length, forming a distinct costal groove. Anterolaterally, the intercostal ridge forms the border of this concavity (and the boundary between the anterior and lateral surfaces of the shaft). The ridge presumably extended for the entire length of the shaft. The shaft is thus U-shaped in cross-section. The lateral surface of the rib is convex and meets the posterior surface at a prominent angle.

The dorsal 41 mm of a rib preserves a near-complete capitulum and tuberculum but only the dorsal ‘neck’ of the shaft (Fig. 16C). It is preserved in close association with three shaft fragments. From the lateral edge of the tuberculum to the medial tip of the capitulum it is 44 mm long, and from their apparent closeness it would seem that this rib was from the anterior part of the ribcage, probably representing one of the first five dorsal ribs. What appears to be a pneumatic recess is present at the dorsal end of the shaft (this pneumatic structure confirms that the rib belongs to E. lengi) but is partially concealed by a shelf that overhangs it from the lateral side. The preserved part of the shaft is sub-oval in cross-section.

The longest rib fragment has a preserved length of 115 mm and represents the ventral end of a shaft (Figs. 16D, 16E). Its dorsal end is oval in cross-section but the ventral part is rounded with a blunt termination capped with unfinished bone. The dorsal part of the medial surface is compressed into a keel-like edge that extends c. 50 mm ventrally. The ventral 50 mm of the shaft is narrower mediolaterally (c. 13 mm) than the dorsal section (c. 25 mm). On either the anterior or posterior surface (it is not possible to determine which is which) a lateral flange-like extension projects from the shaft. No similar structure has been reported elsewhere in Theropoda, suggesting either that this rib does not belong to E. lengi or that E. lengi was unique among theropods in this respect.

Pectoral girdle and forelimb skeleton

The morphology of the E. lengi forelimb was well characterised by Hutt et al. (2001). That preliminary study noted the presence in E. lengi of a gracile humerus, trochleated carpus and gracile metacarpal I with a strongly asymmetrical distal end, as well as proportionally elongate phalanges in at least two of the digits and strongly curved unguals (Hutt et al., 2001). The literature on theropod limb osteology features inconsistencies in the application of directional terms, at times because authors interpret bones in their imagined life postures (e.g., Johnson & Ostrom, 1995; Charig & Milner, 1997). Here, bones are described in the conventional fashion, i.e., with the flexor surface of the manus described as ventral even though this likely faced medially in life (Gishlick, 2001; Senter & Robins, 2005). E. lengi is assumed to have possessed three or four metacarpals on the basis of comparison with non-tyrannosaurid tyrannosauroids represented by better remains (Xu et al., 2004, 2006).

Scapulae

Both scapulae of E. lengi are preserved, although only part of the right scapula blade is present. The main new discovery regarding the morphology of the scapula since Hutt et al. (2001) is the morphology of the bone’s dorsal termination.

The left scapula is almost complete (Fig. 17) and preserved in partial articulation with part of the left coracoid (it is also attached to a manual phalanx II-2). At its ventral end the scapula expands to form a large acromion process. Here the bone is c. 85 mm long anteroposteriorly. The anteroventral margins of the scapula are not preserved so it cannot be determined if the acromion region was squared-off as is the case in tyrannosaurids and some other theropods (e.g., Madsen, 1976; Currie & Carpenter, 2000; Brochu, 2003). In lateral view, the anterior margin of the acromion processes possesses a small, anterodorsally projecting convexity. The posteroventral margin of the scapula (ventral to the blade section of the bone) has an arcuate, uninterrupted edge. Dorsally, the bone is composed of a mediolaterally compressed, strap-like blade c. 280 mm long. Both the anterior and posterior margins of the blade are expanded, with the expansion along the anterior edge beginning further dorsally than is the case on the posterior side. The anterior and posterior borders of the blade are subparallel and the medial surface of the blade is concave. At approximate mid-length the blade is c. 32 mm wide. The blade has a mediolateral thickness of c. 5 mm.

Figure 17 Scapulae of Eotyrannus lengi IWCMS: 1997.550.

(A) Partial shaft of left scapula in lateral view, as preserved in matrix; (B) reconstructed left scapula combining segment of shaft shown in (A) with reconstructed shape of dorsal end, and ventral region including acromion and partial glenoid. (C) Incomplete shaft of right scapula in lateral view. acrp acromion process.

The dorsal tip of the blade is preserved within one of the blocks and can only be viewed in cross-section. It fits on to the rest of the blade and probably added 20 mm or so to the blade’s length. With a maximum width of c. 50 mm this fragment (which was unknown to Hutt et al., 2001) shows that the tip of the scapula was wider than the shaft, and it would appear that this expansion was abrupt. The scapula’s surface is not sufficiently well preserved to reveal the locations of muscle attachment sites.

The right scapula is represented only by two fragments of the blade, one of which appears to belong to the dorsal end and the other to a more ventral part of the bone (Fig. 17C). The more dorsal part is subrectangular, being 70 mm long and c. 40 mm wide. The cross-section of this fragment is lenticular and c. 8 mm thick. The second part of the right scapula has a maximum preserved length of 130 mm, and one end is wider than the other (36 mm vs 30 mm). The longer end appears to be the more dorsal one. At its probable dorsal end, the blade fragment is strongly compressed in cross-section, being at most 7 mm thick, whereas its ventral end is thicker (15 mm). The inferred lateral surface of the blade fragment is convex while the inferred medial surface is flat.

The slender-bladed scapula with its broad acromion and dorsally expanded tip resembles those of tyrannosaurids (Holtz, 2004). However, several of the features present in E. lengi have a wide distribution within coelurosaurs, including an expanded dorsal tip, subparallel anterior and posterior edges and broad acromion, and tyrannosaurids tend to have a scapula that is narrow ventrally and gradually widens dorsally. The scapula of Dilong widens gradually in the dorsal direction (Xu et al., 2004). Guanlong lacks a dorsal expansion of the scapular blade, and its blade widens only slightly towards its dorsal end (Xu et al., 2006). Ornithomimosaurs and maniraptorans lack an expanded scapular tip and are thus also unlike E. lengi (Rauhut, 2003b).

Coracoid

The left coracoid (Fig. 18) remains partially articulated with the base of the left scapula. It is mostly complete and preserves an intact glenoid cavity and margin posteroventral to the glenoid (Figs. 18B, 18C); most of the other margins are damaged. The coracoid appears to have been typical for tetanurans in being semicircular overall with a posteriorly directed glenoid and a short posteroventral process which is separated from the coracoid body by a shallow posterior notch. The preserved height of the coracoid is 85 mm and its length is 70 mm. The body is deeply concave medially and convex ventrolaterally. A prominent coracoid tubercle projects c. 8 mm from the lateral coracoid surface (Figs. 18A, 18D); it is robust and subtriangular in lateral view, but with a flattened apex. As discussed by Hutt et al. (2001, p. 233) there is no indication of a coracoid foramen. Bambiraptor feinbergi was also reported to lack this feature (Burnham et al., 2000). However, damage to the bone surface in E. lengi renders this inconclusive: it is most likely that a coracoid foramen was present originally (given that one is typical across those tetrapods that possess a coracoid) but now obscured by damage. Dorsal to the coracoid tubercle, and adjacent to the coracoid’s dorsal edge, there is an elongate, oval concavity (Fig. 18D).

Figure 18 Left coracoid of Eotyrannus lengi IWCMS: 1997.550.

(A) Coracoid in anterior view; (B) oblique posteroventral view; (C) posterior view; (D) lateral view, anterior to the left; (E) posterior view; (F) medial view, anterior to the right; (G) oblique dorsomedial view to show concave medial surface; (H) ventral view. cortu coracoid tubercle, doc dorsal concavity, epm embayed posterior margin, glef glenoid fossa, mec medial concavity, pvp posteroventral process.

The glenoid fossa is slightly concave and, as in tyrannosaurids (Brochu, 2003, p. 94) and dinosaurs in general, the coracoid probably formed the ventral half of the fossa. The fossa is wide (26 mm) and at least 27 mm tall. The adjacent region of the coracoid body is also thick, being 15 mm wide just ventral to the glenoid. The posteroventral process is incomplete posteriorly; much of the process is positioned posterior to the glenoid if the bone is imagined in life position (with the glenoid directed somewhat dorsally). When complete, the process probably extended for a further 20 mm or so. In Dilong, the coracoid’s posteroventral margin lacks an embayment (Xu et al., 2004, fig. 1i).

The ventral part of the medial surface of the bone is flat. The anterior and ventral margins are 1–3 mm thick and thus far thinner than the posterior margin. The anterior margin is damaged and the bone has been deflected away from the original union with the acromion process of the scapula. What might be part of the anterior region of the coracoid is attached to the scapula but does not provide any useful information.

Humerus

Both humeri of E. lengi are known (Fig. 19) and a preliminary description of the right humerus was provided by Hutt et al. (2001, p. 233). The right humerus is almost complete though preserved in two pieces (it is broken just distal to the deltopectoral crest) and lacking part of the proximal end, parts of the shaft and deltopectoral crest, and some of the distal condyles. With a total length of 240 mm, the humerus is gracile and long-shafted with an anteriorly curving distal end and prominent subtriangular deltopectoral crest. Because the left element is less complete, most observations presented here are based on the right element.

Figure 19 Humeri of Eotyrannus lengi IWCMS: 1997.550.

(A) Right humerus in posterior view; (B) anterior view; (C) lateral view; (D) proximal view; (E) medial view; (F) proximal part of right humerus in medial view; (G) proximal part of left humerus in lateral iew; (H) distal part of right humerus in anterior view; (I) distal part of right humerus in oblique view to show concave area in middle of distal end; (J) distal part of right humerus in posterior view; (K) distal part of right humerus in lateral view; (L) distal end of right humerus, anterior toward top of page; (M) cross-sectional view of shaft of right humerus to show internal cavities. dpc deltopectoral crest, ent entepicondyle, grt greater tubercle, hh humeral head, int internal tuberosity, lco lateral condyle, mco medial condyle.

The long axes of both the proximal and distal ends are parallel. In this respect E. lengi more closely recalls Dryptosaurus (Brusatte, Benson & Norell, 2011) and tyrannosaurids (Brochu, 2003) than Dilong or Guanlong: in the latter two, the distal end of the humerus is deflected such that its flexor surface is angled somewhat medially (Xu et al., 2004, 2006). The rounded proximal head is wide and subcylindrical and without the inflated, hemispherical morphology present in tyrannosaurids (Brochu, 2003). E. lengi is more similar in its humeral head morphology to the plesiomorphic tyrannosauroid condition of Dilong or Guanlong in this respect (Xu et al., 2004, 2006). The head is tallest and most bulbous medially and is connected laterally to a lower convexity that recalls the “greater tubercle” of Madsen & Welles (2000) (Figs. 19A, 19B). The latter feature is unexpected since there does not appear to be a similar structure in Dilong or Tyrannosauridae (Brochu, 2003; Xu et al., 2004), raising the possibility that this is an autapomorphy of E. lengi. The condition in both Tanycolagreus and Guanlong is unclear due to damage (Carpenter, Miles & Cloward, 2005; Xu et al., 2006).

At its medial border the proximal end protrudes anteromedially to produce an internal tuberosity (Fig. 19B), though this is incomplete. Brochu (2003, p. 97) noted that E. lengi appeared superficially similar in internal tuberosity morphology to tyrannosaurids but wondered if this was due to damage. On the posterior surface of the humerus, the head and greater tubercle appear notably convex and posteriorly prominent relative to the humeral shaft but there is no distinct furrow or other structure that demarcates them from the rest of the bone.

Extending distally along the humeral shaft for approximately 80 mm, the deltopectoral crest is one-third the length of the humerus, similar in extent to the crests of Dryptosaurus (Brusatte, Benson & Norell, 2011) and tyrannosaurids (Brochu, 2003). In Dilong and Guanlong the crest is 40–50% of humerus length (Brusatte, Benson & Norell, 2011). Viewed laterally, the deltopectoral crest is subtriangular with a distally located apex and a 90° angle between its distal margin and the anterior face of the humeral shaft (Figs. 19C, 19E, 19F). The crest’s distal margin grades smoothly into the shaft, its edge describing a shallow arc. The crest’s anterior edge is rugose and serrated, though this mostly appears to be the result of erosion. None of this apical, anteriormost area overhangs the adjacent part of the anterior face of the humerus or is wider than the crest at mid-height. The crest’s base is wide relative to the apex, such that the crest is subtriangular in cross-section. The crest’s lateral surface is rugose but muscle scars cannot be identified.

The proximal part of the anterior surface of the humeral shaft is concave, being bordered on its medial side by a raised rim that extends proximally to reach the internal tuberosity. The humeral shaft is cylindrical and measures 30 mm in width and 25 mm perpendicular to this at its point of breakage. Greater bone thickness in the anterior part of the shaft is associated with the presence of the deltopectoral crest. Hutt et al. (2001, p. 233) described the presence of four internal compartments: a large posterior one (15 × 10 mm) and three smaller anterior ones (between 2 and 6 mm in width) (Fig. 19M). The left humerus is also broken across the shaft, though in this case at a level some 60 mm distal to the deltopectoral crest. Though the shaft is crushed and damaged, it is internally hollow with no indication of distinct compartments. The internal structure suggested to be diagnostic for E. lengi by Hutt et al. (2001, p. 229) has therefore proved not significant after close anatomical investigation.

The humerus is 48 mm wide across its distal condyles. The distal condyles are located more on the anterior surface of the humerus than on the distal end. They are damaged with most of the bone surface missing but it does not appear from their proportions that either was particularly bulbous. The medial condyle is slightly larger than the lateral condyle and connected to it by a shelf of bone, proximal and ventral to which are concavities. Proximomedial to the medial condyle is a prominent entepicondyle (Figs. 19A, 19B, 19H–19J). Similar structures are known in alvarezsaurids (Novas, 1996, 1997), oviraptorosaurs (Osmólska, Currie & Barsbold, 2004), therizinosauroids (Barsbold, 1976; Perle, 1981; Clark, Maryańska & Barsbold, 2004; Kirkland et al., 2005) and dromaeosaurids (Ostrom, 1969; Brinkman, Cifelli & Czaplewski, 1998). The massive entepicondyle of alvarezsaurids is bulbous, subconical and slightly curved medioproximally (Novas, 1996, 1997), unlike the more block-like, medially projecting structure of E. lengi. In oviraptorosaurs the entepicondyle has been described as projecting anteriorly (Osmólska, Currie & Barsbold, 2004, p. 175). This is unlike the condition in E. lengi in which the structure is directed medially. The hypertrophied entepicondyle of therizinosauroids is unlike that of E. lengi in being clearly demarcated at its distal edge from the medial condyle but is similar to the entepicondyle of E. lengi in approximate proportions. Similar large entepicondyles are not present in other tyrannosauroids including Tanycolagreus, Dilong or members of Tyrannosauridae (Russell, 1970; Brochu, 2003; Carpenter, Miles & Cloward, 2005; Xu et al., 2004), nor are they typical outside of Coelurosauria (e.g., Madsen, 1976); in these taxa the same region of the humerus forms a subtle medial convexity, a structure that might be termed an entepicondyle being absent. The form of the E. lengi entepicondyle is thus regarded as autapomorphic within Tyrannosauroidea. Proximal to the entepicondyle, the medial margin of the shaft is concave when the humerus is viewed anteriorly (Fig. 19I). There is no suggestion of an ectepicondyle. The lateral side of the distal end of the humerus is flattened with a sharp ridge marking the anterior extent of the flattened lateral surface.

Radius

What appears to be a fragmentary radius is preserved on the same block as the putative second and third metacarpals, the axial neural arch and a cervical centrum. It was identified as such by Hutt et al. (2001, p. 233). The element is 74 mm long and is broken proximally and distally (Figs. 20A–20C). It consists of a robust, subcylindrical shaft that flares out to one side at one of its ends. A shallow subtriangular concavity is present on the side of this flaring region (Fig. 20B). This concavity recalls the lateral radial facet that receives the proximal coronoid process of the ulna; if this is correct, the bone is from the left side. The opposite end resembles the ulna in having a teardrop-like cross-section (Fig. 20C), which is 32 mm deep and 16 mm wide. The end with the flaring region is identified as the proximal end.

Figure 20 Partial ?left radius and left ulna of Eotyrannus lengi IWCMS: 1997.550.

(A) ?Left radius in dorsal view; (B) in medial or lateral view; (C) proximal cross-section; (D) left ulna in medial view; (E) proximal cross-section; (F) distal cross section; (G) reconstructed forelimb anatomy of E. lengi showing inferred positions of known elements. subco subtriangular concavity, vk ventral keel.

Ulna

The incomplete shaft of a presumed right ulna was described by Hutt et al. (2001, p. 233). The preserved length of the element is 90 mm and its lateral and laterodorsal surfaces are obscured by matrix. Both the proximal and the distal ends are broken (Figs. 20D–20F). At one end the element is subrectangular in cross-section (Fig. 20F), being 25 mm wide and 22 mm deep, while at the other it is subovoid, being 30 mm tall and 16 mm wide at mid-height and wider dorsally than ventrally. The resulting “inverted tear-drop” shape (Fig. 20E) seems to be unique and is regarded as a potential autapomorphy of E. lengi; the same condition is present in the oviraptorosaur Avimimus (Vickers-Rich, Chiappe & Kurzanov, 2002) but is assumed to be convergent. In tyrannosaurids the ulna in cross-section is subrectangular distally but subovoid proximally, so the ends of the element are identified accordingly. Viewed in profile, one edge of the element is slightly concave while the other is slightly convex and marked with a longitudinal keel. The concave edge is assumed to be the dorsal one and the convex edge the ventral one, in which case the longitudinal keel is a ‘ventral keel’ (Figs. 20D, 20E). It is less easy to determine whether the element is from the left or right side. One of its sides is convex while the other is flat; the convex side seems more likely to be the lateral one, in which case the element is from the animal’s left. The description given by Hutt et al. (2001, p. 233) does not match the element and should be ignored.

Distal carpal I

A well-preserved, complete distal carpal is known for E. lengi and is regarded here as a left distal carpal I (Fig. 21). Hutt et al. (2001, p. 233) described this bone briefly and suggested that it was a radiale due to its similarity to the bone of Deinonychus antirrhopus described as a radiale by Ostrom (1969, pp. 98-99) but now known to be a distal carpal I (Chure, 2001). The distal carpal of E. lengi is complex; due to its discovery in isolation (that is, its being unconnected to any adjacent element), our identification of its several surfaces is based on a perceived homology with distal carpal I in those other tetanurans for which this element has been adequately figured and described, namely Allosaurus (Chure, 2001), Coelurus (Carpenter et al., 2005), Tanycolagreus (Carpenter, Miles & Cloward, 2005), Guanlong (Xu et al., 2006) and Falcarius (Zanno, 2006). The element in E. lengi is especially similar to that of Guanlong. In both proximal and distal view, the element is slighter wider than tall, the articular surface covering virtually the whole of the bone and with a transverse groove or step dividing the surface into equal dorsal and ventral facets (Figs. 21A, 21B). On the distal surface, these two facets have different orientations: the dorsal facet faces distolaterally while the ventral facet is directed distomedially. Presumably, the dorsal facet received the proximal end of metacarpal I while the ventral facet received metacarpal II. This is consistent with the dorsoventrally shallow proximal articular face of metacarpal I (the longest axis of which is obliquely oriented relative to the bone’s long axis: Fig. 22F) and perhaps with the broad, shallow proximal end of metacarpal II (Fig. 22H). If interpreted correctly, this requires that metacarpal I articulated with the carpal in a more dorsal position than metacarpal II, a configuration depicted elsewhere in Tanycolagreus (Carpenter, Miles & Cloward, 2005). A block-like dorsal process projects dorsolaterally, being more continuous with the proximal surface than the distal one (Figs. 21A, 21B). Its apex is eroded and it was originally a slightly larger structure.

Figure 21 Left distal carpal I of Eotyrannus lengi IWCMSS: 1997.550.

The directional terms used here imagine the manus with its extensor surface oriented dorsally. (A) Distal view; (B) proximal view; (C) dorsal view; (D) lateral view; (E) ventral view; (F) medial view. dof dorsal facet, dorpr dorsal process, trgr transverse groove, vef ventral facet.

Figure 22 Metacarpals of Eotyrannus lengi IWCMS: 1997.550.

The directional terms used here imagine the manus with its extensor surface oriented dorsally. (A) Left metacarpal I in dorsal view; (B) ventral view; (C) medial view; (D) lateral view; (E) proximal view; (F) distal view; (G) block containing proximal ends of metacarpals II and III, with distal ends of metacarpals facing viewer; (H) incomplete left metacarpal II in probable ventral view; (I) distal view showing broken cross-section (with orientation rotated relative to (H) so that probable ventral view is in ventral position); (J) incomplete left metacarpal III in distal view showing broken cross-section; (K) probable medial view; (L) block showing proximal ends of metacarpals II and III, seen from above (at 90° relative to view shown in G). fmc facet for metacarpal II, lco lateral condyle, mcII metacarpal II, mcIII metacarpal III, mco medial condyle, metub medial tubercle.

The dorsal surface of the carpal is incomplete along its distomedial margin, the bone being broken and eroded along the remainder of the medial margin as well (Fig. 21C). The surface is otherwise concave and surrounded by a raised margin that is thickest and most promiment distolaterally. This thick margin merges with a rounded ridge that extends ventrally along the bone’s lateral edge. Mid-way along the distal edge, a projecting peak is present, the result being a subtriangular margin. The proximal margin of the dorsal surface is simply convex. The concave part of the dorsal surface is rugose and marked with numerous foramina, small furrows and a wrinkled texture.

The ventral surface of the carpal is also mostly concave and this concave region is flanked on all sides by tall, thick margins (Fig. 21E). The surface is curved along its transverse axis, the proximal rim of the surface being convex while the distal rim is concave, the result being a crescentic outline. Again, the bone surface is marked with foramina and has a rugose texture.

If the interpretation proposed here is correct, E. lengi lacked fusion between distal carpals I and II on its probable left side at least (carpal fusion is known to be sometimes asymmetrical in coelurosaurs: see Zanno, 2006). The fact that the E. lengi holotype was seemingly not skeletally mature also raises the possibility that distal carpal II might have fused to distal carpal I at a later ontogenetic stage, forming a semilunate. While tyrannosaurids have what can be regarded as a “reduced” carpal skeleton, consisting only of flattened disc-like carpalia that lack trochleated surfaces (Carpenter & Smith, 2001), Holtz (1994) argued that this arose through simplification in the course of evolution from taxa with trochleated complex carpalia, and the data on E. lengi is consistent with this proposal. Our interpretation of the carpal given here–and what it might mean for the articulation and mobility of adjacent elements–should be considered preliminary and a more detailed study would be rewarding.

Metacarpals

Three probable metacarpals are known for E. lengi: the left metacarpal I (mc I; briefly described by Hutt et al., 2001, p. 233), and two bones interpreted here (see also Hutt et al., 2001, p. 233) as the proximal ends of mc II and III. Both proximal ends are preserved on a block between the axial neural arch and the cervical centrum; both are incomplete distally and would have been much longer when complete.

The left mc I is well preserved, 56 mm long and with an asymmetrical distal end (Figs. 22A–22F). The proximal articular surface is near-complete and damaged by erosion, lacking the corner of the ulnar side and a small area of bone from the radial side. It is subtriangular with the longest axis (25 mm wide) being obliquely inclined and extending from the medial edge of the articular surface to its laterodorsal part (Fig. 22F). This is approximately similar to the cross-sectional shape of the left metacarpal I of Deinonychus antirrhopus (Ostrom, 1969, fig. 63), except that the ventrolateral projection seen in that taxon is absent in E. lengi. The proximal end of metacarpal I of E. lengi exhibits a slightly concave 18 mm long facet for reception of metacarpal II on its lateral side (Fig. 22D). The dorsomedial surface of metacarpal I is also slightly concave, while the ventral surface is flat. The two distal condyles are markedly asymmetrical with the bulbous lateral condyle extending 10 mm further distally than the medial condyle (Fig. 22E). The medial condyle is strongly convex distally but is flat on its medial surface and lacks a collateral ligament fossa. It has a maximum dorsoventral height of 18 mm. A shallow U-shaped notch separates it from the lateral condyle. The latter is 20 mm tall, with a wide, strongly convex distal surface. An oval collateral ligament fossa is present on the lateral side of the lateral condyle and is c. 7 mm long proximodistally and c. 4 mm deep. The strong asymmetry of the metacarpal’s condyles shows that the pollex of E. lengi must have been strongly divergent. Mc I in E. lengi is highly similar to that of Deinonychus, the main difference being that the lateral condyle is more distally prominent in E. lengi. Tanycolagreus, Dilong and Guanlong are also similar with regard to the form of mc I (Xu et al., 2004, 2006; Carpenter, Miles & Cloward, 2005). In tyrannosaurids, the shaft is even more robust; furthermore, the distal articular end is less differentiated from the shaft, the condyles are less prominent and the intercondylar groove is shallower (Russell, 1970; Carpenter & Smith, 2001; Brochu, 2003; Holtz, 2004). Compared with non-coelurosaurs, the E. lengi mc I is longer and more gracile than mc I of Allosaurus and Acrocanthosaurus (Madsen, 1976; Currie & Carpenter, 2000; Chure, 2001), and it is also gracile relative to that of the compsognathid Sinosauropteryx (Currie & Chen, 2001).

The left mc II has a preserved length of 76 mm and includes the proximal articular end and part of the shaft (Fig. 22H). The proximal articular surface is broad (34 mm) and flat. Only one side of the bone is exposed and this surface is probably the ventral one because it is flat: the opposite side (visible only in cross-section) is convex. At its broken distal end, the shaft is sub-oval with the longest axis being the dorsoventral one: this is 18 mm tall. The shaft is 6 mm wide mediolaterally; its probable lateral side is flat while the probable medial side is weakly convex. The cross-sectional shape of the metacarpal shaft indicates that the broad proximal end was oriented mediolaterally. Such a large difference between proximal breadth and shaft width is seen in non-coelurosaurian theropods including allosauroids (Gilmore, 1920; Madsen, 1976; Currie & Carpenter, 2000) but appears less typical of coelurosaurs, in most of which the proximal end of mc II is similar in width to the shaft. This is also true in compsognathids and Scipionyx (Currie & Chen, 2001; Dal Sasso & Maganuco, 2011). However, Nqwebasaurus and therizinosauroids possess a proximally broad mc II (Barsbold, 1976; Russell & Dong, 1994; De Klerk et al., 2000; Clark, Maryańska & Barsbold, 2004) and this also appears to be the case in tyrannosaurids (Lambe, 1917; Russell, 1970; Carpenter & Smith, 2001; Brochu, 2003; Holtz, 2004).

A differently shaped but also incomplete element is preserved adjacent to the probable proximal end of mc II. It is also expanded at its proximal end with a preserved mediolateral width of 56 mm. In contrast to the putative mc II, the longest axis of the sub-oval cross-section of the broken shaft is parallel to that of the proximal expansion. This suggests that the expanded end may have been oriented dorsoventrally within the metacarpus, rather than mediolaterally. It is tentatively identified as mc III. If this element was oriented with its longest axis in cross-section aligned mediolaterally, the mc III of E. lengi would have been unusually broad compared to that of other tetanurans. The exposed surface of the bone is therefore probably medial (Fig. 22K). A prominent medial tubercle is located approximately 16 mm ventral to the dorsal edge and adjacent to the proximal articular surface (Fig. 22K). The bone dorsal to the tubercle is medially flat, whereas that ventral to the tubercle forms a concavity. When the metacarpus was articulated, the proximal end of mc II presumably fitted against one of these surfaces, and the tubercle may have helped prevent it from being displaced ventrally or dorsally.

What appears to represent a more distal part of the same bone (based its similar width) adheres to the shaft of a manual phalanx and is 45 mm long (Fig. 23M). In cross section it is hollow with bone walls c. 2 mm thick; it is taller (15 mm) than broad (10 mm). It is impossible to establish which end of the shaft is the proximal one. One end is more subcircular in cross-section than the other: the latter is taller and more compressed mediolaterally.

Figure 23 Manual phalanges of Eotyrannus lengi IWCMS: 1997.550.

The directional terms used here imagine the manus with its extensor surface oriented dorsally. (A) Possible left manual phalanx I-1 in medial view; (B) dorsal view; (C) distal view; (D) lateral view; (E) ventral view; (F) proximal view; (G) possible right phalanx II-1 in lateral view; (H) dorsal view; (I) proximal end in oblique view; (J) medial view; (K) ventral view; (L) proximal view; (M) possible right manual phalanx II-2 in medial view (the incomplete shaft of what is probably metacarpal III adheres to it surface); (N) lateral view; (O) oblique proximal view; (P) diagrammatic representation of possible right manual phalanx II-2 in medial view; (Q) diagrammatic representation of possible right manual phalanx II-2 in lateral view; (R) partial proximal end of left manual phalanx II-2 in lateral view; (S) view; (T) proximal view; (U) medial view; (V) ventral view; (W) proximal view; (X) left or right possible manual phalanx III-2 in lateral or medial view; (Y) dorsal view, distal end to left; (Z) oblique anterodorsal view; (A′) lateral or medial view (opposite side to that shown in (X)); (B′) ventral view; (C′) proximal view; (D′) possible left manual phalanx III-3 in medial view; (E′) dorsal view; (F′) distal view; (G′) lateral view; (H′) ventral view; (I′) proximal view; (J′) impression of ventral surface of unidentified manual phalanx as preserved on matrix, distal end to left; (K′) incomplete distal end of unidentified manual phalanx consisting only of articular condyles ad part of shaft in distal view; (L′) dorsal or ventral view, distal end toward top of page; (M′) dorsal or ventral view (though showing opposite surface to that shown in (L′)), distal end toward bottom of page; (N′) oblique distolateral view of possible right phalanx II-1 (also shown in (G)–(L)); (O′) possible left manual phalanx I-1 shown in articulation with left metacarpal I; (P′) metacarpals and phalanges arranged to show probable positions within the manus. Several phalanges are unknown (III-2, III-4).

Manual phalanges

Several manual phalanges of E. lengi are present, representing elements from both hands (Fig. 23). None are articulated, so their positions within the manus are inferred with varying degrees of certainty (Fig. 23P′). The identifications proposed here are based on the proportions of the phalanges relative to those of other taxa and to one another, and to the manner in which they sometimes articulate. Phalanges were provisionally identified as belonging to the left or right depending on the proportions of the distal condyles and shapes of the articular surfaces.

A complete, relatively robust phalanx 60 mm long is tentatively identified as the left I-1 (Figs. 23A–23F) on the basis of the relatively precise articulation it has with left metacarpal I (Fig. 23O′). When this phalanx and metacarpal I are articulated, the phalanx is directed medially relative to the long axis of the manus. However, the fact that this possible I-1 is only slightly longer than mc I renders this identification suspicious: it either indicates that E. lengi had an unusually proportioned pollex relative to other non-tyrannosaurid tyrannosauroids–in these taxa I-1 is much longer than mc I (Xu et al., 2004, 2006)–or the phalanx actually belongs to another digit. The proximal articular surface of this phalanx (25 mm wide, 22 mm tall) is biconcave with a weakly developed, vertically oriented central ridge (Fig. 23F). Bony rims surround the articular surface with the dorsolateral section of the rim exhibiting a shallow concavity. The shaft is deepest adjacent to the proximal articular surface. The proximal part of the ventral surface of the shaft is excavated by a shallow concavity at its proximal end and low convexities–probably weakly developed, paired flexor processes–flank this concavity medially and laterally. Distally, the left condyle is taller than the right condyle (20 mm vs 19 mm). This minor difference suggests that the phalanx is from the left manus. The distal end is 20 mm wide. The deep collateral ligament fossae are oval and are located high on the sides of the condyles; the fossa on the left side is slightly larger than that on the right. The larger of the two preserved unguals articulates well with this phalanx.

A relatively short, robust phalanx, 62 mm long, is identified as the right II-1 (though an identification as II-2 is also plausible) (Figs. 23G–23L, 23N′). The height and breadth of the proximal articular surface are similar (27 mm and 26 mm, respectively) and this surface is surrounded by a symmetrical bony rim. The collateral ligament fossae are asymmetrical, that on the left being deep and that on the right shallow. Similarly, the left distal articular condyle extends further distally than the right (Fig. 23N′). This phalanx articulates perfectly with the one identified as the right II-2.

A long, gracile manual phalanx (total length 85 mm) is preserved attached to the blade of the left scapula (Figs. 23M–23Q); the incomplete shaft of what is probably metacarpal III adheres to it surface. This is the phalanx suggested to be II-2 by Hutt et al. (2001, p. 233) and is extremely similar to the manual phalanx II-2 of Deinonychus (Ostrom, 1969, fig. 63). The proximal articular surface is biconcave and 20 mm tall. The surface is c. 13 mm wide at its dorsal end but broadens ventrally to c. 18 mm. Flaring rims are present on the lateral and medial sides. The articular rim on the right side is uniformly convex; that on the left possesses two successive convexities. The two halves of the articular surface are only weakly separated by a vertical ridge. The phalangeal shaft is straight and is taller than broad such that the cross section is a mediolaterally compressed oval. Viewed laterally or medially, the shaft is tallest adjacent to the proximal articular surface. The dorsal and ventral margins converge as they begin to pass distally but are subparallel at the shaft’s mid-length. The shaft then expands distally to form the prominent distal articular condyles. These are long (that on the left having a total ventral length of 25 mm) and strongly convex ventrally. They are separated by a deep intercondylar groove but, due to damage, it cannot be determined how far this extends dorsally. Unlike in some of the other manual phalanges, the ventral surface of the proximal end is flat. It cannot be determined with certainty whether this phalanx is from the left or right manus. The broken proximal end of a highly similar phalanx (Figs. 23R–23W), 29 mm long, is a near mirror-image of this element and is assumed to be II-2 from the opposite hand. The articular surface of this phalanx (24 mm wide, 27 mm tall) is biconcave with a convex dividing ridge. On the right side of the articular surface, the rim forms a prominent ‘shoulder’. The ventral surface has a shallow transverse concavity flanked by two low convexities.

Hutt et al. (2001, p. 233) described a manual phalanx 70 mm long and suggested that it belonged to either digit II or III. This specimen is probably III-3 (Figs. 23D′–23I′); both a ventrally located convexity on the right side of the rim surrounding the proximal articular surface and the fact that the right distal ligament fossa is deeper and larger than the left suggests that it is from the left manus. It is robust with a broad proximal end and prominent distal condyles. The biconcave proximal articular surface is 26 mm tall and 28 mm broad. The ventral surface is flat with, again, low convexities flanking its proximal region. Both ligament fossae are elliptical, with that on the left being more dorsally located. The right fossa is 10 mm long and c. 4 mm tall; the shallower left fossa is c. 5 mm long.

An additional phalanx, 60 mm long, is intermediate in robustness between those identified as II-1 and II-2 (Fig. 23X, 23C′). The distal condyles are essentially missing, although their ventralmost portions are preserved. These show that the condyles were small, each with a poorly developed distal convexity, and that the intercondylar groove was shallow. Across the distal end the specimen is 15 mm wide. The proximal articular surface is subtriangular and (in contrast to some of the other phalanges) wider than tall (25 mm vs 23 mm). Unlike in all other E. lengi manual phalanges, the proximal surface is not biconcave. An undivided proximal articular surface is seen elsewhere in phalanx III-1 of certain allosauroids (Gilmore, 1920, p. 62; Currie & Carpenter, 2000, p. 230). Given that a longer, more robust phalanx described above is inferred to be phalanx III-3 it is unlikely but not impossible that the present phalanx represents III-1. There does not seem to be a way of determining whether this phalanx is from the left or right manus.

Some additional non-ungual phalanges are preserved but their positions are difficult to infer. The distal 40 mm of a phalanx, comprising the articular condyles and a fragmentary part of the shaft (Figs. 23K′–23M′), cannot be identified with confidence: it is extremely similar to the distal end of the manual phalanx identified as the right II-1 or II-2 but may belong to the foot. The right condyle has a height of 16 mm and a width of c. 8 mm and appears to extend further distally (by c. 3 mm) than does the left; the left condyle is damaged, with some of the ventral surface missing. The right collateral ligament fossa is circular (4 mm high and 4 mm long) and deep. The right condyle flares laterally in one direction, which is presumably ventral.

Two manual unguals are known for E. lengi, and both were briefly described by Hutt et al. (2001, pp. 233-234). The first is large, 85 mm along its proximodistal axis and 103 mm along its curved dorsal margin (Figs. 24A–24D). The distal tip is missing and conceivably added another 10–15 mm to this curve. The proximal articular surface is deep (26 mm), narrow (15 mm) and surrounded by a bony rim. The flexor tubercle is well developed, bulbous and most prominently convex on the ventral surface of the ungual’s base (Figs. 24A–24C). The tubercle is not continuous with the articular facet. Distally, the tubercle is not distinct from the curved portion of the ungual but grades into it along the ungual’s ventral margin. Where the tubercle is most pronounced, the ungual has a maximum dorsoventral depth of 42 mm. The ventral surface of the rest of the ungual is transversely convex and does not form a keel or ridge, though the part of the ventral edge just distal to the flexor tubercle is somewhat mediolaterally compressed and keel-like (Fig. 24C). The ungual’s dorsal margin is continuously convex. The sides are flattened but become more convex toward the distal tip. The claw grooves are symmetrically positioned but that on the left is deeper and better developed than that on the right. It is approximately 3 mm deep at its proximal end but shallows to 1 mm at the ungual’s tip. Near the ungual tip, the bone ventral to the groove (on both sides) forms a lamina that overlaps the ventral margin of the groove (Figs. 24A, 24B). Hutt et al. (2001, p. 233) identified this ungual as from the pollex, but there seems to be no way of knowing whether it is from the left or right side.

Figure 24 Manual unguals of Eotyrannus lengi IWCMS: 1997.550.

(A) Possible pollex ungual in lateral or medial view; (B) same element from opposite side; (C) ventral view; (D) dorsal view; (E) possible digit II ungual in ventral view; (F) lateral or medial view; (G) same element from opposite side; (H) proximal view; (I) oblique dorsoproximal view of proximal end of possible digit II ungual from same side as shown in (F); (J) ventral view. clgr claw groove, dcon dorsoproximal concavity, dli dorsal lip, dol dorsal lamina, flt flexor tubercle.

The second manual ungual is very slightly smaller (Figs. 24E–24J) and was regarded by Hutt et al. (2001, p. 233) as belonging to digit II. Its proximal end is cracked and somewhat crushed. As with the pollex ungual, there does not seem to be a way of determining whether it belonged to the left or right manus. Its ‘straight line’ preserved length is 84 mm while the length along its dorsal curve is 97 mm. Again, 10–15 mm of the distal tip is missing. Overall, it is highly similar to the larger ungual but is somewhat straighter, has a less bulbous flexor tubercle, and has a more prominent lateral groove. A similar degree of variation is present between unguals I and II in Guanlong (Xu et al., 2006). The proximal articular surface is tall (26 mm), narrow (14 mm) and approximately symmetrical. This surface is weakly biconcave with a poorly developed vertical ridge separating the two halves. A proximodorsal lip overhangs the articular surface and, in contrast to the larger ungual, there is a concavity between the convex dorsal surface of the ungual and the rim (Figs. 24F, 24G). In some maniraptorans, the manual unguals of digit II and III possess a proximodorsal lip while the pollex ungual lacks it (Senter et al., 2004); E. lengi is somewhat maniraptoran-like in this respect. Where the flexor tubercle is most prominent, the maximum dorsoventral depth of the ungual is 38 mm. Half-way along its length it is 22 mm deep and the distal tip is 7 mm deep. The left and right claw grooves are at the same dorsoventral level but the groove on the right side is less defined. The right claw groove is 4 mm deep proximally. It becomes narrower toward the ungual’s tip where its depth is c. 1.5 mm deep. Again, the bone ventral to the grooves possesses dorsal laminae that overlap the ventral margins of the grooves.

E. lengi clearly had a gracile manus similar to that of Guanlong, Dilong, Tanycolagreus and maniraptorans like Deinonychus (Ostrom, 1969; Gishlick, 2001; Xu et al., 2004; Carpenter, Miles & Cloward, 2005) (Fig. 23P′). However, several of the phalanges–if correctly identified–are more robust than their equivalents in these taxa, and in this respect are intermediate between a maniraptoran– or Dilong–like manus and a tyrannosaurid-type manus. This is clear from the proportions of the phalanx identified as the left I-1. This element is superficially like the I-1 of Allosaurus and tyrannosaurids in shape (Madsen, 1976; Brochu, 2003) and substantially less gracile than that of Dilong, Tanycolagreus and Deinonychus (Ostrom, 1969; Gishlick, 2001; Xu et al., 2004; Carpenter, Miles & Cloward, 2005); it is probably proportionally shorter than that of Guanlong as well (Xu et al., 2006). The elongate left II-2 possesses a proximal articular surface almost identical to that seen in Deinonychus (Ostrom, 1969, fig. 63). The proximal surface of II-2 differs from that of Guanlong in that the dorsal part of the proximal articular surface in Guanlong is concave on both its lateral and medial sides and is almost twice as wide ventrally as dorsally (Xu et al., 2006, supp info, fig, 2j). A more E. lengi-like morphology appears to be present in Tanycolagreus (Carpenter, Miles & Cloward, 2005). The E. lengi phalanx suggested to be a right II-1 also resembles the equivalent bone in Deinonychus: the proximal articular surface is relatively broad, the phalangeal shaft is deep proximally but shallow adjacent to the distal condyles, and the condyles are only slightly asymmetrical. The same features characterize II-1 in allosauroids (Madsen, 1976; Currie & Carpenter, 2000), but in allosauroids the phalanx is more robust.

If the phalanx suggested to be a III-3 is correctly identified, E. lengi had a proportionally shorter, more robust digit III than Deinonychus and other maniraptorans (Ostrom, 1969; Osmólska, Currie & Barsbold, 2004), Dilong (Xu et al., 2004) Tanycolagreus (Carpenter, Miles & Cloward, 2005) and probably Guanlong (Xu et al., 2006), and was instead more like allosauroids in the form of this digit (Madsen, 1976; Currie & Carpenter, 2000). The distal end of III-3 would almost certainly have articulated with an ungual and there is no indication that the third manual digit was partially reduced.

The two manual unguals appear less strongly curved than those of many maniraptorans (Ostrom, 1969; Osmólska, Currie & Barsbold, 2004) but this might be because their distal tips are missing. The subtle development of a proximodorsal lip and concavity in E. lengi is interesting: this structure is typical of maniraptorans and does not generally appear elsewhere in non-maniraptoran Coelurosauria, including in Tyrannosauridae and apparently Guanlong and Dilong (Lambe, 1917; Carpenter & Smith, 2001; Brochu, 2003; Holtz, 2004; Xu et al., 2004, 2006). However, a dorsoproximal concavity (albeit not associated with a distinct lip) appears present in the pollex ungual of Tanycolagreus where it appears longer that it is in E. lengi (Carpenter, Miles & Cloward, 2005). The presence of a dorsoproximal lip and concavity is therefore considered a possible autapomorphy for E. lengi.

Ilium

A segment of left ilium representing the region dorsal and posterodorsal to the acetabulum as well as part of the pubic peduncle is known for E. lengi and was referred to in passing by Hutt et al. (2001, pp. 228, 236). The segment is preserved as two pieces, the larger of which is embedded within a block of plaster. Placed together, the two form an irregularly shaped sheet of bone 137 mm long which is deeper anteriorly (122 mm) than posteriorly (where breakage means that it is reduced to a depth of 8 mm at its posteriormost tip). The presence of a prominent, vertically oriented ridge in the middle of the sheet shows that the exposed side is the lateral one (Fig 25). Anteroventral to the section preserving the ridge, a descending, mediolaterally narrow strip of bone appears to represent part of the pubic peduncle: it is too narrow to be a partial ischial peduncle (Brusatte & Benson, 2013) and provides further confirmation that the preserved section is from the animal’s left side. What seems to be the true dorsal margin of the ilium is preserved in the posterior part of the fragment, where it forms a narrow ridge between 2 and 4 mm wide. It has a straight dorsal margin (Fig. 25A). However, this section of preserved margin is so short that it cannot be considered representative of the dorsal margin in its entirety: when complete, the ilium’s dorsal margin may have been either dorsally arched–as it is in Stokesosaurus, Juratyrant and Tyrannosauridae (Benson, 2008; Brusatte & Benson, 2013)–or dorsally straighter, as it is in Aviatyrannis and Guanlong (Rauhut, 2003a; Xu et al., 2006). The more ventral parts of the blade are thicker than the dorsal margin (9 mm ventral to the posterior end and 12 mm anteroventrally). The flat form of the preserved dorsal part of the ilium may indicate that the ilia were not dorsomedially inclined and hence not in contact across the dorsal midline.

Figure 25 Segment of Eotyrannus lengi IWCMS: 1997.550 left ilium representing the region dorsal and posterodorsal to the acetabulum.

(A) Lateral view; (B) ventral view; (C) lateral view with median ridge outlined for clarity. dm dorsal margin, frd flat region dorsal to median ridge, mr median ridge.

A robust ridge projects from the body of the blade’s lateral surface. The ridge is not perpendicular to the segment’s dorsal margin but, rather, inclined posterodorsally at an angle of about 20° relative to the vertical (Figs. 25A, 25C). This posterodorsal inclination has previously been considered autapomorphic of Juratyrant (Benson, 2008) and is distinct from the vertical attitude of the median ridge seen in Aviatyrannis, Guanlong and tyrannosaurids (Rauhut, 2003a; Xu et al., 2006; Benson, 2008). Outside of Tyrannosauroidea, a posterodorsally inclined ridge above the acetabulum is present in Siamotyrannus and Iliosuchus (Buffetaut, Suteethorn & Tong, 1996). The ridge merges into the body of the blade and terminates more than 30 mm ventral to the bone’s dorsal margin, leaving a region of flat, featureless bone between the dorsal end of the ridge and the bone’s dorsal margin. In most other tyrannosauroids, the ridge extends further dorsally, terminating close to the bone’s dorsal margin. Juratyrant is an exception and exhibits the same condition as Eotyrannus (Benson, 2008; Brusatte & Benson, 2013). Juratyrant possesses two additional distinctive (autapomorphic) features of the ilium: a narrow preacetabular notch and an iliac body with a strongly arched, semioval outline (Benson, 2008). It remains unknown whether these were also present in Eotyrannus.

Ventrally, the ridge protrudes beyond the preserved margin of the bone as a blunt-tipped, finger-like process. In ventral view, the ridge forms a robust triangle, 25 mm across and 25 mm tall. The broken ventral end, irregular bone texture across the ventral margin of the whole segment, and lack of a supra-acetacular shelf show that the segment does not preserve the acetabular border but instead represents a region somewhat dorsal to it (Fig. 25B).

The two ilium fragments were discovered in close association with the tibia. A thin, plate-like bone still embedded in the same block as the tibia likely represents more of the iliac blade. Only its cross-section, which is 120 mm long (probably representing part of the dorsoventral height of the iliac blade) and 3–6 mm thick mediolaterally, is visible.

Tibia

Virtually the whole length of the left tibia is known, though it is broken into fragments that were not preserved in close association (Figs. 26, 27). A proximal section c. 360 mm long was preserved on the same block as metatarsal IV and the left humerus (Fig. 26). A distal section, c. 210 mm long, was not discovered in close association with the proximal section but the two fit together at various points of contact and their shafts are similar in width and cross-sectional shape. Placed together, they form a tibia c. 570 mm long (Fig. 27D). Much of the tibial shaft is fractured and mediolaterally compressed.

Figure 26 Incomplete left tibia and fibula of Eotyrannus lengi IWCMS: 1997.550.

(A) Left tibia in posteromedial view; (B) section of left tibia showing fibular crest in medial view; (C) left tibia in proximal end, anterior toward bottom of page; (D) proximal part of left tibia in anterior view; (E) shaft of left tibia at broken proximal end; (F) shaft of left tibia at broken distal end; (G) broken distal end of left tibia and fibula (same segment as shown in (F)) in lateral view. cncr cnemial crest, fcr fibular crest, fi fibula, tfor tibial foramen, lco lateral condyle, mco medial condyle.

Figure 27 Left tibia of Eotyrannus lengi IWCMS: 1997.550.

(A) Distal end of left tibia in anterior view; (B) posterior view; (C) oblique anterodistal view; (D) complete tibia with all segments placed in their approximate original positions. fma flat medial area; lach lateral channel; lash lateral shoulder; mm medial malleolus.

The proximal articular surface is c. 70 mm long anteroposteriorly, 35 mm wide and with a cnemial crest that curves laterally, as is typical for coelurosaurs (Figs. 26C, 26D). The cnemial crest is simple, convex proximally, poorly developed, and with no trace of an accessory ridge on its lateral side. It grades distally into the anterior margin of the shaft and does not project with a squared-off profile as do the cnemial crests of Tanycolagreus (Carpenter, Miles & Cloward, 2005), Juratyrant (Benson, 2008), tyrannosaurids (Brochu, 2003) and numerous other theropods. Notably, the cnemial crest in Guanlong does not appear squared-off but is a subtriangular projection that grades distally into the shaft (Xu et al., 2006). In Eotyrannus, the apex of the crest is eroded; nevertheless, this lack of a prominent squared-off profile does appear natural. The lateral surface of the shaft laterodistal to the cnemial crest is concave due to post-mortem compaction. Neither proximal condyle is prominent along the posterior edge of the proximal surface and the intercondylar groove is shallow and poorly defined (Fig. 26C); this might also be due to erosion and damage. There is no indication of an anterolateral projection on the lateral condyle. The posteromedial edge of the proximal end is higher than the lateral edge, so the articular surface faces somewhat laterally in this region. Overall, the proximal articular surface is morphologically simpler than is typical for theropods, most of which exhibit a prominent cnemial crest that curves laterally to a marked degree, well defined proximal articular condyles, and a distinct posterior intercondylar groove. It is assumed that the cnemial crest and articular condyles were more prominent and more sharply defined in their original condition.

The tibial shaft tapers along its length, that part distal to the fibular crest being notably narrower (50 mm thick) than the proximal region between the articular surface and fibular crest (where the shaft is 80 mm thick). The fibular crest is a D-shaped flange, 80 mm long, that begins 90 mm distal to the margin of the proximal end: it is thus distinctly separate from the proximal articular surface (Figs. 26A, 26B). The fibular crest is similar in size and position to that of other coelurosaurs (Ostrom, 1969; Carpenter, Miles & Cloward, 2005) and is especially similar to that of Juratyrant (Benson, 2008). The crest is robust with the approximate shape of a broad V in cross-section. A large foramen (c. 7 mm long distoproximally and 3 mm in width) is located posteromedial to the distal 20 mm of the crest (Figs. 26A, 26B). The tibial foramen is located adjacent to the distal part of the crest as it is in Juratyrant (Benson, 2008). Distally, the shaft becomes less compressed mediolaterally, taking on a circular cross-section. At about mid-length the shaft is 50 mm long anteroposteriorly and at most 33 mm wide, but at its major break it is c. 36 × 36 mm (Figs. 26E, 26F). Internally, the bone is composed of tubular, shell-like layers that decrease in thickness toward the middle of the bone (it is likely that the boundaries between these layers correspond to histological features, like lines of arrested growth. We hope to see histological analysis carried out on E. lengi in future).

The distal portion of the tibia preserves a subcircular section of shaft and the anteroposteriorly flattened distal-most region with its facets for the astragalus and calcaneum (Figs. 27A–27C). This segment (preserved separately from the rest of the tibia) is peculiar and was suggested by Hutt et al. (2001) to be the incomplete radius of an additional theropod taxon. However, the proximal end of the shaft is almost identical in proportions to the distal end of the other section; the two are identical in colour and style of preservation and fit together well.

The anterior surface of the distal end consists of three structures, described here in order of position from medial to lateral. The distomedial section of the surface is occupied by a large flat facet, the lateral and medial edges of which are slightly convex, meaning that the facet as a whole projects somewhat relative to the remainder of the anterior face of the bone’s distal end. Occupying the middle of the anterior surface, adjacent to this facet on its lateral side, is a poorly defined concavity shaped somewhat like an inverted U. It does not extend as far proximally as the facet. Finally, the distolateral part of the bone possesses a distinct projecting ‘shoulder’ along its lateral edge that merges into the shaft proximally. Distal to this projection, it appears as if the distolateral corner of the bone has been broken away. This appears likely based on the shape of the distal tibia in other coelurosaurs (Rauhut & Xu, 2005; Benson, 2008). Raised rims form the proximal and medial borders to this broken section; the raised medial border separates it from the midline concavity. Proximal to the distolateral shoulder-like structure, a distoproximally aligned channel runs parallel to the shaft’s lateral border (Figs. 27A, 27C). We are not aware of any similar channel being reported for any other theropod taxon and thus regard this character as an autapomorphy of E. lengi.

A similar distal tibial configuration was illustrated for the coelurosaur Tugulusaurus faciles (Rauhut & Xu, 2005). On its posterior surface, the distal end of the E. lengi tibia is mostly taken up by a concave area that is bordered laterally by a thick, distoproximally aligned ridge. The distal end has a maximum width of 64 mm.

Fibula

The incomplete shaft of the left fibula of E. lengi is preserved in two pieces, both of which are attached to the middle section of the left tibia’s shaft (Figs. 26F, 26G). A proximal segment 134 mm long is beneath the tibia’s posterior surface while a more distal segment, 35 mm long, is preserved subparallel to the tibial shaft. Hutt et al. (2001, p. 236) described the fibula as an “elongate, slender element in which the proximal third is expanded craniocaudally”. The preserved proximal end terminates well short of the original proximal end and, contra Hutt et al. (2001, p. 236), has the same cross-sectional dimensions as the preserved distal end (Fig. 26F). At both ends, the shaft has an anteroposterior length of 15 mm and a maximum width of 7 mm. It is mediolaterally compressed, convex on its lateral side and slightly concave medially. This fibular cross-sectional shape is typical for tetanurans (Osmólska, Roniewicz & Barsbold, 1972; Madsen, 1976; Osmólska, 1996; Charig & Milner, 1997; Carpenter, Miles & Cloward, 2005), though the fibula of Deinonychus is described as being nearly circular in cross-section (Ostrom, 1969) and that of tyrannosaurids has been described as D-shaped in cross-section (Brochu, 2003, p. 115). The more distal fragment does not include the true distal end of the fibula. These fragments are extremely gracile relative to the tibial shaft and suggest proportions similar to those known for other early tyrannosauroids (Carpenter, Miles & Cloward, 2005; Xu et al., 2004, 2006). It is assumed that the shaft tapered continually from its broad proximal end towards its narrower distal part but this cannot be confirmed: a distinct condition, where the fibula narrows markedly distal to the insertion point of the m. iliofibularis tendon, is present in Bagaraatan and maniraptorans but not in other theropods (Rauhut, 2003b). It is also assumed–based on the condition in other non-tyrannosaurid tyrannosauroids (Carpenter, Miles & Cloward, 2005; Xu et al., 2006)–that the fibula reached the proximal tarsals.

Metatarsals

Sections of metatarsals II, III and IV are known for E. lengi (Figs. 28, 29) and show that it had a gracile metatarsus, as expected for a tyrannosauroid. The proximal ends of mt II and IV show that E. lengi was not arctometatarsalian, in contrast to Appalachiosaurus and Tyrannosauridae but like Guanlong and Dilong (Holtz, 2004; Xu et al., 2004, 2006; Carr, Williamson & Schwimmer, 2005). The distal ends of the metatarsals are not ginglymoid, as they are in some maniraptoran taxa (Ostrom, 1969; Norell & Makovicky, 1997; Rauhut, 2003b), and the deep and prominent collateral ligament fossae are typical for tetanurans.

Figure 28 Metatarsal II of Eotyrannus lengi IWCMS: 1997.550.

(A) Distal end of left metatarsal II in lateral view; (B) medial view; (C) anterior view; (D) posterior view; (E) distal view; (F) proximal view at broken end; (G) distal end of right metatarsal II in anterior view; (H) right metatarsal II in anterior view; (I) lateral view; (J) medial view; (K) posterior view; (L) distal view; (M) proximal view. antc anterodistal concavity; latc lateral condyle, medc medial condyle, mtfIII facet for mt III.

Figure 29 Metatarsals of Eotyrannus lengi IWCMS: 1997.550.

(A) Distal end of left metatarsal III in anterior view; (B) medial view; (C) lateral view; (D) posterior view; (E) distal view; (F) proximal end of left metatarsal IV in oblique medial view; (G) lateral view; (H) medial view; (I) proximal view, posterior surface toward top of page; (J) distal end of left metatarsal IV in posterior view; (K) known metatarsal elements of E. lengi arranged in approximate in-life configuration to mimic appearance of a left metatarsus, though with right metatarsal II flipped to appear like a left and metatarsal IV shown in posterior aspect as the anterior surface is not available. extfo extensor fossa, latc lateral condyle, medc medial condyle, mtfIII facet for mt III.

The right mt II of E. lengi was figured and described by Hutt et al. (2001, p. 236, fig. 4A) and the distal end of left mt II is known as well (consisting only of the condylar region and the distalmost part of the shaft). The more complete right mt II is elongate and gracile with a total length of 253 mm (Figs. 28H–28K). The shaft is broken in several places and the proximal 100 mm is slightly artificially rotated so that the anterior surface faces somewhat laterally. The shaft is straight and the distal end is not deflected medially. Viewed anteriorly or posteriorly, the lateral and medial margins of the shaft are subparallel. With the exception of the lateral articular facet for mt III, most surfaces of the shaft are convex, though the proximal and distal parts of the anterior surface of the shaft are flattened. The proximal articular surface of the metatarsal is semicircular with a convex medial surface and flat lateral surface (Fig. 28M). This morphology is typical for tetanurans (Gilmore, 1920, fig. 51; Ostrom, 1969, fig. 70; Currie & Zhao, 1994, fig. 26C; Currie & Carpenter, 2000, fig. 14A) and differs from the more complex shape present in arctometatarsalian tyrannosauroids (Brochu, 2003, fig. 103; Carr, Williamson & Schwimmer, 2005, fig. 19F). The anteroposterior length of the proximal articular surface (44 mm) exceeds that of the shaft (c. 25 mm) so it is accurate to describe the proximal end as expanded relative to the shaft. The proximal articular end has a maximum width of 23 mm. The flat lateral facet for the articulation of mt III extends distally for approximately 70 mm from the proximal articular end. More distally, the lateral surface of the shaft becomes convex, although a low distoproximal ridge extends along the anterolateral surface and indicates distal continuation of the articular area. Because of the slight distortion of the proximal part of the shaft, in life the facet for mt III was probably directed laterally rather than anterolaterally as preserved.

No distinct facet for mt I could be detected. If mt I was present it was–based on the relative position of the mt I facet in other tyrannosauroids (Brochu, 2003; Carpenter, Miles & Cloward, 2005)–presumably located approximately 100 mm proximal to the distal articular end of the bone. The scar for the insertion of M. gastrocnemius, often mistaken for the mt I facet (Tarsitano, 1983; Carrano & Hutchinson, 2002), could not be detected either.

The distal end is wider than the shaft because the bone surrounding the collateral ligament fossae flares medially and laterally, giving the posterior surface of the distal end a width of 40 mm. The distal end appears to form a single condyle when viewed anteriorly (Figs. 28G, 28H) but is in fact bilobed, comprising a bulbous, more prominent lateral condyle (20 mm wide) that is separated from a smaller medial condyle (c. 9 mm wide) by a shallow intercondylar canal 13 mm wide (Fig. 28L). The medial condyle is only complete in the left element and is a prominent subrectangular eminence with an anteromedial inclination (Fig. 28E). A similar distal metatarsal II morphology is seen in allosauroids (Madsen, 1976, plate 54; Currie & Zhao, 1994, fig. 27), Appalachiosaurus (Carr, Williamson & Schwimmer, 2005, fig. 19) and tyrannosaurids (Brochu, 2003, fig. 103). Both condyles are restricted to the posterior part of the distal surface of the bone and the lateral condyle extends 11 mm further distally than the medial condyle. Both collateral ligament fossae are well defined and deep, with the lateral one being larger (c. 11 × 13 mm) and more distally located than the medial fossa (c. 9 × 11 mm).

The incomplete left mt II has a preserved length of 45 mm and is 40 mm wide across the condyles. Breakage of the subcircular shaft shows that the bone was hollow as far distally as the condyles (Fig. 28F). The bone walls are 3–5 mm thick. As in the right mt II, the lateral ligament fossa (c. 10 × 13 mm) is larger and deeper than the medial fossa (c. 7 × 5 mm).

The distal end of what is almost certainly the left mt III of E. lengi is known but this fragment consists only of the distal 116 mm (Figs. 29A–29E). It was not mentioned by Hutt et al. (2001). Even allowing for crushing at the preserved proximal end, the shaft is compressed anteroposteriorly and subrectangular in cross-section. The inferred anterior surface of the shaft is smoothly convex while the inferred posterior surface is flat. The distal articular end is broader than the shaft, being 30 mm wide across the posterior surface. Viewed medially or laterally, the distal end is symmetrical. However, a shallow extensor fossa just proximal to the articular end on one side identifies the surface concerned as the anterior one (Fig. 29A), a deduction supported by the fact that this inferred anterior surface is narrower (26 mm) than the inferred posterior surface. As expected for mt III, the distal end is block-like and not differentiated into separate condyles (Fig. 29E). However, one side of the distal end is anteroposteriorly deeper than the other (33 mm vs c. 29 mm), suggesting that it is the medial side. Accordingly, the specimen is here identified as belonging to the left pes. Both collateral ligament fossae are prominent and subcircular; the right and left fossae have dimensions of 12 × 12 mm and 15 × 15 mm, respectively. The distal end of what appears to be the right mt III, consisting of the distal condyle and the adjacent part of the shaft, is preserved within matrix. The shaft is subrectangular in cross-section, having a width of 24 mm and a maximum anteroposterior length of 20 mm. One side, possibly the posterior one, is flat, while the medial and lateral surfaces are convex. The visible collateral ligament fossa is large and circular, measuring 14 × 14 mm. These dimensions are similar to those of the ligament fossae of the left mt III.

A near-complete left mt IV, broken into two pieces, is known for E. lengi and was stated by Hutt et al. (2001, p. 236) to be 260 mm long. Again, this element is long and gracile (Figs. 29F–29J). Most of the bone is embedded within a block and only its posterior surface is visible. The proximal 96 mm is free of matrix and largely complete. When the two pieces are united the total length is more like 280 mm, but this is probably exaggerated by breakage and distortion. The metatarsal was discovered immediately beneath the tibia. The proximal end is complex (Figs. 29F–29I). Although the posterior face of the metatarsal shaft is flat, it is overhung by the proximal articular surface, especially medially. Proximally, the articular surface is 38 mm wide across the posterior face of the bone, the bone narrowing in width to c. 20 mm distally. A ridge demarcates the proximal 50 mm of the posterior surface of the shaft from the convex lateral side. A similar ridge also demarcates the proximal part of the posterior surface from the medial surface. Anteromedial to this ridge, the proximal end of the medial surface is convex, but passes distally into a deep concavity that would have been directed anteromedially. This concavity extends 30 mm distally down the shaft, is c. 20 mm wide proximally, and is for reception of the proximal end of mt III (Figs. 29F, 29H, 29I). Similar well-developed facets for mt III are absent in most theropods but one was figured for Sinraptor dongi (Currie & Zhao, 1994, fig. 26A). In arctometatarsalian tyrannosauroids like Appalachiosaurus (Carr, Williamson & Schwimmer, 2005, fig. 19D) and tyrannosaurids (Brochu, 2003, fig. 103) the facet is shorter anteroposteriorly, shaped more like a ‘U’ in proximal view, and located closer to the posterior surface of the shaft. The proximal end of mt IV is also blockier and more robust in these taxa.

Few details of the distal end can be discerned but, in contrast to tyrannosaurids, the distal end is not laterally deflected relative to the shaft’s long axis (Fig. 29J). The distal articular surface is 35 mm wide and bilobed, with the two halves of the condyle restricted to the posterior surface of the distal end and separated by a 9 mm wide intercondylar groove. Accurate measurements of the two halves of the condyle cannot be made but the medial part appears to have been distally bulbous and c. 35 mm long anteroposteriorly. Any collateral ligament fossae are obscured by immoveable matrix.

Pedal phalanges

Six pedal phalanges, one of which is an ungual, are known for E. lengi (Figs. 30, 31). Essentially, they appear typical for a tetanuran that is intermediate in size and proportions between small and giant taxa.

Figure 30 Pedal phalanges of Eotyrannus lengi IWCMS: 1997.550.

(A) Probable right II-1 in dorsal view; (B) lateral view; (C) distal view; (D) ventral view; (E) probable right IV-3 or IV-4 in lateral view; (F) dorsal view; (G) distal view; (H) medial view; (I) ventral view; (J) proximal view; (K) pedal phalanx of undetermined identity (preserved on same block as left humerus) in dorsal view; (L) lateral or medial view; (M) incomplete pedal ungual in lateral or medial view. clgr claw groove, prdp proximodorsal process, pdr proximal dividing ridge.

Figure 31 Left or right pedal phalanx III-1 of Eotyrannus lengi IWCMS: 1997.550.

(A) Oblique dorsolateral or dorsomedial view; (B) distal view; (C) proximal view; (D) oblique dorsal view to show dorsal concavity at distal end; (E) dorsal view; (F) lateral or medial view; (G) lateral or medial view, opposite side of (F); (H) ventral view. extfo extensor fossa.

A pedal phalanx 85 mm long is preserved on the same block as the blade of the left scapula (Figs. 30A–30D), but the left side of the phalanx cannot be examined and much of the left condyle is absent. The proximal articular surface, likewise, cannot be examined but the shaft adjacent to the articular surface is 34 mm tall. The shaft is shallowest just proximal to the distal condyles, where it is only 16 mm deep. The proximoventral part of the shaft is flattened and low ridges mark the boundaries between the ventral surface and the sides of the shaft. Dorsally, a deep concavity is proximal to the distal condyles. Both condyles of this phalanx are more extensive dorsally and ventrally than the condyles of other preserved pedal phalanges: in turn, the right condyle on this phalanx is more prominent than the left. The right condyle is 27 mm tall and extends c. 6 mm dorsal to the adjacent part of the shaft. The intercondylar groove is shallow but, on the dorsal surface, extends as far proximally as do the articular surfaces of both condyles. The ventral part of the right condyle is angled to the right but this may be the result of deformation. The right collateral ligament fossa is rounded and taller than it is long (8 × 6 mm). The large size of this phalanx suggests that it is II-1. If the “right-ward” inclination of the right distal condyle is a genuine feature, this phalanx is probably from the right pes.

The largest preserved phalanx of E. lengi is 94 mm long (Fig. 31) and was suggested by Hutt et al. (2001, p. 236) to be phalanx III-1. It is broken at mid-length, distorted dorsoventrally and lengthened by matrix that has infilled the break. Hutt et al. (2001, p. 236) estimated the original length of the phalanx to be 87 mm. Its proximal articular surface is concave and subcircular, 37 mm wide and 35 mm tall, and has a rugose articular surface and bony rim (Fig. 31C). The proximal articular surface is not biconcave, supporting its identification as the most proximal phalanx of the digit. In lateral view, the shaft is deepest (34 mm) proximally and shallowest (17 mm) just proximal to the distal condyles. The proximal part of the ventral surface of the shaft is flattened and flanked by two low convexities, both of which are more prominent on this phalanx than on any other. They mark the boundaries between the sides of the phalangeal shaft and its ventral surface. A deep extensor fossa is present on the dorsal side of the shaft (Fig. 31D). The part of the ventral surface of the shaft adjacent to the condyles is flat. The condyles themselves are poorly expressed on the dorsal and ventral surfaces and the intercondylar groove is shallow. However, the collateral ligament fossae are large, deep and well rounded, with the left one being more elliptical (Fig. 31F). The right fossa measures c. 10 × 10 mm, and that on the left is 12 mm long and 7 mm tall. The maximum width across the distal condyles is 33 mm. There is no reliable way of determining whether this III-1 belongs to the left or right foot.

The smallest preserved pedal phalanx is 45 mm long (Figs. 30E–30J). This bone appears too broad and robust to be a manual phalanx; it is assumed to belong to E. lengi due to its similarity to the other pedal phalanges of this taxon but the possibility remains that it belongs to the associated dryosaurid. This is the ‘small, isolated phalanx’ discussed by Hutt et al. (2001, p. 236), who suggested that it was IV-3 or IV-4. Given its length compared to those of the inferred pedal phalanges III-1 and II-1, this could be correct. Because the left distal condyle is deeper than the right condyle, the small phalanx is regarded as belonging to the right foot. The proximal articular surface is broader than tall (26 × 23 mm) and biconcave, its two concave areas separated by a low vertical ridge. The proximodorsal process dorsal to the ridge is well developed, extending further proximally than the lateral and medial bony rims that surround the articular surface. In lateral or medial view, the ventral surface of the shaft is concave, the shaft being only 14 mm tall at its shallowest point but 22 mm deep adjacent to the proximal articular surface. The shaft is convex dorsally, laterally and medially, though flat to slightly concave on its ventral surface. Viewed dorsally, the shaft narrows slightly to 21 mm at mid-shaft. The distal condyles are not extensive either ventrally or dorsally, and the intercondylar groove is shallow. A shallow extensor fossa is present on the dorsal surface, just proximal to the distal condyles. The right condyle extends slightly further distally than the left. The collateral ligament fossae are rounded and deep, but not as deep as those on the other pedal phalanges. The phalanx is 26 mm wide across the distal condyles.

A pedal ungual (Fig. 30M) is preserved on the same block as the left humerus, a pedal phalanx (Figs. 30K, 30L) of undetermined identity and other fragments. The ungual was suggested by Hutt et al. (2001, p. 236) to pertain to digit IV but it is not possible to determine whether it belongs to the left or right foot. The ungual is only exposed in medial (if it is from the left pes) or lateral (if it is from the right pes) view, and its maximum length is c. 60 mm. The distalmost 20 mm or so appears to be missing. The distal part of the preserved length of the ungual curves to the left, but a vertical break separating this distal part from the rest of the bone suggests that this represents post-mortem distortion. The bone is 26 mm deep proximally, and tapers gradually toward its tip. A shallow concavity is present near the proximal end of the dorsal surface: this is unusual within Tetanurae but has been reported for Appalachiosaurus within Tyrannosauroidea (Carr, Williamson & Schwimmer, 2005). No flexor tubercle is present, although some of the bone surface on the proximal part of the ventral surface is striated. A shallow lateral or medial groove is present c. 5 mm from the ventral edge of the ungual’s lateral surface.

Revised diagnosis of Eotyrannus lengi

E. lengi exhibits several unique morphological features and thus is diagnosable. Reevaluation shows that most of the supposedly distinctive features mentioned in the preliminary description of E. lengi are not diagnostic, and the original diagnosis is here critiqued. Hutt et al. (2001, p. 229) provided the following diagnosis of E. lengi (individual features are numbered for ease of reference below):

Tyrannosauroid coelurosaurian theropod with [1] serrated carinae on D-shaped premaxillary teeth. [2] Maxillary and dentary teeth with apically complete denticulation; [3] rostral carinae bear denticles for less than half the length of the denticle-bearing part of the caudal carinae. [4] Denticle size difference index of c. 1.5. [5] Anterior portion of maxilla laterally flattened with anterior border to the antorbital fossa sharply defined, [6] ventral edge of maxilla straight. [7] Coracoid with prominent mediolaterally-wide, subcircular glenoid directed caudally. [8] Humerus with large internal cavity situated dorsally (anconally) with several smaller cavities situated ventrally. [9] Manus proportionally long (digit II c. 95% humerus length) with [10] three well-developed metacarpals. [11] Carpals not reduced to simple elements as in tyrannosaurids.

The new information on E. lengi presented here substantially updates our understanding of the morphology of this species (Figs. 32–34), and a huge amount of new information on the morphology and diversity of tyrannosauroids in general has become available since Hutt et al. (2001) was published (e.g. Xu et al., 2004, 2006; Carr, Williamson & Schwimmer, 2005; Benson, 2008; Averianov, Krasnolutskii & Ivantsov, 2010; Li et al., 2009; Brusatte et al., 2010b, 2016; Nesbitt et al., 2019; Zanno et al., 2019). Accordingly, the above diagnosis can now be replaced. On the numbered points made in the diagnosis of Hutt et al. (2001) the following points can now be made:

Figure 32 Cranial reconstruction of Eotyrannus lengi IWCMS: 1997.550.

Known cranial elements reconstructed in assumed life position, excluding isolated teeth and possible vomer. Broken and missing areas mean that the nature of many articulations are unknown. Some elements (like premaxilla) reversed from right side. antf antorbital fenestra, dent dentary, extn external nostril, jug jugal, max maxilla, nas nasal, pal palatine, pmax premaxilla, quad quadrate, sura surangular.

Figure 33 Skeletal reconstruction of Eotyrannus lengi IWCMS: 1997.550.

New skeletal reconstruction of Eotyrannus lengi, depicting only those elements preserved in the holotype. The positions shown for some of the isolated vertebrae and ribs are conjectural. Scale bar: 100 cm.

Figure 34 Skeletal reconstruction, with extrapolation of entire skeleton.

New skeletal reconstruction of Eotyrannus lengi IWCMS: 1997.550 depicting extrapolated appearance of entire skeleton (with estimated soft tissue outline). The shapes and proportions of those elements unknown from Eotyrannus lengi are based on those of other non-tyrannosaurid tyrannosauroids. Image by Dan Folkes. Scale bar: 100 cm.

The presence of “serrated carinae on D-shaped premaxillary teeth” is problematic. Firstly, describing the premaxillary teeth of E. lengi as D-shaped is misleading since they are better described as “U-shaped” in cross section (Hendrickx, Mateus & Araújo, 2015), as described above, and are not unique among tyrannosauroids in this respect. Secondly, the presence of serrations on premaxillary teeth is not unique either; in fact, this condition is present in most other tyrannosauroid taxa (Currie, Rigby & Sloan, 1990; Holtz, 2004).

Apically complete denticulation is not rare or unusual in Theropoda and is widespread across the group, including within coelurosaurs (Sereno & Brusatte, 2008; Brusatte, Benson & Hutt, 2008; Brusatte et al., 2010b). Within Tyrannosauroidea, it is certainly not unique to E. lengi (Holtz, 2004; Brusatte, Carr & Norell, 2012).

The condition of having rostral carinae (= mesial carinae) that bear denticles for less than half the length of the denticle-bearing part of the caudal carinae (= distal carinae) probably would be diagnostic for E. lengi, were it present. Restudy failed to identify it and E. lengi seems to be much like other tetanurans in the distribution of denticles on its lateral teeth (Currie, Rigby & Sloan, 1990).

The DSDI of E. lengi is not c. 1.5 but rather 1.16 (with 1.21 reported by Sweetman (2004)). This lower figure is comparable to those obtained for many other tyrannosauroids and thus cannot be regarded as diagnostic for E. lengi.

Neither the presence of a laterally flattened anterior region on the maxilla nor a pronounced rim to the antorbital fossa are unique to E. lengi–both features are widespread in Tetanurae and Tyrannosauroidea (e.g. Currie & Dong, 2001; Hwang et al., 2004; Xu et al., 2004; Dal Sasso & Maganuco, 2011).

The presence of a straight ventral edge on the maxilla is not unique to E. lengi, being present in Dilong, Suskityrannus and other coelurosaur taxa. Furthermore, this condition is clearly plesiomorphic for Coelurosauria and normal for non-tyrannosauroid coelurosaurs (e.g. Hwang et al., 2004; Holtz, Molnar & Currie, 2004; Dal Sasso & Maganuco, 2011): E. lengi thus retains a primitive condition that distinguishes it from tyrannosaurids and their closest relatives.

The morphology of the coracoid part of the glenoid in E. lengi is not diagnostic and is similar to that seen in other tyrannosauroids (Xu et al., 2004) and non-tyrannosauroid tetanurans (e.g. Currie & Zhao, 1994).

The use of internal cavities within the humerus as part of the diagnosis of E. lengi seems unwise as internal structures such as these often cannot be observed across a wide range of taxa. Furthermore, the internal cavities in the humerus of E. lengi do not seem to differ from those present in other theropod humeri.

E. lengi does appear to have a proportionally long manus, with digit II measuring c. 95% the length of the humerus. However, this condition seemingly represents the plesiomorphic state for Tyrannosauroidea: the humeral fragments figured for Dilong suggest that its hand was as proportionally elongate as that of E. lengi relative to humerus length (Xu et al., 2004). Furthermore, both Tanycolagreus and Guanlong possess a manual digit II whose length exceeds 95% of that of the humerus (Carpenter, Miles & Cloward, 2005; Xu et al., 2006).

The presence of three metacarpals is obviously the plesiomorphic state for Tyrannosauroidea. Actually, the presence of at least three metacarpals is primitive, since Guanlong possesses four (Xu et al., 2006).

Similarly, the presence of a distal carpal with a trochlear articular surface in E. lengi represents the plesiomorphic state for Tyrannosauroidea.

In conclusion, the 11 purportedly diagnostic features proposed for E. lengi by Hutt et al. (2001) can all be rejected as potentially diagnostic for E. lengi since they are either plesiomorphic for Tyrannosauroidea, shared with at least some other tyrannosauroid taxa, or not truly present in E. lengi. It is now clear, however, that E. lengi possesses a number of unique characters that allow an emended diagnosis to be formulated.

Emended diagnosis

The following unique suite of features are as yet unknown in other tyrannosauroids or in those coelurosaurian lineages close to Tyrannosauroidea and are hence regarded as probable autapomorphies of Eotyrannus lengi: lateral surface of dentary bearing five shallow arcuate furrows that extend anterodorsally from a common origin on the ventral part of the bone; large, block–like humeral entepicondyle; distal end of tibia with distoproximally aligned channel, demarcated laterally by a low ridge, located close to the lateral border of the shaft.

Five other characters may represent additional autapomorphies of E. lengi, but their status remains uncertain. The first of these is the presence of a concave notch and accompanying anteromedial tooth-like projection on the anterodorsal part of the dentary. This feature is ambiguous as a potential autapomorphy, however, since its poor preservation means that it might have been misinterpreted. The second potential autapomorphy is the presence of a sinuous ridge that extends across the base of the vomeropterygoid process of the palatine: the bone dorsal to this ridge is inset or embayed relative to the ventral part. This character is also difficult to evaluate given our poor knowledge of palatine anatomy in non-tyrannosaurid tyrannosauroids and more data are needed before it can be evaluated further. The third potential autapomorphy also pertains to the palatine and concerns the long, straight dorsal margin present between the vomeropterygoid and pterygoid processes: this contrasts with the shorter, dorsally concave edge present in other tyrannosauroids (Currie, 2003; Carr, Williamson & Schwimmer, 2005; Xu et al., 2006; Brusatte, Carr & Norell, 2012). Again, however, a lack of data from other taxa prevents us from being more confident about use of this configuration as an autapomorphy.

The fourth potential autapomorphy is the apparent tear-drop-shaped cross-sections of the shafts of the radius and ulna. However, identification of the relevant partial bone shafts as a radius and ulna is uncertain, so more information is needed before their cross-sectional geometry can be considered diagnostic.

Finally, one other character can be considered a potential autapomorphy since, while not unique to E. lengi relative to all other theropod taxa, it is unique within Tyrannosauroidea. As described here, E. lengi possesses a proximodorsal lip and adjacent concavity on at least one of its manual unguals. These structures are a familiar feature of oviraptorosaurs and some other maniraptorans but are, excepting E. lengi, unknown in Tyrannosauroidea (Lambe, 1917; Carpenter & Smith, 2001; Brochu, 2003; Holtz, 2004; Xu et al., 2004, 2006). As discussed above, what may be a subtly developed proximodorsal lip and adjacent concavity has been figured for the pollex of the possible tyrannosauroid Tanycolagreus (Carpenter, Miles & Cloward, 2005).

Comments on other Wealden Supergroup theropods

Numerous theropod specimens, most recently reviewed by Naish (2011), have been reported from the Wessex Formation (Fig. 35) and the possibility that at least some might represent additional E. lengi specimens was kept in mind throughout our research on this dinosaur. Some taxa can be removed from consideration immediately. Baryonychine spinosaurids are represented in the Wessex Formation by teeth and an isolated dorsal vertebra (Buffetaut, 2010; Naish, 2011) (Figs. 35M–35P), elements that differ greatly in morphology from their counterparts in tyrannosauroids. The carcharodontosaurian allosauroid Neovenator salerii (Brusatte, Benson & Hutt, 2008), known from the excellent holotype and several referred specimens, is osteologically well known and clearly has no close affinity with E. lengi. Benson et al. (2009) described an additional large, as yet unnamed Wessex Formation theropod, presently known only from the distal end of the femur, the dorsal end of the left pubis, and the pubic boot and adjacent parts of the pubic shafts (listed together as MIWG 6350). The presence of an extensor groove on the femur and a slit-shaped pubic fenestra shows that MIWG 6350 is a tetanuran, but the additional presence of a proportionally broad pubic boot excludes the specimen from Coelurosauria. It cannot, therefore, be considered referable to E. lengi. Numerous smaller, and often very poorly known, theropods have also been recovered from the Wessex Formation. As noted by Hutt et al. (2001), and as explained in full here, it does not seem that any of these can be considered conspecific with E. lengi.

Figure 35 Other Wessex Formation theropods.

Montage showing selection of Lower Cretaceous theropod elements described from the Wessex Formation of the Isle of Wight. (A) Holotype partial cervical vertebra of Thecocoelurus daviesi (NHMUK R181) in left lateral view; (B) right lateral view; (C) ventral view; (D) anterior view; (E) holotype sacrum of Aristosuchus pusillus (NHMUK R178) in left lateral view; (F) holotype pubes with pubic boot of Aristosuchus pusillus (NHMUK R178) in left lateral view; (G) pubic boot in ventral view; (H) one of the two holotype cervical vertebrae of Calamosaurus foxi (NHMUK R901) in anterior view; (I) posterior view; (J) right lateral view; (K) dorsal view; (L) ventral view; (M) isolated dorsal vertebra of Baryonyx cf. walkeri (UOP C001.2004) in anterior view; (N) left lateral view; (O) right lateral view; (P) posterior view; (Q) holotype sacrum of Ornithodesmus cluniculus (NHMUK R187) in dorsal view; (R) right lateral view; (S) ventral view; (T) the so-called “Calamosaurus tibia” NHMUK R186 in anterior view; (U) posterior view; (V) isolated left coelurosaur femur MIWG 6124 in lateral view; (W) anterior view; (X) medial view; (Y) isolated left coelurosaur tibia MIWG 5137 in medial view; (Z) posterior view; (A′) lateral view; (B′) anterior view. cncr cnemial crest, dp diapophysis, epi epipophysis, fcr fibular crest, ftr fourth trochanter, hyp hyposphene, ligfo ligament fossa, lco lateral condyle, mco medial condyle, mm medial malleolus, mp metapophyses, par parapophysis, pefo pedicular fossa, pnfos pneumatic fossa, spdl spinodiapophyseal lamina, vlg ventrolateral groove, vs ventral sulcus. e–g modified from Owen (19,876), m–p by Steve Hutt, q–s modified from Howse & Milner (1993).

The first Wealden theropod to be named was Calamospondylus oweni Fox in Anon (1866), said by Fox (in Anon, 1866) to consist of “five cemented vertebrae with the sacral ribs and portions of the other iliac bones”. The current location of the holotype is unknown, so the only source of information on this specimen is the brief, semi-popular publication in which it was first described (Naish, 2002). However, C. oweni is a nomen dubium because its describer (Fox in Anon, 1866) failed to provide diagnostic features for the taxon (Naish, 2002). The small size and possible vertebral pneumaticity of C. oweni suggest that it was a coelurosaur but it cannot be directly compared with E. lengi in the absence of both the C. oweni holotype and any reported diagnostic features (Naish, 2002).

Aristosuchus pusillus (Owen, 1876) is based on a sacrum and partial pelvis NHMUK R178 (Figs. 35E–35G) that have been suggested to belong to a compsognathid (Naish, Hutt & Martill, 2001; Naish, Martill & Frey, 2004). More recently, a tyrannosauroid identification has been considered plausible (Naish, 2011) since A. pusillus strongly resembles the possible tyrannosauroid Mirischia asymmetrica. The latter possesses an anterodorsal concavity on the anterior margin of the ilium and an anterodorsally concave margin on the pubic peduncle (Naish, Martill & Frey, 2004) and hence is tyrannosauroid-like. However, both characters are also present in some non-tyrannosauroids (Rauhut, 2003a, 2003b; Dal Sasso & Maganuco, 2011). We presently, therefore, interpret these characters as tyrannoraptoran symplesiomorphies. Whether A. pusillus is a tyrannosauroid or not, the overlapping material known for A. pusillus and E. lengi (sacral vertebrae) reveals profound differences. The posterior-most sacral vertebrae of A. pusillus are fused together, indicating that the holotype was closer to skeletal maturity than was the holotype of E. lengi. However, the A. pusillus sacrum is c. 120 mm long, suggesting a total length of c. 2 m, whereas the subadult holotype of E. lengi represents an animal c. 4.5 m in length. The sacral vertebrae of A. pusillus differ from those of E. lengi in being ventrally rounded rather than bearing ventral keels.

Ornithodesmus cluniculus Seeley, 1887 was named for six fused sacral vertebrae (NHMUK R178) (Figs. 35Q–35S). It has been given a variety of phylogenetically disparate suggested identities but seems most likely to represent a dromaeosaurid (Norell & Makovicky, 1997; Naish, 2011). O. cluniculus recalls E. lengi in possessing lateral foramina on its sacral centra. However, while the openings present in E. lengi are likely sacral nerve foramina, those present in O. cluniculus are smaller and located lower on the centra, and hence appear to be pneumatic. The sacral fusion present in O. cluniculus indicates skeletal maturity. With a sacrum length of 96 mm (suggesting a total length of approximately 1.5 m), this apparent adult would have been a far smaller animal than the subadult holotype of E. lengi. In addition, O. cluniculus possesses a ventral sulcus that extends continuously along the ventral surfaces of the second to sixth sacral vertebrae and the ventral surfaces of its sacral centra are flattened (Howse & Milner, 1993). In E. lengi, no ventral sulcus is present and the ventral surface of the sacral centrum is keeled.

A partial cervical vertebra from the Wessex Formation (NHMUK R181) was named Thecocoelurus daviesi (Seeley, 1888) (Figs. 35A–35D). Similarities between this specimen and the cervical vertebrae of both oviraptorosaurs and abelisauroids have been noted (Naish, 2011). A lack of extensive cervical material of E. lengi makes detailed comparison with T. daviesi difficult. However, the two taxa differ in that the single known cervical vertebra of T. daviesi possesses an oval pneumatic fossa on the side of the centrum, a deep interspinous ligament pit, ventrolateral ridges and a ventral sulcus, none of which are present in the known cervical vertebrae of E. lengi.

Calamosaurus foxi is also based on cervical material, in this case the two articulating vertebrae NHMUK R901 (Lydekker, 1889) (Figs. 35H–35L). Based on their small size and strong opisthocoely these were previously referred to Compsognathidae (Naish, Hutt & Martill, 2001) but they are similar in shape and proportion to those of Dilong and hence may also be from a small tyrannosauroid (Naish, 2011). Because the neurocentral sutures in C. foxi are closed (though not fused), despite the fact that each vertebra is only 40 mm long, it seems unlikely that they could represent the same taxon as E. lengi. The posterolaterally flaring postzygapophyses in E. lengi differ from the shorter, less flaring ones in C. foxi and, while the more complete C. foxi vertebra possesses a short neural spine, the one cervical neural spine known for E. lengi extends for much of the centrum’s length. However, these differences could reflect positioning within the cervical series. It is possible that C. foxi and E. lengi might be synonymous but there is no good evidence to support this.

Several isolated hindlimb and pelvic elements from the Wessex Formation have been referred to Calamosaurus and Aristosuchus (Lydekker, 1891; Galton, 1973; Naish, Hutt & Martill, 2001; Naish, 2002). The tibia NHMUK R186, long known as the “Calamosaurus tibia” (Figs. 35T, 35U), has an unusually prominent medial malleolus that projects medially as a distinct flange (Naish, 2011). No such structure is present in E. lengi and NHMUK R186 most likely represents a different non-maniraptoran coelurosaur. An additional small tibia (MIWG 5137) (Figs. 35Y, 35B′) differs from E. lengi in possessing well separated proximal condyles, and also lacks the distinctive distal tibial morphology of E. lengi. Two small femora (NHMUK R5194 and MIWG 6214) (Figs. 35V–35X) from the Wessex Formation (Galton, 1973; Naish, 2000) likely belong to non-maniraptoran coelurosaurs but cannot be identified more precisely and do not overlap with any E. lengi material. Finally, the partial ischium NHMUK R6426 (Naish, 2002) also does not overlap with any E. lengi material, does not possess any tyrannosauroid characters, and cannot be identified more precisely than Tetanurae indet. It should be noted that all of these specimens belong to animals substantially smaller than the E. lengi holotype.

Most of the small Wessex Formation theropods are too poorly known to allow confident identification but they seemingly include one or more non-maniraptoran coelurosaurs, such as compsognathids or small tyrannosauroids (e.g., Calamosaurus, Aristosuchus), and maniraptorans (Ornithodesmus, isolated teeth described by Sweetman (2004)) (Naish, 2011). None of the material reported for these taxa is congeneric with E. lengi, meaning that this taxon is currently represented only by its holotype. In additional to the enigmatic smaller theropods, E. lengi lived alongside a large, non-coelurosaurian tetanuran (Benson et al., 2009), baryonychine spinosaurids (Charig & Milner, 1997; Naish, 2011) and the carcharodontosaurian Neovenator (Brusatte, Benson & Hutt, 2008).

Phylogenetic analysis

In order to test the phylogenetic affinities of Eotyrannus, we incorporated it into a phylogenetic analysis of Theropoda that focuses on non-maniraptoran coelurosaurs and non-coelurosaurian tetanurans (see Appendix 1 and 2 for character list and sources for coding, and Appendix 3 for data matrix). We compiled a data matrix describing the distribution of 1,145 phylogenetically informative morphological characters in 83 ingroup neotheropods and 3 non-neotheropod saurischian outgroup taxa. Eoraptor was chosen to root the tree. The data matrix was analysed with the Hennig Society version of TNT (Goloboff, Farris & Nixon, 2008). The phylogenetic analysis protocol consisted of a heuristic search using the ‘New Technology’ settings of TNT (Goloboff, Farris & Nixon, 2003): driven search, 100 addition sequences; using sectorial searches and tree fusing. The resulting most parsimonious trees (MPTs) from this first search round were then submitted to an additional round of tree bisection and reconnection (TBR) branch swapping to ensure a thorough sampling of tree space. Exploration of character optimization was performed using TNT. Bremer Support (BS, Bremer, 1994) for nodes was calculated by saving 10,000 suboptimal topologies up to 10 steps longer than the MPTs in TNT. The analysis recovered 12 shortest trees of 4,349 steps each, with a Consistency Index and Retention Index of 0.2752 and 0.5230 respectively, the strict consensus of which is shown in Fig. 35. The analysis supports the monophyly of Coelophysoidea (including ‘dilophosaurs’, Tykoski & Rowe, 2004, BS = +4), Averostra (sensu Ezcurra & Novas, 2007, BS = +3), Ceratosauria (sensu Rauhut, 2003b, BS = +2) and Tetanurae (BS = +2). Within Tetanurae, the bizarre Chilesaurus (Novas et al., 2015) was recovered as outside a clade that includes all other tetanurans, and Xuanhanosaurus and Zuolong are of undetermined position and have been depicted within a polytomy that also involves Neotetanurae (BS = +2). Megalosauroidea (Benson, Carrano & Brusatte, 2010, BS = +2) is recovered as the sister-taxon to Allosauroidea ((BS = +3); Rauhut, 2003b). Coelurosauria (sensu Gauthier, 1986, BS = +3) includes Compsognathidae as its earliest-diverging lineage, in addition to Tyrannoraptora (BS = +2), the latter including Tyrannosauroidea (BS = +2), and the lineage including Ornitholestes, Aorun and maniraptoriforms.

In the strict consensus of the shortest trees (Fig. 36), Tyrannosauroidea includes a pectinate series of early-diverging lineages leading to Tyrannosauridae. The Juratyrant + Stokesosaurus clade (BS = +2) is found to be outside the clade that contains all remaining tyrannosauroids. Coeluridae, including Coelurus, Tanycolagreus and Tugulusaurus, is recovered as the sister-group of remaining tyrannosauroids. The nodal support values among these early-diverging tyrannosauroids are weak, mainly due to the inclusion of fragmentary taxa like Stokesosaurus and Tugulusaurus. Among those members of Tyrannosauroidea whose evolution post-dates the divergence of coelurids, Proceratosauridae, including Dilong, forms the earliest-diverging branch. Yutyrannus is recovered as sister-taxon to the clade containing Eotyrannus and remaining tyrannosauroids (BS = +2). The latter subclade includes Xiongguanlong as sister-taxon of the clade that includes Dryptosaurus and arctometatarsalian tyrannosauroids (including Tyrannosauridae; BS = +2), in addition to megaraptorans (Benson, Carrano & Brusatte, 2010; BS = +3). The relationships among megaraptorans are well resolved but weak, mainly due to the inclusion of fragmentary taxa like Chilantaisaurus, Orkoraptor and Siats, the latter recovered as outside the clade that includes remaining megaraptorans. The enigmatic South American Aniksosaurus (Martínez & Novas, 2006) is recovered as a megaraptoran. Chilantaisaurus and Fukuiraptor are recovered as successively closer to Megaraptoridae (Novas et al., 2013). The topology among arctometatarsalian tyrannosauroids places Appalachiosaurus, Bistahieversor and Teratophoneus outside Tyrannosauridae (the ‘Gorgosaurus + Tyrannosaurus’ node in our ingroup; BS = +2).

Figure 36 Strict consensus topology.

Strict consensus topology of the shortest trees found by the analysis (tree length: 4,349 steps; CI = 0.2752; RI = 0.5230). Numbers at nodes are Bremer Support values.

The analysis found no support for a close relationship between the two European tyrannosauroids, Eotyrannus and Juratyrant, previously discussed by Brusatte et al. (2010b) and recovered by Brusatte & Carr (2016), Zanno et al. (2019) and Nesbitt et al. (2019). Such disagreement is probably an artifact of the different taxon sampling between the two analyses (e.g., megaraptorans are not included in the dataset of Brusatte & Carr, 2016). The two European taxa are distinct among theropods in possessing a posterodorsally inclined supracetabular ridge that fails to reach the dorsal margin of the ilium; our finding that they are not close relatives indicates that this specific anatomical configuration evolved independently. With regard to the position of Eotyrannus specifically, its inclusion within Tyrannosauroidea is supported by the lack of a prominent keel on the ventral surface of the cervical centra (char. 207.0) and the presence on the lateral surface of the ilium of a vertical crest dorsal to the acetabulum (char. 382.1). Within Tyrannosauroidea, Eotyrannus is recovered as a member of the “Coeluridae + remaining tyrannosauroids” clade on the basis of a fibular crest on the tibia that does not extend proximally to the level of the proximal end of the bone (char. 909.0), as a member of the “Proceratosauridae + remaining tyrannosauroids” clade on the basis of the nasal pneumatic recesses (char. 47.1), the medially fused nasals (char. 874.1), the distinct dorsal expansion on its scapula (char. 896.0) and a medial condyle on the humerus that is larger than the lateral condyle (char. 1,164.1), and as a member of the “Yutyrannus + remaining tyrannosauroids” clade on the basis of a maxilla that lacks a lateral ridge ventral to the antorbital fossa (char. 24.0) and manual ungual I being longer than its preceding phalanx (char. 309.1). Finally, Eotyrannus is recovered as closer to Tyrannosauridae than Yutyrannus, Dilong and other proceratosaurids because it lacks both a distinct median nasal crest (char. 45.0) and a deep lateral groove on the dentary (char. 178.0), possesses premaxillary teeth whose longest axis is labiolingually aligned (char. 793.1), bears a deep surangular shelf (char. 1,570.1), lacks a distinct extensor sulcus on the second metatarsal (char. 481.0) and possesses a transversely compressed fourth metatarsal (char. 560.0). Eotyrannus lacks several synapomorphies of tyrannosaurids and tyrannosaurid-like tyrannosauroids, including the absence of nasal participation in the antorbital fossa, paired nasal crests, an enlarged quadrate foramen, an acute anterodorsal corner on the dentary (as seen in lateral view), an enlarged posterior surangular foramen, shortened cervical neural arches, and posterior dorsal pleurocoels.

The tyrannosauroid affinities of megaraptorans–first suggested by Novas et al. (2013) and subsequently supported by Porfiri et al. (2014) and discussed by Bell et al. (2015)–are here confirmed using the largest morphological dataset and a wider taxon sample among non-coelurosaurian tetanurans and coelurosaurs than employed in previous analyses. We do not, however, support Porfiri et al.’s (2014) inclusion of Eotyrannus within Megaraptora: they described how this position was supported by the presence of (1) strongly opisthocoelous cervical centra and (2) pleurocoels in dorsal vertebrae in this taxon; this is an error, since Eotyrannus possesses amphicoelous or weakly opisthocoelous cervical centra and lacks pleurocoels (pneumatic foramina) in its dorsal vertebrae. The hypothesis that megaraptorans are tyrannosauroids seemingly explains why certain controversial Lower Cretaceous specimens from Australia have been identified as possible tyrannosauroids by some (Benson et al., 2010) but linked to megaraptorans by others (Herne, Nair & Salisbury, 2010). Furthermore, our analysis confirms a megaraptoran affinity for both Chilantaisaurus and Siats (Benson, Carrano & Brusatte, 2010; Zanno & Makovicky, 2013; Bell et al., 2015). This phylogenetic model indicates that Cretaceous tyrannosauroids were a more successful and diverse clade than previously suggested. Large-bodied ‘mid-Cretaceous’ forms like Chilantaisaurus and Siats–previously placed among non-coelurosaurian tetanurans–are now interpreted as a ‘second wave’ of tyrannosauroid gigantism that evolved later than the Early Cretaceous taxa Sinotyrannus and Yutyrannus (Brusatte & Carr, 2016), but still prior to the emergence of Tyrannosauridae. Furthermore, the ‘mid-Cretaceous’ megaraptoran radiation fills a stratigraphically and morphologically significant gap present in tyrannosauroid evolution between the Jurassic-Early Cretaceous early-diverging tyrannosauroids (e.g., proceratosaurids) and the Late Cretaceous tyrannosaurids. It is noteworthy that our results also support the suggestion of Brusatte et al. (2016) that a grade of mid-Cretaceous, mid-sized, longirostine tyrannosauroids (including Xiongguanlong, and according to our study, megaraptorans too) were ancestral to the advanced large-bodied tyrannosaurids.

By placing our phylogeny on a stratigraphic timescale (Fig. 37), we speculatively infer that tyrannosauroids are primitively Eurasian (Bell et al., 2015), with eastern Asia perhaps being more important in their evolution during the Jurassic and Early Cretaceous than Europe or North America, though they also occurred in these regions (Madsen, 1974; Foster & Chure, 2000; Rauhut, 2003a; Benson, 2008). Most of the younger and most anatomically modified lineages within Tyrannosauroidea are North American, including the early-diverging megaraptoran Siats, indicating invasion of that region after the divergence of Xiongguanlong (Fig. 37). A novel result of our analysis is that megaraptorans underwent a global radiation during the ‘middle’ Cretaceous, including large-bodied forms in Laurasia (Zanno & Makovicky, 2013) and gracile-limbed species in Gondwana (Bell et al., 2015). Hardly anything is known about the Dryptosaurus + Tyrannosauridae clade prior to the Campanian. An intriguing issue is whether the late radiation of the tyrannosaurid-like forms in Laurasia was delayed by the megaraptoran radiation (see Zanno & Makovicky (2013) for a discussion of early-diverging lineages within the megaraptoran radiation, therein interpreted as carcharodontosaurian allosauroids). Furthermore, while tyrannosauroids are now known from the Middle and Upper Jurassic (Rauhut, 2003a; Xu et al., 2006; Benson, 2008; Averianov, Krasnolutskii & Ivantsov, 2010; Rauhut, Milner & Moore-Fay, 2010), comparatively few early representatives of the group have been discovered. Close relatives of E. lengi likely await discovery in the Upper Jurassic, Berriasian, Valanginian and Hauterivian strata of Eurasia.

Figure 37 Stratigraphically calibrated phylogeny of Tyrannosauroidea.

Geochronologic units modified from Carr & Williamson (2010). The black bars represent the stratigraphic range for the taxon when known with precision; the grey bars represent the possible stratigraphic ranges for taxa whose age is not well resolved. Stratigraphic abbreviations: AA, Aalenian; AL, Albian; AP, Aptian; BA, Barremian; BAJ, Bajocian; BAT, Bathonian; BE, Berriasian; CA, Carnian; CAL, Callovian; CA, Campanian: CE, Cenomanian; CO, Coniacian; HA, Hauterivian; HE, Hettangian; J/K, Jurassic-Cretaceous boundary; KI, Kimmeridgian; MA, Maastrichtian; NO, Norian; OX, Oxfordian; PL, Pliensbachian; RH, Rhaetian; SA, Santonian; SI, Sinemurian; TI, Tithonian: TO, Toarcian; Tr/J, Triassic-Jurassic boundary; TU, Turonian; VA, Valanginian.

Conclusions

Eotyrannus lengi Hutt et al., 2001 is a valid tyrannosauroid taxon from the Barremian Wessex Formation of the Isle of Wight, presently known only from the holotype (IWCMS: 1997.550). Substantial cranial material, cervical, dorsal and sacral vertebrae, the scapulocoracoid and much of the forelimb and hindlimb are known. These allow us to characterise E. lengi as a mid-sized, long-handed tyrannosauroid with a tyrannosaurid-like scapulocoracoid and elongate, gracile distal hindlimbs (its femur remains unknown). Thickened, pneumatic, fused nasals, a premaxilla with a steep anterior border, a tyrannosaurid-like quadrate and premaxillary teeth that are U-shaped in cross-section show that E. lengi was tyrannosaurid-like in cranial morphology. The relatively short preantorbital ramus of the maxilla shows that E. lengi was not longirostrine as are such tyrannosauroids as Guanlong, Dilong, Xiongguanlong and Alioramus (Xu et al., 2004, 2006; Li et al., 2009; Brusatte, Carr & Norell, 2012). Longirostry is also present in at least some megaraptorans (Porfiri et al., 2014). Given that longirostrine taxa do not form a clade and are surrounded in their phylogenetic placement by non-longirostrine taxa (like Yutyrannus and Eotyrannus; the Proceratosaurus holotype is non-longirostrine but its late juvenile or subadult status (Rauhut, Milner & Moore-Fay, 2010) means that its adult condition is unknown), it would appear that this condition evolved separately within Tyrannosauroidea on three or more occasions and was not a primitive trait of the clade. This indicates that tyrannosauroids were not consistently and perpetually specialised for robust snouts optimised for powerful biting across their history, nor that all lineages within the group were part of a pattern of ‘correlated progression’ like that inferred for tyrannosaurids by Snively, Henderson & Phillips (2006). Instead, it appears that specialisation toward lifestyles not dependent on the presence of a more robust snout occurred several times. It is possible that this was, in cases, an expression of heterochronic processes (Carr, 1999; Li et al., 2009) but further consideration of this idea is beyond the scope of the present study.

The diagnostic characters of E. lengi include the presence of peculiar curving furrows on the lateral surface of the dentary, a large, block–like humeral entepicondyle and a tibia with a distoproximally aligned, laterally positioned channel on the distal end. The femur, pubis and ischium remain unknown and virtually nothing is known of the caudal skeleton.

While several theropods have been named from the Wessex Formation, none can be shown to be synonymous with E. lengi. The fragmentary nature of the holotypes of most of these taxa renders their affinities uncertain, but E. lengi was contemporaneous with baryonychine spinosaurids, carcharodontosaurian allosauroids, probable compsognathids and maniraptorans (Naish, Hutt & Martill, 2001; Sweetman, 2004; Benson et al., 2009, 2010; Naish, 2011). Like the majority of other early-diverging tyrannosauroids (Brusatte et al., 2010b), E. lengi was a mid-sized predator in a fauna whose dominant large predators were megalosauroids or allosauroids.

Our study confirms Hutt et al.’s (2001) proposal that E. lengi is a non-tyrannosaurid tyrannosauroid. Of several Jurassic and Early Cretaceous tyrannosauroids described since Hutt et al. (2001) was published, E. lengi seems to be among those most closely related to Tyrannosauridae. Our phylogenetic analysis recovers a topology broadly consistent–bar its suggested placement for Megaraptora–with other analyses of Tyrannosauroidea (e.g., Senter, 2007, 2010; Li et al., 2009; Brusatte et al., 2010a, 2010b, 2016; Brusatte & Benson, 2013; Loewen et al., 2013; Brusatte & Carr, 2016; Zanno et al., 2019; Nesbitt et al., 2019), with Proceratosauridae, Yutyrannus, Eotyrannus and Xiongguanlong being successively closer to the tyrannosauroid clade that includes Dryptosaurus, Appalachiosaurus, Bistahieversor and Tyrannosauridae. We support a tyrannosauroid identity for megaraptorans and suggest that they are an important ‘mid-Cretaceous’ clade that represent a second wave of large-bodied tyrannosauroids, the diversification of which may even have slowed the radiation of the tyrannosaurid lineage. E. lengi shares two characters of the ilium with Juratyrant (posterodorsally inclined vertical ridge, failure of vertical ridge to reach dorsal margin of ilium) (Benson, 2008), but we did not recover a close relationship between these two taxa.

Supplemental Information

Supplemental Information 1 OTUs and references.

List of taxa used in our analysis with a list of the references consulted for character information.

Click here for additional data file.

Supplemental Information 2 Informative character statements list.

List of the 1145 phylogenetically informative characters included in the data set. Character statement numeration follows the complete list, from Lee et al. (2014) and Cau et al. (2015).

Click here for additional data file.

Supplemental Information 3 Nexus file.

Character codings used in our analysis.

Click here for additional data file.

This work results primarily from the first author’s PhD thesis, completed at the University of Portsmouth (UK). Thanks are due to Dave M. Martill for his assistance with that project, Bob Loveridge for help with photography and SEM techniques, and Dave Hughes and Mike Barker (both formerly of SEES, University of Portsmouth) for co-operation and support. Stig Walsh, Steve Sweetman, Tony Butcher, Dean Bullen and Sarah Fielding are thanked for technical assistance and advice; David Loydell and Eric Buffetaut are thanked for comments on an earlier version of this work. For discussion, opinions and identifications of Eotyrannus material, we thank Roger Benson, Chris Brochu, Steve Brusatte, Jonah Choiniere, Christophe Henrickx, Thomas R. Holtz Jr., Mathew Wedel and Peter Makovicky. We thank our reviewers–Jonah Choiniere and Corwin Sullivan–for numerous corrections, observations and suggestions which substantially improved the quality of the manuscript, and Steve Brusatte and Peter Makovicky for further comments and advice on the manuscript. For access to specimens we thank Angela Milner and Sandra Chapman (NHM) and Martin Munt, Alex Peaker, Dan Pemberton and Trevor Price (IWCMS). Steve Hutt (IWCMS) is thanked for his thoughts on theropod anatomy, functional morphology and evolution and for putting so much work into the preparation and interpretation of Eotyrannus. Chris Barker, Roger Benson, Dan Folkes, Christophe Henrickx and Alex Peaker are warmly thanked for providing images. The program TNT is made publicly available via the sponsorship of the Willi Hennig Society.

Institutional abbreviations

IWCMS Dinosaur Isle Visitor Centre, Isle of Wight County Museums Service, Sandown, UK

MIWG Museum of Isle of Wight Geology, Sandown, UK (collection now incorporated into that of IWCMS)

NHMUK The Natural History Museum, London, UK

UOP University of Portsmouth, Portsmouth, UK. NB–IWCMS accession numbers have been published in a number of different ways. Naish, Hutt & Martill, 2001 and Hutt (2002) used the convention ‘IWCMS.1997.550’ and Hutt et al. (2001) used the convention ‘IWCMS 1997.550’. The preferred way (D. Pemberton, pers. comm. 2003) is ‘IWCMS: 1997.550’ and this convention is adopted here.

Additional Information and Declarations

Competing Interests

Author Contributions

Data Availability

The authors declare that they have no competing interests.

Darren Naish conceived and designed the experiments, performed the experiments, analyzed the data, prepared figures and/or tables, authored or reviewed drafts of the paper, references, editing, making changes post-review, preparing Appendices, and approved the final draft.

Andrea Cau conceived and designed the experiments, performed the experiments, analyzed the data, prepared figures and/or tables, authored or reviewed drafts of the paper, references, editing, making changes post-review, preparing Appendices, and approved the final draft.

The following information was supplied regarding data availability:

The phylogenetic scores is available in Appendix 3.

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
