# Peer review of "The osteology and affinities of Eotyrannus lengi, a tyrannosauroid theropod from the Wealden Supergroup of southern England"

_PeerJ, doi:10.7717/peerj.12727_

## Round 0.1 · original submission · Minor Revisions

Thank you for your patience during the review process. Although I had reviews in-hand from a previous journal submission, because the overall comments (and subsequent changes) amounted to major revisions, I elected to send it out for one additional round of review. The previous reviewers graciously accepted the invitation, and agree that the manuscript has been much improved. Remaining suggestions are relatively minor on the whole, and I expect this will not likely require another round of review.

Particular points to note:

- Regarding the figures, some additional labeling is strongly advised. Reviewer 1 recommends some additional figure improvements, which should be considered but are not mandatory. Note that both reviewers call attention to the palatine illustrations and description, and this must be addressed in revision.
- Both reviewers suggest a little more attention to some of the more novel phylogenetic results for tyrannosauroids; please consider augmenting the relevant text slightly.
- Consider including a table for measurements of the appendicular skeleton, as done for the rest of the skeleton.
- If possible, I would suggest swapping out the grayscale photographs for color photographs. Knowing the long history of this manuscript and the fact that the original files may not exist, I recognize that this may be easier said than done. However, I do think that color photos might illustrate the fossils and their morphology more effectively.
- The reviewers' other comments should all be considered during revision.

·

Basic reporting

This article gives a monographic description of the anatomy of an Early Cretaceous tyrannosauroid. It is an important contribution to science, and provides data useful for theropod systematists. The manuscript also presents the results of a large phylogenetic analysis and gives a testable hypothesis for the phylogenetic placement for Eotyrannus. The supplemental data are clear, unambiguous, and support the authors conclusions.

I read a previous version of the manuscript, and it has been greatly improved since that version. Some issues still remain, however, and I summarize these below.

Structurally, the authors should consider incorporating a phylogenetic methods section before the morphological description, and providing a separate phylogenetic results section after the description. These are currently conjoined in the discussion section. The authors should also consider moving the emended diagnosis to the section immediately after "systematic palaeontology" to conform to a more typical layout, however I don't think this is a mandatory change.

The figures remain an issue as many of the line drawings come through poorly in the digital medium. Given that this is a definitive monograph, the authors should strive to improve the figures wherever possible. This can particularly be done through the use of photographic images paired side-by-side with interpretive drawings. It can also be done by varying line weights on the illustrations to give more depth to the images. In some of the illustrations, the font sizes and the leader line weights are inconsistent, and I found this distracting. On the plate for the palatine, the figure caption did not jibe with the descriptive text. I find the authors' argument in their cover letter regarding the prohibitive cost of improving the images to be a poor one - travel to the Isle of Wight is reasonable for UK citizens and free programs like GIMP and Inkscape make image editing essentially free in the modern era. However, I also understand that the images were done as part of a degree before these options were available at low cost. I recommend that the authors include as many images as possible (even if just in the supplement) to back up their descriptions.

The phylogenetic results are surprising: support for a Tyrannosauroid relationship for Megaraptora is not particularly common, and the groupings at the base of Tyrannosauroidea are, to my knowledge, unique. The authors don't delve into the former at all, and spend essentially no time explaining the implications of the latter. They should dedicate a section of the discussion to explaining these relationships in greater detail, providing the supporting homology information. The methodology in the phylogenetic section is also insufficient to replicate their search parameters - however I was able to verify their results in TNT relatively easily thanks to the attached nexus file in the supplement.

The palatine description still has many issues, which I have flagged on the manuscript itself. It needs to be completely reworked.

It would be beneficial to present a table of measurements for appendicular bones, including the circumference of the stylopodial and zeugopodial elements. These are tremendously useful for reconstructing body mass and locomotor performance and would increase the citeability of the paper.

There are many minor grammatical errors in the document, some of which require clarification, some of which are matters of writer's preference, and some of which require simple typo correction. I have flagged these in the attached document and made comment where necessary to explain myself.

Referencing is thorough, and I have suggested one or two places to make additional citations in the document.

Experimental design

This constitutes original primary research, with a well-defined research question that fills a identified gap in our knowledge of tyrannosaur evolution.

The methods section needs to be presented before the results, and should include a more detailed description of the phylogenetic analysis (see above). Currently the information is insufficient to exactly replicate the authors' search parameters.

Validity of the findings

I have analysed the phylogenetic data myself, and I have spent time looking at this specimen and nearly all of the comparative material. The authors' phylogenetic result is replicable (given their dataset), and it is my opinion that their morphological observations are objective.

Additional comments

Thank you for your attention to the many comments I made on a previous version of the article. The research has improved greatly and I think this will make a valuable contribution. I urge you to go through each bone description paying great attention to wording and clarity - I think that many sections could be substantially reduced in length without reducing content by making sentence structure more efficient. Some descriptive sections suffer from poor organization - for example the humerus description contains general information about the deltopectoral crest buried after specific information. Efficiency and precision in language and a bit of reorganization will greatly improve the readability.

Your discussion is relatively brief, but I do understand that this is primarily a descriptive manuscript. Still, I would like to see an expansion on the phylogenetic relationships of early branching tyrannosauroids and perhaps a bit more morphological discussion e.g., on the hand and on the tyrannosauroid features of Eotyrannus.

Finally, I still urge you to do your best to improve the figures. As a fellow systematist, I can tell you that many times I either seen features not included in line drawings, or doubted my scoring when only a line drawing was available. If you can simply include some photographs of every element, that would be a significant improvement.

·

Basic reporting

The article is well-organised and well-written, and I've made only small editorial suggestions in my "general comments" below. The illustrations are generally clear and informative, but in my opinion would benefit in many cases from slightly more extensive labelling – this will save readers some time when they try to locate features described in the text on the figures. In my "general comments" I’ve pointed out some specific features that would be worth labelling, but I’d suggest that the authors make their own pass through the manuscript with this issue in mind and try to provide the reader with a bit more help in interpreting the figures correctly. The paper does a very good job of explaining how the current redescription of Eotyrannus fits into the context established by previous discoveries, and relevant literature is copiously and appropriately cited throughout.

Experimental design

The paper's objectives - to comprehensively describe the osteology of Eotyrannus, compare this taxon to other tyrannosauroids and to other theropod material from the Isle of Wight, and to use the new anatomical information brought to light in the paper as a basis for reassessing the phylogenetic position of Eotyrannus - should be obvious to anyone used to reading palaeontological papers but are never quite stated explicitly in the Introduction. I recommend adding such a statement. The methods of phylogenetic analysis used in the paper are adequately described, and pretty standard in any case.

Validity of the findings

The anatomical interpretations in the paper mostly seem quite sound to me, but the description of the incomplete (right?) palatine given on pp. 25-26 is difficult to follow and to reconcile with the illustration on Figure 7 (which is upside-down, with ventral towards the top of the page, if the labelling of the vomeropterygoid, maxillary and jugal processes is correct). The pterygoid process is described in the first paragraph of the section on the palatine as a “broken” structure that would originally have been “far longer”, but as a “fan-shaped process” that has sustained only limited damage (to its exposed surface and “dorsal” end) in the last paragraph. The fan-shaped process must be the one that is labelled in Figure 7b as bearing the jugal articulation, and in my opinion is indeed more likely to represent the jugal process than the pterygoid (medial) process; the latter is probably buried within the block containing the incomplete palatine. In any case, the description of the palatine should be revised to be clearer about the identifications of the various processes and more consistent with Figure 7. More extensive labelling of the figure might also be helpful in this respect. I’d encourage the authors to reconsider the possibility of adding the palatine to the reconstruction in Figure 25, since I don’t see why the preserved portion of this bone couldn’t be included as has been done for the ilium and several other incomplete skeletal elements. Furthermore, the treatment of the palatine-based possible autapomorphies of E. lengi (p. 71, lines 2157-2166) may need to be altered after the description of the palatine has been revised as suggested above. I’m also a little suspicious of the identification of the putative distal portion of the left tibia, but I have no convincing alternative identification to offer.

The comparisons to other Isle of Wight theropods are clearly explained, and I agree with the authors' conclusion that no other fossils known from the Isle are at all likely to be referable to Eotyrannus. The phylogenetic analysis presented here will certainly not be the last word on tyrannosauroid relationships, but the results are for the most part unsurprising and highly plausible. The discussion of the analysis makes some good points about the implications of recovering megaraptorans within Tyrannosauroidea, but fails to touch on a couple of other unusual results that did crop up, particularly the recovery of Tugulusaurus as a coelurid tyrannosauroid and of Dilong as a proceratosaurid. Commenting on what may have caused these results, and how reliable they are likely to be, would be highly worthwhile.

Additional comments

I should note that I reviewed an earlier version of this manuscript several years ago, and that most of the suggestions I made at that time are reflected the current submission. I am happy to recommend publication of this paper in PeerJ, following minor revisions to correct the problems (almost all of which are quite small) pointed out in this review.

Apart from the (slightly) larger concerns mentioned above, I have a long list of minor comments regarding specific passages in the manuscript. All page numbers refer to the PDF copy that I received for review.

1. (p. 5, line 12) Consider replacing the rather colloquial expression “as goes” with something like “with regard to”. This also applies to a couple of subsequent uses of “as goes” within the text. Delete “a” before “subadult”.
2. (p. 7, line 51) Consider changing “additional specimens” to “newly discovered specimens”.
3. (p. 8, line 84) I’d suggest replacing “Siberia in western Russia” with either “western Siberia, Russia” or just “western Russia” to avoid appearing to imply that all of Siberia is in western Russia.
4. (p. 8, line 99) Delete “as” before “outside”.
5. (p. 8, line 107) Consider changing “mostly been interpreted” to “most often been interpreted”.
6. (p. 9, line 114) Change “renders” to “render”.
7. (p. 9, line 122) Change “uncertainly” to “uncertainty”.
8. (p. 9, line 128) “and to be closer to the latter than Proceratosaurus and Guanlong” could probably be deleted, since this goes without saying if E. lengi and Dilong form a polytomy with Xiongguanlong and Tyrannosauridae.
9. (p. 9, line 137) Change “feature” to “features”.
10. (p. 9, line 138) Change “though” to “though they”.
11. (p. 11, lines 202-203) Consider changing “fragmentary and/or broken” to “broken or even fragmentary”, since a fragmentary bone is broken by definition. Also consider changing “suffers from being” to “is”, on the grounds that the fossil itself (unlike preparators and palaeontologists who might wish to extract more information from it) is probably not suffering very much from the hardness of the matrix.
12. (p. 12, line 215) Change “question as to” to “question of”. More importantly, the case being made in this paragraph for regarding the holotype of E. lengi as a subadult seems a little weak. The only evidence adduced is the separation of the neural arches from the centra, and that would be compatible with a juvenile condition as well as a subadult one. Is there any compelling reason to infer that the specimen was a subadult rather than a juvenile at the time of death?
13. (p. 12, line 220) Consider changing “distorted during diagenesis” to “diagenetically distorted”.
14. (p. 13, line 244) Delete “this” before “resulting”.
15. (p. 13, line 246) It would be worth giving a brief definition of the premaxillary angle.
16. (p. 13, line 259) For clarity, it would be worth labelling the groove mentioned here on Figure 3.
17. (p. 13, lines 271-275) Are the tooth “locations” mentioned here defined with reference to the mesial edges of the bases of the tooth crowns? This point seems worth addressing given the high level of precision with which the locations are being specified. Also, if the presence of interdental plates is uncertain it seems a bit misleading to label them (at least without a question mark) in Figure 3b.
18. (p. 14, line 286) Probably only one of the words “tiny” and “fine” is necessary here.
19. (p. 14, line 287) Consider replacing “lateral margin of the premaxillary body” with something like “ventral margin of the premaxillary body on the lateral side”.
20. (p. 14, line 295) Delete the word “margins” after “anterodorsal”.
21. (p. 14, line 299) Is the “dorsally convex” shape of the preserved part of the wall of the antorbital fossa natural, or a result of breakage? I assume the latter, but this should be made clear.
22. (p. 15, lines 316-322) Consider labelling the probable contact area on the maxilla for the nasal, and/or the part of the margin of the maxilla that probably contributed to the external naris.
23. (p. 16, line 346) Change “short medial extension” to something like “limited medial prominence”.
24. (p. 16, line 365) Replace “are covered with” with “show”.
25. (p. 17, line 392) If I’m understanding correctly, it might be better to replace “lateral surface of the maxilla” with something like “prominent rim of the antorbital fossa”.
26. (p. 18, line 409) Change “nasal thought” to “naris though”, and “tip” to “tips”.
27. (p. 18, lines 424-425) The “ventral premaxillary process” mentioned here (and indicated in Figure 5) must be the same structure as the “subnarial process” mentioned elsewhere, but one term should be used throughout the description of the nasal. Similarly, the other anterior process of the nasal is called the “medial premaxillary process” in Figure 5 and the “medial process” on p. 19, line 444, but the “premaxillary process” elsewhere.
28. (p. 19, lines 461-462) Consider changing “the ventral side of the recess, however, bears” to “the ventral edge of the recess, however, is defined by”.
29. (p. 19, lines 470-471) I’d suggest deleting the last sentence of this paragraph and changing “vertical lamina” to “subvertical, although posterodorsally inclined, lamina” in the preceding sentence.
30. (p. 20, line 478) Looks like a period, or maybe “to them” followed by a period, is needed after “non-homologous”.
31. (p. 20, line 481) Consider changing “it actually seems that they belong” to “they more likely belong”.
32. (p. 20, line 496) Change “edges” to “edge”.
33. (p. 21, line 505) The word “exact” is unnecessary here.
34. (p. 21, lines 517-518) If in fact the posterolateral “prong-like structures” formed by the nasals had no “direct relationship” with the lacrimals, I assume they must have articulated instead with the frontals, but it would be worth making this clear.
35. (p. 21, line 519) Consider changing “a concave area is present” to “lies a concave area”.
36. (p. 22, line 545) Consider changing “more recalls” to “more closely recalls”.
37. (p. 22, lines 549-550) I’d suggest changing the wording here to “…indicates that the descending process of the lacrimal was somewhat posterodorsally inclined in life”.
38. (p. 22, line 556) The words “Viewed anteriorly” seem unnecessary here.
39. (p. 23, line 571) I’d suggest changing “In anterior view, the descending ramus” to “The anterior face of the descending ramus”.
40. (p. 23, lines 574-575) Is it possible to determine whether the “large, ovoid concavity” mentioned here is a relatively shallow blind recess or a foramen that penetrates deeply into the bone?
41. (p. 23, line 594) It might be clearer to say that the medial ridges on the descending ramus of the lacrimal are “closer to”, rather than “more associated with”, the posterior edge of the descending ramus than the anterior edge. In the next sentence, the statement that the ridge is “a feature” of the anterior edge in Alioramus and Appalachiosaurus also seems needlessly vague. The wording of this paragraph also gives the impression that there are two ridges in Eotyrannus but only a single ridge in at least some other tyrannosauroids, and this point should be clarified.
42. (p. 23, line 599) It might be worth labelling the two foramina mentioned here in Figure 6.
43. (p. 24, lines 617-618) For clarity, I’d suggest changing “position for this” to “position to represent the prefrontal” and “facet” to “prefrontal facet”.
44. (p. 25, lines 634-635) I’d suggest mentioning the articulation between the larger jugal fragment and the lacrimal sooner in the description of the jugal, before discussing the implications of this articulation for the orientation of the lacrimal (lines 631-632).
45. (p. 25, line 641) Consider changing “posterior part of the element” to “posterior part of the same bone”, just to avoid repeating “element”.
46. (p. 25, lines 649-650) Change “Little data… is available” to “Few data… are available”.
47. (p. 25, line 672) It might be worth labelling the “sinuous ridge” mentioned here in Figure 7.
48. (p. 26, lines 700-708) The text here repeatedly uses the term “quadrate fenestra”, but this feature is labelled in Figure 8 as a “quadrate foramen”, and a single term should be used consistently. I assume the “concavity” mentioned on line 700 is identical to the “quadrate fossa” labelled in Figure 8, but this should be made clear.
49. (p. 26, line 719) The “medial fossa” labelled here is simply labelled “dep” in Figure 8, but perhaps something like “mfos” would be more appropriate. In any case, “dep” does not appear in the list of abbreviations in the caption.
50. (p. 26, line 726) If possible, the large foramen mentioned here should be labelled in Figure 8. I would also recommend labelling the pterygoid process of the quadrate in Figure 8b, even if the process is not very apparent in the anterior view shown in that figure panel.
51. (p. 26, lines 728-729) In Figure 8d the medial condyle appears much closer to being perpendicular to the mediolateral axis of the quadrate than is the lateral condyle. The long axes of the two condyles are certainly not parallel, so they should not be described as being oriented at similar angles.
52. (p. 27, line 733) The words “it is” could be omitted here (and again on line 738).
53. (p. 28, line 752) Change “color” to “colour”, for consistency with the British spelling conventions used elsewhere in the manuscript.
54. (p. 29, line 791) Change “similar and better preserved than that on the left dentary” to “similar to that on the left dentary, but better preserved”.
55. (p. 29, lines 796-797) Above (line 777) the text states that “at least four” furrows are present. Is one of the five furrows difficult to confirm as a genuine feature? If so, this should be briefly explained. Also, line 796 should probably refer to the left dentary, not simply the dentary.
56. (p. 30, line 803) I’d suggest changing “flexion” to “flexure” or “bending”.
57. (p. 30, lines 811-812) I assume the anteromedially located ridge and groove described here are in fact symphyseal features for articulation with the opposite dentary, but it would be worth making this clear.
58. (p. 30, line 818) Assuming the “shallow medial groove” described here is distinct from the Meckelian groove, it should be labelled in Figure 9. If the “shallow medial groove” is in fact the Meckelian groove, this should be clearly indicated.
59. (p. 30, line 828) Change “interdental plate” to “interdental plates”.
60. (p. 30, line 830) Only three interdental plates are clearly visible in Figure 9b, and Figure 9f the number of plates is difficult to determine from the figure. Labelling the individual plates by number, or at least having a line extend from the “intpl” label to every plate, would clarify matters.
61. (p. 31, line 846) Consider replacing “at its dorsal end” with “along its dorsal edge”.
62. (p. 31, lines 847-848) I assume the text here should refer to a “retroarticular process”, singular.
63. (p. 31, lines 856-857) In figure 10e the surangular shelf appears to trend anteroventrally, a point that would be worth mentioning.
64. (p. 31, line 861) Change the colon on this line to a semicolon.
65. (p. 31, line 863) Change “that is typical” to “than is typical”.
66. (p. 32, last paragraph) The “unidentified” bone described here as a possible articular is identified as the anterior section of the left surangular in the caption to Figure 10a-c. I have my doubts about both identifications, but I have no convincing alternative to propose. In any case, the figure caption and text should be consistent with regard to the identification of this element.
67. (p. 33, line 909) I assume the “length” mentioned here was measured anteroposteriorly, but it would be worth making this clear.
68. (p. 33, lines 919-920) I’d suggest giving here the explanation provided below (p. 69, lines 2090-2093) for why the teeth are being described as U-shaped rather than D-shaped in cross-section.
69. (p. 33, line 924) Change “color” to “colour”.
70. (p. 33, line 928) Change “texture looks appears to represent” to “texture that appears to have resulted from”.
71. (p. 34, line 942) Delete the words “the base of” before “c. 2 mm”.
72. (p. 34, last paragraph) It seems odd to use the phrase “lateral teeth” to refer to maxillary and/or dentary teeth, and I’d suggest that this bit of terminology (which is non-standard, as far as I know) be reconsidered. Also, I don’t see why “teeth still embedded in the left… dentary” (line 953) are being included in the category of “remaining lateral teeth”, given that the left dentary teeth were described just above.
73. (p. 34, line 960) Delete the phrase “and without apical hooking”, as this point was made previously in the sentence.
74. (p. 35, line 963) Consider replacing “unworn denticles” with “unworn distal denticles”, just for clarity.
75. (p. 36, lines 996-998) The anteroposteriorly broad morphology of the neural spine, evident in Figure 12, may be another indication that the vertebra being considered here is the axis.
76. (p. 36, line 1000) Delete the word “preserved” after “facet”. Also, the space between the prezygapophyses appears more subtriangular than subcircular in Figure 12b.
77. (p. 36, line 1006) If the spine is indeed “complete apically excepting its posterodorsal portion” (line 1002), then the neural spine apex can be “missing” only posterodorsally and the transverse flaring, if originally present, must have been limited to the posterodorsal part of the spine. This point would probably be worth briefly explaining. Also, the word “in” on this line should be written in lower case letters.
78. (p. 36, line 1013) “postzygapophyseal” is misspelled here.
79. (p. 36, line 1015) Delete the words “is preserved” at the end of this line.
80. (p. 37, line 1031) Change “their precise” to “the precise”.
81. (p. 37, line 1041) Consider replacing “lateral concavity” with “concave articular surface”, assuming it is indeed the articular surface of the parapophysis that is concave.
82. (p. 37, line 1045) Consider changing “axial foramen on each side” to “foramen on each side of the axis”.
83. (p. 37, line 1050) Change “surface” to “surfaces”.
84. (p. 38, line 1062) I assume the “centrum” referred to here is the second, shorter cervical centrum rather than the potential axial centrum, but this should be clarified.
85. (p. 38, line 1063) Does the reference here to the “broken end” of the rib fragment imply that the other end is not broken? Or is it just that only one end is exposed?
86. (p. 38, lines 1073-1074) Despite the statement here that five centra “seem to” belong to dorsal vertebrae of E. lengi, only four dorsal centra are unambiguously attributed to E. lengi in the description on this page and the following one. This apparent contradiction should be resolved. If the “highly pyritised” fragment mentioned on p. 40 is considered to represent the fifth dorsal centrum attributable to E. lengi, as is perhaps implied but not stated explicitly, this should be made clear.
87. (p. 38, line 1076) The word “Table” is misspelled here.
88. (p. 38, line 1079) Consider changing “vertebrae of these positions” to “these vertebrae”.
89. (p. 38, lines 1087-1088) Consider changing “the bony rim that surrounds it is more prominent ventrally, laterally and dorsally than that” to “the ventral, lateral and dorsal portions of the bony rim that surrounds it are more prominent posteriorly than the corresponding parts of the rim”, if that is indeed the intended meaning.
90. (p. 39, line 1127) Consider changing “The most likely identification is that it is from” to “It is likely from”.
91. (p. 41, line 1162) It might be worth explaining the reason for the “reinterpretation” mentioned here. What is the evidence that the “heavily scarred” articular surface of the centrum is actually the posterior one?
92. (p. 42, line 1212) It would be worth indicating the orientation of the concavity that separates the postzygapophysis from the centrum, by using a descriptor such as “distally facing”.
93. (p. 42, line 1214) Change “chevrons facets” to “chevron facets”.
94. (p. 43, lines 1239-1240) Consider changing “the rib’s identity as that of E. lengi” to “that the rib belongs to E. lengi”.
95. (p. 43, line 1246) If the rib shown in Figure 14a can indeed be confidently assessed as being exposed in anterior view (which I believe is the case), then it must be from the left side of the body given the orientations of the capitulum and tuberculum, contradicting the earlier statement (lines 1229-1230) that the ribs cannot be assigned to one side or the other.
96. (p. 44, line 1282) The word “made” seems unnecessary here.
97. (p. 44, lines 1284-1285) I’d suggest clarifying that the scapula is attached to only a small piece of the coracoid, since “partial articulation” could mean that the scapula retains a partial attachment to the complete coracoid.
98. (p. 44, line 1288) It would be worth clarifying whether the “small, anterodorsally projecting bump” mentioned here is the same as the “acromion process” labelled in Figure 15e. In any case, the final two sentences of this paragraph should be moved to an earlier position and integrated into the rest of the description of the flared ventral end of the scapula.
99. (p. 45, line 1290) Insert “the” before “acromion region”.
100. (p. 45, line 1309) Should the word “longer” at the end of this line be replaced by “wider”? The previous sentence describes a difference in width between the two ends, not a difference in length.
101. (p. 45, lines 1317-1318) Consider changing “scapula that is narrow ventrally and wider in its dorsal part” to “scapular blade that is narrow ventrally and gradually widens dorsally”, in order to clarify that the ventrally “narrow” condition seen in tyrannosaurids does not imply a reduced acromion and that the dorsal widening involves much of the blade rather than just the dorsal end (as in E. lengi). I’d also recommend changing “lacks subparallel edges” to “widens gradually in the dorsal direction” in the brief comparison with the scapula of Dilong.
102. (p. 46, line 1327) Change the colon on this line to a semicolon; also delete “of the rest”.
103. (p. 46, line 1331) The words “strongly convex ventrolaterally” seem like an exaggeration, given that the ventrolaterally-facing edge of the coracoid in Figure 15a,c appears only slightly convex.
104. (p. 46, lines 1332-1333) Insert “and” before “subtriangular”, and a comma after “view”.
105. (p. 47, line 1361) Change “more” to “more closely”.
106. (p. 47, lines 1363-1364) I’d suggest changing “it is angled medially relative to the humeral long axis” to “its flexor surface is angled somewhat medially”.
107. (p. 47, lines 1374-1375) I’d suggest labelling the internal tuberosity in Figure 16.
108. (p. 47, line 1387) For clarity, consider changing “overhangs the shaft” to something like “overhangs the adjacent part of the anterior face of the humerus”.
109. (p. 48, line 1397) Delete the word “broken”.
110. (p. 48, lines 1399-1405) Perhaps I’m being obtuse, but I still (to revisit an issue that came up in my review of the original version of this paper) don’t see how “greater bone thickness in the anterior part of the shaft associated with the presence of the deltopectoral crest” could be a sufficient explanation for an “apparent internal demarcation” between the main cavity within the shaft and smaller cavities within the area of thickened bone just distal to the crest. In principle, why couldn’t such an area of thickened bone contain either (1) no internal cavities at all or (2) a narrow anterior extension of the main cavity within the shaft? If the small internal cavities happen to be within a thickening of the humeral shaft associated with the deltopectoral crest, that’s certainly worth mentioning, but in my opinion the presence of the thickened area shouldn’t be offered an as explanation for the fact that the small cavities exist and are separated from the main cavity within the shaft. If this part of the text is retained, however, change “between then” to “between them” on line 1404.
111. (p. 48, line 1415) I’d suggest citing Figures 16a and 16f here, as well as 16b, since all three show the entepicondyle.
112. (p. 49, line 1427) Change “form” to “forms”.
113. (p. 49, line 1436) Delete the word “long”.
114. (p. 49, line 1439) Change “is is” to “it is”.
115. (p. 49, line 1441) Change the colon on this line to a semicolon.
116. (p. 50, lines 1479-1480) I don’t think I’d describe any of the views shown in Figure18c-f as “subrectangular”, given that d and f show a good deal of constriction whereas c and e are much broader and more bulbous at one end than the other.
117. (p. 51, line 1489) Change “this concave region” to “and this concave region is”.
118. (p. 51, lines 1490-1491) In Figure 18e, which I tentatively interpret as a ventral (flexor) view, it looks as though the distal rim of the ventral surface of the carpal is actually concave (not merely less convex than the proximal rim, as the wording here sort of implies).
119. (p. 51, lines 1499-1500) If the “base” of a metacarpal is meant to be the same thing as the “proximal surface”, I’d suggest using one term consistently in this sentence to avoid implying a distinction. If there is a meaningful distinction between the two terms, a brief explanation should be provided.
120. (p. 51, line 1504) Change “carpal I” to “distal carpal I”.
121. (p. 51, lines 1507-1508) It would be worth noting that the distal carpal I of E. lengi, being complex and trochleated in form, is compatible with the suggestion of Holtz referred to here.
122. (p. 51, line 1513) For clarity, change “Both are preserved” to “Both proximal ends are preserved”.
123. (p. 51, line 1518 to p. 52, line 1519) In figure 18j the longest axis of the proximal end of MC I appears to be oblique rather than strictly mediolateral. In any case I don’t see how the mediolateral axis can be considered the longest if its also “shorter than the dorsoventral height”.
124. (p. 52, line 1521) Insert “the” before “cross-sectional”.
125. (p. 52, line 1541) Insert “of” before “the compsognathid”.
126. (p. 52, lines 1542-1543) Does the statement about early tyrannosauroids having an especially gracile MC I shaft hold even in the light of the observation mentioned above (line 1536) that Guanlong has a more robust MC I shaft than E. lengi? It would be worth commenting on this point specifically.
127. (p. 52, line 1544) Insert “the” before “left mc II”.
128. (p. 53, line 1564) Consider replacing “arranged” with “aligned”.
129. (p. 54, line 1588) Change “its good level of articulation” to something like “the relatively precise articulation”.
130. (p. 54, line 1593) Replace “mc 1” with “mc I”.
131. (p. 54, line 1599) In Figure 19e (evidently a ventral view mislabelled as dorsal) the putative flexor processes appear to lie medial and lateral, rather than proximomedial and proximolateral, to the proximal concavity on the ventral surface.
132. (p. 54, lines 1606-1607) In Figure 19l the proximal articular surface appears considerably higher than wide.
133. (p. 55, lines 1617-1618) I’d suggest noting which scapula the phalanx is attached to, and also mentioning the phalanx in the description of the scapula given above.
134. (p. 55, line 1623) In Figure 19n the ventral convexity appears to be the more prominent of the two on the left side of the proximal surface, contrary to what is stated here.
135. (p. 55, line 1638) For clarity, I’d suggest replacing “ventral convexity” with something like “ventrally situated convexity”.
136. (p. 55, lines 1640-1641) Figures 19u,x indicate that the right collateral ligament fossa is larger than the left, contrary to what is stated here.
137. (p. 55, line 1647) Replace “trochlea” with something like “distal convexity”, since the trochlea is formed by both condyles together. It’s incorrect to speak of each condyle as having its own trochlea.
138. (p. 56, line 1653) Change “this is unlikely but not impossible” to “it is unlikely but not impossible that the present phalanx represents III-1”.
139. (p. 56, line 1672) Consider changing “convex” to “transversely convex”.
140. (p. 56, line 1674) Change “continuous” to “continuously”.
141. (p. 56, line 1676) I’m not sure how a groove can be “prominent”, which to me implies “protruding”. Would something like “sharply defined” be preferable here? This issue comes up again on the next page, in line 1699.
142. (p. 57, line 1688) The steeply angled slope distal to the flexor tubercle wasn’t noted in the description of the putative pollical ungual given above, and it might be worth inserting a brief mention.
143. (p. 57, line 1689) Change “variation is” to “variation in”.
144. (p. 57, line 1693) Consider replacing “surrounds” with “overhangs”.
145. (p. 57, lines 1701-1702) The description of the putative pollical ungual implied that an overlapping dorsal lamina was present only in the case of the left groove, rather than on both sides. Is this a difference between the two unguals?
146. (p. 57, line 1709) Is phalanx I-1 of Eotyrannus “shorter” than those of Allosaurus and tyrannosaurids in proportion to its own diameter, or in proportion to some other measurement? If the shortness is in proportion to diameter or circumference, then this phalanx at least is presumably more robust than in tyrannosaurids, rather than intermediate between tyrannosaurid-like and Dilong-like (or maniraptoran-like) conditions as stated above.
147. (p. 57, lines 1714-1715) It would be worth indicating whether there is no proximoventral projection at all on phalanx II-2 in Deinonychus, or whether a projection is present but simply differs in shape from the one seen in Eotyrannus.
148. (p. 58, lines 1716-1718) The widening of the ventral part of the proximal articular surface of phalanx II-2 in Guanlong isn’t simply a “result” of the surface being concave both laterally and medially, but rather reflects the specific shapes of the two concavities. It might be better to just say here that the phalanx II-2 of Guanlong differs from that of Eotyrannus in having a proximal surface that has concave lateral and medial sides and is almost twice as wide ventrally as dorsally.
149. (p. 58, lines 1730-1731) I’d suggest replacing “undergoing reduction” with “partially reduced” or “incipiently reduced”, since these alternatives sound less teleological.
150. (p. 58, line 1740) Is the “concavity” mentioned on this line identical to the “demarcation between the convex dorsal surface of the ungual and the rim” referred to on p. 57, lines 1693-1694?
151. (p. 58, line 1748) I’d suggest writing “ridge in the middle of the sheet” rather than “median ridge”, since the ridge isn’t “median” in an anatomical sense (i.e. it isn’t on the mediolateral midline of the body, or that of an individual bone). On lines 1749, 1765 and 1780, the word “median” could simply be deleted.
152. (p. 59, line 1751) Change “confirmation the the” to “confirmation that the”.
153. (p. 59, line 1755) Change “in entirety” to “in its entirety”.
154. (p. 59, line 1770) Consider changing “the ridge is also posterodorsally inclined” to “a posterodorsally inclined ridge above the acetabulum is present”, since not all theropods have a ridge in this position.
155. (p. 59, line 1773) Delete the words “is present”.
156. (p. 61, line 1815) Change “simpler that” to “simpler than”.
157. (p. 61, line 1817) Change “condylars” to “condyles”.
158. (p. 61, lines 1823-1824) The wording here seems to imply that the fibular crest of Eotyrannus is more similar in some way to that of Juratyrant than to those of other coleurosaurs, but this point should be clarified.
159. (p. 61, line 1826) Consider replacing “adjacent and posterior” with “posteromedial”.
160. (p. 61, line 1830-1834) Given that it is indeed fairly “typical for theropod long bones” for the cortical bone to be composed of concentric layers, at least in my experience, I’d suggest deleting the description of this condition in Eotyrannus. If these sentences are retained, I’d suggest cleaning them up to get rid of the parentheses, which awkwardly contain about a sentence and a half.
161. (p. 61, lines 1844-1846) The wording here could be changed to “…described here in order of position from medial to lateral.”
162. (p. 62, line 1849) Should “distal surface” be changed to “anterior surface”?
163. (p. 62, line 1853) Change “appear” to “appears”, and delete the word “present”.
164. (p. 62, lines 1860-1862) In my opinion it’s an exaggeration to say that the “only obvious difference” between the distal part of the tibia in the only known specimens of Eotyrannus and Tugulusaurus is the damage to the distolateral corner of the bone in the former. Based on illustrations in Rauhut and Xu (2005), Tugulusaurus has a more proximally extensive fossa (the astragalar facet) on the anterior surface of the tibia than does Eotyrannus, as well as a medial expansion of the distal end of the tibia that is at best poorly developed in Eotyrannus. Tugulusaurus also appears to lack the groove proximal to the lateral “shoulder” of the distal end of the tibia mentioned in the text.
165. (p. 62, lines 1863-1864) Figure 22b also seems to show a ridge medial to the groove on the posterior surface of the distal end of the tibia, unless I’m misinterpreting the photo.
166. (p. 63, line 1880) Is the “fragment” mentioned here the one illustrated in Figure 21b? If so, this should probably be pointed out. Also, I don’t see how this fragment can be both “separate from the block” and “preserved adjacent to the shaft of the tibia”, assuming the “block” mentioned here is the one in which the proximal part of the tibia is preserved.
167. (p. 63, line 1899) Insert a comma before “and the deep”.
168. (p. 64, line 1921) Delete the words “the presence of”.
169. (p. 64, line 1922) I’m not sure what is meant by “distal continuation” of the anterolateral surface. If the ridge is on this surface, how could the ridge be responsible for its “continuation”?
170. (p. 65, line 1955) The distal end of the left MT III certainly does not appear to be “convex on all sides” in Figure 23k.
171. (p. 65, line 1959) Insert “and” between the dimensions of the right and left collateral ligament fossae.
172. (p. 65, line 1961) I’d suggest specifying which side of the putative right MT III is exposed. If there are multiple possibilities, they can be briefly enumerated.
173. (p. 65, line 1962) Delete the word “wide”.
174. (p. 65, line 1972) Fix the spelling of “Although”.
175. (p. 65, lines 1977-1978) Consider replacing “the proximal end of the bone is convex, passing into” with something like “the proximal end of the medial surface is convex, but passes distally into”.
176. (p. 66, line 1996) Replace “pedal unguals” with “pedal phalanges”.
177. (p. 66, line 2000) Delete the word “it’s”.
178. (p. 66, line 2004) I assume the “deep concavity” mentioned here is on the ventral side of the phalanx, but it might be worth making this clear.
179. (p. 66, line 2005) The words “deeper and” seem redundant here.
180. (p. 66, line 2009) Consider changing “dorsoproximally” to “proximally on the dorsal surface”.
181. (p. 66, line 2010) Would “is deflected” or “is angled” be better than “extends”?
182. (p. 67, line 2021) For better fit with the rest of the sentence, change “concave, subcircular” to “concave and subcircular”.
183. (p. 67, line 2022) Change “It is not biconcave” to “Its proximal surface is not biconcave”.
184. (p. 67, line 2024) Change “ventral part” to “ventral surface”.
185. (p. 67, line 2026) Change “others” to “other”.
186. (p. 67, lines 2027-2028) Consider changing “Ventrally, the part of the shaft” to “The part of the ventral surface of the shaft”.
187. (p. 67, lines 2034-2036) It would be worth commenting explicitly here on whether there is any good reason to assign the small phalanx to the pes of Eotyrannus, rather than to the dryosaurid.
188. (p. 68, line 2058) Consider moving the words “is present” to just after “concavity”.
189. (p. 68, lines 2061-2062) Can the lateral surface really be distinguished from the medial surface? If not, it might be better to say “lateral or medial groove” on line 2061 and “right surface” on line 2062. Similarly, “right side view” might be more appropriate than “right lateral view” on line 2054. If the right side is definitely the lateral one, this should be changed to simply “lateral view” and it should be noted that the ungual belongs to the right pes.
190. (p. 69, lines 2094-2096) I’m sceptical of the claim that apically complete denticulation “should probably be considered typical within coelurosaurs”. In my experience, there are many coelurosaur teeth that show well-developed denticulation on the distal carina but few if any denticles on the mesial carina, and none around the apex. While I can’t swear that apically incomplete denticulation is the more common condition in Coelurosauria, I don’t think the opposite should be asserted without a citation or some kind of more detailed explanation. In any case it would be sufficient here to point out that apically complete denticulation is widespread in tyrannosauroids, which is certainly the case.
191. (p. 70, line 2127) Change “possesses” to “possess”.
192. (p. 70, line 2133) Change “articulate” to “articular”.
193. (p. 70, line 2139) Delete “unusual”, which is redundant with “unique”.
194. (p. 71, line 2147) Delete the words “member of those”.
195. (p. 71, line 2152) The word “represent” is misspelled here.
196. (p. 71, lines 2154-2155) Delete the words “is an additional possible autapomorphy”.
197. (p. 71, line 2160) Change “tyrannosauroid” to “tyrannosauroids” and “data is” to “data are”.
198. (p. 71, lines 2167-2168) Delete “presence of” and change “to the shafts” to “of the shafts”.
199. (p. 72, lines 2177-2178) I’d suggest indicating explicitly here whether the morphology of the proximodorsal lip in Tanycolagreus is comparable to that seen in E. lengi, or whether this structure is more “subtly developed” in Tanycolagreus than in E. lengi.
200. (p. 72, line 2207) Change “both of” to “of both”, and delete the word “of” before “any reported”.
201. (p. 73, lines 2237-2238) The parenthetical comment “no such structure is present in E. lengi” should be deleted, since this point is made in the following sentence.
202. (p. 73, line 2240) Delete the word “ventrally”, which is unnecessary given that it has already been established that the ventral surface is the one being referred to.
203. (p. 74, line 2251) Delete the words “mean that”.
204. (p. 74, line 2267) The word “separated” is misspelled here.
205. (p. 74, line 2268) The specimen number for one of the femora mentioned on this line is given as MIWG 6214 here, but MIWG 6124 in the caption to figure 26. Also, the text is incorrect in citing panels w-z of this figure; panels v-x actually depict MIWG 6214, whereas y and z are the first two views of the tibia MIWG 5137.
206. (p. 75, line 2292) Consider changing “outgroups” to “outgroup taxa”.
207. (p. 75, line 2296) Change “calculated” to “calculated by”.
208. (p. 76, line 2331) Delete the word “as” before “outside”.
209. (p. 76, line 2335) Change “biased by” to “a result of” or perhaps “an artefact of”.
210. (p. 76, line 2337-2338) It might be worth pointing out explicitly that the results of the phylogenetic analysis imply that the distinctive supracetabular ridge morphology seen in Eotyrannus and Juratyrant probably evolved independently in the two taxa.
211. (p. 77, line 2345) Change “ita” to “its” or preferably “the”.
212. (p. 77, line 2346) Insert “the” before “medially fused nasals”.
213. (p. 77, line 2353) Change “where the longest axis” to “whose longest axis”.
214. (p. 77, line 2354) Consider changing “arranged” to “aligned”.
215. (p. 77, line 2357) Change “participation to” to “participation in”.
216. (p. 78, line 2380) Consider changing “both stratigraphically and morphologically significant gaps” to “a stratigraphically and morphologically significant gap”, since I assume there is really just one gap between the Jurassic to Early Cretaceous tyrannosauroids and the Late Cretaceous ones.
217. (p. 78, line 2383) Change “the conclusion of Brusatte et al. (2016), who suggested” to “the suggestion of Brusatte et al. (2016)”.
218. (p. 78, line 2400) Change “hypothesis” to “issue” or “question”.
219. (p. 79, lines 2417-2419) I’d suggest deleting this sentence about the locomotion and predatory abilities of Eotyrannus. No functional analysis was included in the main part of the manuscript, so introducing such interpretations into the conclusions seems inappropriate. This also applies to the comment in the following sentence about powerful biting.
220. (p. 79, line 2425) Insert a period after “skeleton”.
221. (p. 80, line 2449) Delete the word “consistent”, and change “strong” to “close”.
222. (Figure 11) The photos in panels A and B of this figure appear slightly out of focus, and should be replaced with sharper images if possible.
223. (Figure 18) some of the views of distal carpal I are mislabelled and/or in orientations that seem likely to prove confusing. I would suggest rotating panels A, B, D and F so that the dorsal direction is towards the top of the page in each case, and rotating C and E so that the distal surface is towards the top (though I suppose this is a matter of taste). As far as I can see, the views in C-F are all misidentified (they should be, in order, dorsal, lateral, ventral and medial), and some of the abbreviations listed in the caption do not match the ones used in the actual figure. In panels J-L of the same figure, the lateral and medial condyles of MC I are consistently labelled backwards.
224. (Figure 22) In Panel D, the distalmost part of the tibia appears to be flipped over relative to the rest of the bone (at least judging by panel A, in which the fit between the distalmost part and the adjacent part is much better), and the proximalmost part appears misoriented and/or enlarged relative to the bone shaft.
225. (Figure 26) Is the structure labelled in panel C as a “ventrolateral groove” actually one of the “ventrolateral ridges” mentioned on p. 74, line 2247?
226. (Figure 28) The abbreviations listed in the caption do not match the ones used in the figure.

External reviews were received for this submission. These reviews were used by the Editor when they made their decision, and can be downloaded below.

---

## Round 0.2 · accepted · Accept

Thank you for your considerable work to revise this latest version of the manuscript, and for your patience as I looked over this revision. I have no additional comments at this point.